# On the Curse of Memory in Recurrent Neural Networks: Approximation and Optimization Analysis

**Zhong Li**[†]
School of Mathematical Science
Peking University
li_zhong@pku.edu.cn

**Jiequn Han**[†]
Department of Mathematics
Princeton University
jiequnh@princeton.edu

**Weinan E**
Department of Mathematics and PACM
Princeton Univeristy
weinan@math.princeton.edu

**Qianxiao Li**[‡]
Department of Mathematics
National University of Singapore
IHPC, A*STAR, Singapore
qianxiao@nus.edu.sg

## Abstract

We study the approximation properties and optimization dynamics of recurrent neural networks (RNNs) when applied to learn input-output relationships in temporal data. We consider the simple but representative setting of using continuous-time linear RNNs to learn from data generated by linear relationships. Mathematically, the latter can be understood as a sequence of linear functionals. We prove a universal approximation theorem of such linear functionals and characterize the approximation rate. Moreover, we perform a fine-grained dynamical analysis of training linear RNNs by gradient methods. A unifying theme uncovered is the non-trivial effect of memory, a notion that can be made precise in our framework, on both approximation and optimization: when there is long-term memory in the target, it takes a large number of neurons to approximate it. Moreover, the training process will suffer from slow downs. In particular, both of these effects become exponentially more pronounced with increasing memory - a phenomenon we call the "curse of memory". These analyses represent a basic step towards a concrete mathematical understanding of new phenomenons that may arise in learning temporal relationships using recurrent architectures.

## 1 Introduction

Recurrent neural networks (RNNs) (Rumelhart et al., 1986) are among the most frequently employed methods to build machine learning models on temporal data. Despite its ubiquitous application (Baldi et al., 1999; Graves & Schmidhuber, 2009; Graves, 2013; Graves et al., 2013; Graves & Jaitly, 2014; Gregor et al., 2015), some fundamental theoretical questions remain to be answered. These come in several flavors. First, one may pose the *approximation* problem, which asks what kind of temporal input-output relationships can RNNs model to an arbitrary precision. Second, one may also consider the *optimization* problem, which concerns the dynamics of training (say, by gradient descent) the RNN. While such questions can be posed for any machine learning model, the crux of the problem for RNNs is how the recurrent structure of the model and the dynamical nature of the data shape the answers to these problems. For example, it is often observed that when there are long-term dependencies in the data (Bengio et al., 1994; Hochreiter et al., 2001), RNNs may encounter problems in learning, but such statements have rarely been put on precise footing.

In this paper, we make a step in this direction by studying the approximation and optimization properties of RNNs. Compared with the static feed-forward setting, the key distinguishing feature

---

[†]Equal contribution
[‡]Corresponding author

here is the presence of temporal dynamics in terms of both the recurrent architectures in the model and the dynamical structures in the data. Hence, to understand the influence of dynamics on learning is of fundamental importance. As is often the case, the key effects of dynamics can already be revealed in the simplest linear setting. For this reason, we will focus our analysis on linear RNNs, i.e. those with linear activations. Further, we will employ a continuous-time analysis initially studied in the context of feed-forward architectures (E, 2017; Haber & Ruthotto, 2017; Li et al., 2017) and recently in recurrent settings (Ceni et al., 2019; Chang et al., 2019; Lim, 2020; Sherstinsky, 2018; Niu et al., 2019; Herrera et al., 2020; Rubanova et al., 2019) and idealize the RNN as a continuous-time dynamical system. This allows us to phrase the problems under investigation in convenient analytical settings that accentuates the effect of dynamics. In this case, the RNNs serve to approximate relationships represented by sequences of linear functionals. On first look the setting appears to be simple, but we show that it yields representative results that underlie key differences in the dynamical setting as opposed to static supervised learning problems. In fact, we show that memory, which can be made precise by the decay rates of the target linear functionals, can affect both approximation rates and optimization dynamics in a non-trivial way.

Our main results are:

1. We give a systematic analysis of the approximation of linear functionals by continuous-time linear RNNs, including a precise characterization of the approximation rates in terms of regularity and memory of the target functional.

2. We give a fine-grained analysis of the optimization dynamics when training linear RNNs, and show that the training efficiency is adversely affected by the presence of long-term memory.

These results together paint a comprehensive picture of the interaction of learning and dynamics, and makes concrete the heuristic observations that the presence of long-term memory affects RNN learning in a negative manner (Bengio et al., 1994; Hochreiter et al., 2001). In particular, mirroring the classical curse of dimensionality (Bellman, 1957), we introduce the concept of the *curse of memory* that captures the new phenomena that arises from learning temporal relationships: when there is long-term memory in the data, one requires an exponentially large number of neurons for approximation, and the learning dynamics suffers from exponential slow downs. These results form a basic step towards a mathematical understanding of the recurrent structure and its effects on learning from temporal data.

## 2 RELATED WORK

We will discuss related work on RNNs on three fronts concerning the central results in this paper, namely approximation theory, optimization analysis and the role of memory in learning. A number of universal approximation results for RNNs have been obtained in discrete (Matthews, 1993; Doya, 1993; Schäfer & Zimmermann, 2006; 2007) and continuous time (Funahashi & Nakamura, 1993; Chow & Xiao-Dong Li, 2000; Li et al., 2005; Maass et al., 2007; Nakamura & Nakagawa, 2009). Most of these focus on the case where the target relationship is generated from a hidden dynamical system in the form of difference or differential equations. The formulation of functional approximation here is more general, albeit our results are currently limited to the linear setting. Nevertheless, this is already sufficient to reveal new phenomena involving the interaction of learning and dynamics. This will be especially apparent when we discuss approximation rates and optimization dynamics. We also note that the functional/operator approximation using neural networks has been explored in Chen & Chen (1993); Tianping Chen & Hong Chen (1995); Lu et al. (2019) for non-recurrent structures and reservoir systems for which approximation results similar to random feature models are derived (Gonon et al., 2020). The main difference here is that we explicitly study the effect of memory in target functionals on learning using recurrent structures.

On the optimization side, there are a number of recent results concerning the training of RNNs using gradient methods, and they are mostly positive in the sense that trainability is proved under specific settings. These include recovering linear dynamics (Hardt et al., 2018) or training in over-parameterized settings (Allen-Zhu et al., 2019). Here, our result concerns the general setting of learning linear functionals that need not come from some underlying differential/difference equations, and is also away from the over-parameterized regime. In our case, we discover on the contrary

that training can become very difficult even in the linear case, and this can be understood in a quantitative way, in relation to long-term memory in the target functionals.

This points to the practical literature in relation to memory and learning. The dynamical analysis here puts the ubiquitous but heuristic observations - that long-term memory negatively impacts training efficiency (Bengio et al., 1994; Hochreiter et al., 2001) - on concrete theoretical footing, at least in idealized settings. This may serve to justify or improve current heuristic methods (Tseng et al., 2016; Dieng et al., 2017; Trinh et al., 2018) developed in applications to deal with the difficulty in training with long-term memory. At the same time, we also complement general results on "vanishing and explosion of gradients" (Pascanu et al., 2013; Hanin & Rolnick, 2018; Hanin, 2018) that are typically restricted to initialization settings with more precise characterizations in the dynamical regime during training.

The long range dependency within temporal data has been studied for a long time in the time series literature, although its effect on learning input-output relationships is rarely covered. For example, the Hurst exponent (Hurst, 1951) is often used as a measure of long-term memory in time series, e.g. fractional Brownian motion (Mandelbrot & Ness, 1968). In contrast with the setting in this paper where memory involves the dependence of the output time series on the input, the Hurst exponent measures temporal variations and dependence within the input time series itself. Much of the time series literature investigates statistical properties and estimation methods of data with long range dependence (Samorodnitsky, 2006; Taqqu et al., 1995; Beran, 1992; Doukhan et al., 2003). One can also combine these classic statistical methodologies with the RNN-like architectures to design hybrid models with various applications (Loukas & Oke, 2007; Diaconescu, 2008; Mohan & Gaitonde, 2018; Bukhari et al., 2020).

## 3 PROBLEM FORMULATION

The basic problem of supervised learning on time series data is to learn a mapping from an input temporal sequence to an output sequence. Formally, one can think of the output at each time as being produced from the input via an unknown function that depends on the entire input sequence, or at least up to the time at which the prediction is made. In the discrete-time case, one can write the data generation process

$$y_k = H_k(x_0, \ldots, x_{k-1}), \qquad k = 0, 1, \ldots \tag{1}$$

where $x_k, y_k$ denote respectively the input data and output response, and $\{H_k : k = 0, 1, \ldots\}$ is a sequence of ground truth functions of increasing input dimension accounting for temporal evolution. The goal of supervised learning is to learn an approximation of the sequence of functions $\{H_k\}$ given observation data.

Recurrent neural networks (RNN) (Rumelhart et al., 1986) gives a natural way to parameterize such a sequence of functions. In the simplest case, the one-layer RNN is given by

$$h_{k+1} = \sigma(Wh_k + Ux_k), \qquad \hat{y}_k = c^\top h_k. \tag{2}$$

Here, $\{h_k\}$ are the *hidden/latent* states and its evolution is governed by a recursive application of a feed-forward layer with activation $\sigma$, and $\hat{y}_k$ is called the observation or readout. We omit the bias term here and only consider a linear readout or output layer. For each time step $k$, the mapping $\{x_0, \ldots, x_{k-1}\} \mapsto \hat{y}_k$ parameterizes a function $\hat{H}_k(\cdot)$ through adjustable parameters $(c, W, U)$. Hence, for a particular choice of these parameters, a sequence of functions $\{\hat{H}_k\}$ is constructed at the same time. To understand the working principles of RNNs, we need to characterize how $\{\hat{H}_k\}$ approximates $\{H_k\}$.

The model (2) is not easy to analyze due to its discrete iterative nature. Hence, here we employ a continuous-time idealization that replaces the time-step index $k$ by a continuous time parameter $t$. This allows us to employ a large variety of continuum analysis tools to gain insights to the learning problem. Let us now introduce this framework.

**Continuous-time formulation.** Consider a sequence of inputs indexed by a real-valued variable $t \in \mathbb{R}$ instead of a discrete variable $k$ considered previously. Concretely, we consider the input space

$$\mathcal{X} = C_0(\mathbb{R}, \mathbb{R}^d), \tag{3}$$

which is the linear space of continuous functions from $\mathbb{R}$ (time) to $\mathbb{R}^d$ that vanishes at infinity. Here $d$ is the dimension of each point in the time series. We denote an element in $\mathcal{X}$ by $\boldsymbol{x} := \{x_t \in \mathbb{R}^d : t \in \mathbb{R}\}$ and equip $\mathcal{X}$ with the supremum norm $\|\boldsymbol{x}\|_{\mathcal{X}} := \sup_{t \in \mathbb{R}} \|x_t\|_\infty$. For the space of outputs we will take a scalar time series, i.e. the space of bounded continuous functions from $\mathbb{R}$ to $\mathbb{R}$:

$$\mathcal{Y} = C_b(\mathbb{R}, \mathbb{R}). \tag{4}$$

This is due to the fact that vector-valued outputs can be handled by considering each output separately. In continuous time, the target relationship (ground truth) to be learned is

$$y_t = H_t(\boldsymbol{x}), \qquad t \in \mathbb{R} \tag{5}$$

where for each $t \in \mathbb{R}$, $H_t$ is a functional $H_t : \mathcal{X} \to \mathbb{R}$. Correspondingly, we define a continuous version of (2) as a hypothesis space to model continuous-time functionals

$$\frac{d}{dt} h_t = \sigma(W h_t + U x_t), \qquad \hat{y}_t = c^\top h_t, \tag{6}$$

whose Euler discretization corresponds to a discrete-time residual RNN. The dynamics then naturally defines a sequences of functionals $\{\hat{H}_t(\boldsymbol{x}) = \hat{y}_t : t \in \mathbb{R}\}$, which can be used to approximate the target functionals $\{H_t\}$ via adjusting $(c, W, U)$.

**Linear RNNs in continuous time.** In this paper we mainly investigate the approximation and optimization property of linear RNNs, which already reveals the essential effect of dynamics. The linear RNN obeys (6) with $\sigma$ being the identity map. Notice that in the theoretical setup, the initial time of the system goes back to $-\infty$ with $\lim_{t \to -\infty} x_t = 0, \forall \boldsymbol{x} \in \mathcal{X}$, thus by linearity ($H_t(\boldsymbol{0}) = 0$) we specify the initial condition of the hidden state $h_{-\infty} = 0$ for consistency. In this case, (6) has the following solution

$$\hat{y}_t = \int_0^\infty c^\top e^{Ws} U x_{t-s} ds. \tag{7}$$

Since we will investigate uniform approximations over large time intervals, we will consider stable RNNs, where $W \in \mathcal{W}_m$ with

$$\mathcal{W}_m = \{W \in \mathbb{R}^{m \times m} : \text{eigenvalues of } W \text{ have negative real parts}\}. \tag{8}$$

Owing to the representation of solutions in (7), the linear RNN defines a family of functionals

$$\hat{\mathcal{H}} := \cup_{m \geq 1} \hat{\mathcal{H}}_m,$$

$$\hat{\mathcal{H}}_m := \left\{ \{\hat{H}_t(\boldsymbol{x}), t \in \mathbb{R}\} : \hat{H}_t(\boldsymbol{x}) = \int_0^\infty c^\top e^{Ws} U x_{t-s} ds, W \in \mathcal{W}_m, U \in \mathbb{R}^{m \times d}, c \in \mathbb{R}^m \right\}. \tag{9}$$

Here, $m$ is the width of the network and controls the complexity of the hypothesis space. Clearly, the family of functionals the RNN can represent is not arbitrary, and must possess some structure. Let us now introduce some definitions of functionals that makes these structures precise.

**Definition 3.1.** *Let $\{H_t : t \in \mathbb{R}\}$ be a sequence of functionals.*

1. *$H_t$ is* causal *if it does not depend on future values of the input: for every pair of $\boldsymbol{x}, \boldsymbol{x}' \in \mathcal{X}$ such that $x_s = x'_s$ for all $s \leq t$, we have $H_t(\boldsymbol{x}) = H_t(\boldsymbol{x}')$.*

2. *$H_t$ is* linear *and continuous if $H_t(\lambda \boldsymbol{x} + \lambda' \boldsymbol{x}') = \lambda H_t(\boldsymbol{x}) + \lambda' H_t(\boldsymbol{x}')$ for any $\boldsymbol{x}, \boldsymbol{x}' \in \mathcal{X}$ and $\lambda, \lambda' \in \mathbb{R}$, and $\sup_{\boldsymbol{x} \in \mathcal{X}, \|\boldsymbol{x}\|_{\mathcal{X}} \leq 1} |H_t(\boldsymbol{x})| < \infty$, in which case the induced norm can be defined as $\|H_t\| := \sup_{\boldsymbol{x} \in \mathcal{X}, \|\boldsymbol{x}\|_{\mathcal{X}} \leq 1} |H_t(\boldsymbol{x})|$.*

3. *$H_t$ is* regular *if for any sequence $\{\boldsymbol{x}(n) \in \mathcal{X} : n \in \mathbb{N}\}$ such that $x(n)_t \to 0$ for Lebesgue almost every $t \in \mathbb{R}$, $\lim_{n \to \infty} H_t(\boldsymbol{x}(n)) = 0$.*

4. *$\{H_t : t \in \mathbb{R}\}$ is* time-homogeneous *if $H_t(\boldsymbol{x}) = H_{t+\tau}(\boldsymbol{x}(\tau))$ for any $t, \tau \in \mathbb{R}$, where $x(\tau)_s = x_{s-\tau}$ for all $s \in \mathbb{R}$, i.e. $\boldsymbol{x}(\tau)$ is $\boldsymbol{x}$ whose time index is shifted to the right by $\tau$.*

Linear, continuous and causal functionals are common definitions. One can think of regular functionals as those that are not determined by values of the inputs on an arbitrarily small time interval, e.g. an infinitely thin spike input. Time-homogeneous functionals, on the other hand, are those where there is no special reference point in time: if the time index of both the input sequence and the functional are shifted in coordination, the output value remains the same. Given these definitions, the following observation can be verified directly and its proof is immediate and hence omitted.

**Proposition 3.1.** *Let $\{\hat{H}_t : t \in \mathbb{R}\}$ be a sequence of functionals in the RNN hypothesis space $\hat{\mathcal{H}}$ (see (9)). Then for each $t \in \mathbb{R}$, $\hat{H}_t$ is a causal, continuous, linear and regular functional. Moreover, the sequence of functionals $\{\hat{H}_t : t \in \mathbb{R}\}$ is time-homogeneous.*

## 4 Approximation Theory

The most basic approximation problem for RNN is as follows: given some sequence of target functionals $\{H_t : t \in \mathbb{R}\}$ satisfying appropriate conditions, does there always exist a sequence of RNN functionals $\{\hat{H}_t : t \in \mathbb{R}\}$ in $\hat{\mathcal{H}}$ such that $H_t \approx \hat{H}_t$ for all $t \in \mathbb{R}$?

We now make an important remark with respect to the current problem formulation that differs from previous investigations in the RNN approximation: we are **not** assuming that the target functionals $\{H_t : t \in \mathbb{R}\}$ are themselves generated from an underlying dynamical system of the form

$$H_t(\boldsymbol{x}) = y_t \qquad \text{where} \qquad \dot{h}_t = f(h_t, x_t), \qquad y_t = g(h_t) \tag{10}$$

for any linear or nonlinear functions $f, g$. This differs from previous work where it is assumed that the sequence of target functionals are indeed generated from such a system. In that case, the approximation problem reduces to that of the functions $f, g$, and the obtained results often resemble those in feed-forward networks.

In our case, however, we consider general input-output relationships related by temporal sequences of functionals, with no necessary recourse to the mechanism from which these relationships are generated. This is more flexible and natural, since in applications it is often not clear how one can describe the data generation process. Moreover, notice that in the linear case, if the target functionals $\{H_t\}$ are generated from a linear ODE system, then the approximation question is trivial: as long as the dimension of $h_t$ in the approximating RNN is greater than or equal to that which generates the data, we must have perfect approximation. However, we will see that in the more general case here, this question becomes much more interesting, even in the linear regime. In fact, we now prove precise approximation theories and characterize approximation rates that reveal intricate connections with memory effects, which may be otherwise obscured if one considers more limited settings.

Our first main result is a converse of Proposition 3.1 in the form of an universal approximation theorem for certain classes of linear functionals. The proof is found in Appendix A.

**Theorem 4.1** (Universal approximation of linear functionals). *Let $\{H_t : t \in \mathbb{R}\}$ be a family of continuous, linear, causal, regular and time-homogeneous functionals on $\mathcal{X}$. Then, for any $\epsilon > 0$ there exists $\{\hat{H}_t : t \in \mathbb{R}\} \in \hat{\mathcal{H}}$ such that*

$$\sup_{t \in \mathbb{R}} \|H_t - \hat{H}_t\| \equiv \sup_{t \in \mathbb{R}} \sup_{\|\boldsymbol{x}\|_{\mathcal{X}} \leq 1} |H_t(\boldsymbol{x}) - \hat{H}_t(\boldsymbol{x})| \leq \epsilon. \tag{11}$$

The proof relies on the classical Riesz-Markov-Kakutani representation theorem, which says that each linear functional $H_t$ can be uniquely associated with a signed measure $\mu_t$ such that $H_t(\boldsymbol{x}) = \int_{\mathbb{R}} x_s^\top d\mu_t(s)$. Owing to the assumptions of Theorem 4.1, we can further show that the sequence of representations $\{\mu_t\}$ are related to an integrable function $\rho : [0, \infty) \to \mathbb{R}^d$ such that $\{H_t\}$ admits the common representation

$$H_t(\boldsymbol{x}) = \int_0^\infty x_{t-s}^\top \rho(s) ds, \qquad t \in \mathbb{R}, \quad \boldsymbol{x} \in \mathcal{X}. \tag{12}$$

Comparing this representation with the solution (7) of the continuous RNN, we find that the approximation property of the linear RNNs is closely related to how well $\rho(t)$ can be approximated by the exponential sums of the form $(c^\top e^{Wt} U)^\top$. Intuitively, (12) says that each output $y_t = H_t(\boldsymbol{x})$ is simply a convolution between the input signal and the kernel $\rho$. Thus, the smoothness and decay of the input-output relationship is characterized by the convolution kernel $\rho$. Due to this observation, we will hereafter refer to $\{H_t\}$ and $\rho$ interchangeably.

**Approximation rates.** While the previous result establishes the universal approximation property of linear RNNs for suitable classes of linear functionals, it does not reveal to us which functionals can be efficiently approximated. In the practical literature, it is often observed that when there is

some long-term memory in the inputs and outputs, the RNN becomes quite ill-behaved (Bengio et al., 1994; Hochreiter et al., 2001). It is the purpose of this section to establish results which make these heuristics statements precise. In particular, we will show that the rate at which linear functionals can be approximated by linear RNNs depends on the former's smoothness and memory properties. We note that this is a much less explored area in the approximation theory of RNNs.

To characterize smoothness and memory of linear functionals, we may pass to investigating the properties of their actions on constant input signals. Concretely, let us denote by $e_i$ $(i = 1, \ldots, d)$ the standard basis vector in $\mathbb{R}^d$, and $\boldsymbol{e}_i$ denotes a constant signal with $e_{i,t} = e_i \mathbf{1}_{\{t \geq 0\}}$. Then, smoothness and memory is characterized by the regularity and decay rate of the maps $t \mapsto H_t(\boldsymbol{e}_i)$, $i = 1, \ldots, d$, respectively. Our second main result shows that these two properties are intimately tied with the approximation rate. The proof is found in Appendix B.

**Theorem 4.2** (Approximation rates of linear RNN). *Assume the same conditions as in Theorem 4.1. Consider the output of constant signals $y_i(t) := H_t(\boldsymbol{e}_i)$, $i = 1, \ldots, d$. Suppose there exist constants $\alpha \in \mathbb{N}_+, \beta, \gamma > 0$ such that $y_i(t) \in C^{(\alpha+1)}(\mathbb{R})$, $i = 1, \ldots, d$, and*

$$e^{\beta t} y_i^{(k)}(t) = o(1) \text{ as } t \to +\infty, \quad \text{and} \quad \sup_{t \geq 0} \beta^{-k} |e^{\beta t} y_i^{(k)}(t)| \leq \gamma, \qquad k = 1, \ldots, \alpha + 1, \quad (13)$$

*where $y_i^{(k)}(t)$ denotes the $k^{th}$ derivative of $y_i(t)$. Then, there exists a universal constant $C(\alpha)$ only depending on $\alpha$, such that for any $m \in \mathbb{N}_+$, there exists a sequence of width-$m$ RNN functionals $\{\hat{H}_t : t \in \mathbb{R}\} \in \hat{\mathcal{H}}_m$ such that*

$$\sup_{t \in \mathbb{R}} \|H_t - \hat{H}_t\| \equiv \sup_{t \in \mathbb{R}} \sup_{\|\boldsymbol{x}\|_{\mathcal{X}} \leq 1} |H_t(\boldsymbol{x}) - \hat{H}_t(\boldsymbol{x})| \leq \frac{C(\alpha) \gamma d}{\beta m^\alpha}. \quad (14)$$

**The curse of memory in approximation.** For approximation of non-linear functions using linear combinations of basis functions, one often suffers from the "curse of dimensionality" (Bellman, 1957), in that the number of basis functions required to achieve a certain approximation accuracy increases exponentially when the dimension of input space $d$ increases. In the case of Theorem 4.2, the bound scales linearly with $d$. This is because the target functional possesses a linear structure, and hence each dimension can be approximated independently of others, resulting in an additive error estimate. Nevertheless, due to the presence of the temporal dimension, there enters another type of challenge, which we coin the *curse of memory*. Let us now discuss this point in detail.

The key observation is that the rate result requires exponential decay of derivatives of $H_t(\boldsymbol{e}_i)$, but the density result (Theorem 4.1) makes no such assumption. The natural question is thus, what happens when no exponential decay is present? We assume $d = 1$ and consider an example in which the target functional's representation satisfies $\rho(t) \in C^{(1)}(\mathbb{R})$ and $\rho(t) \sim t^{-(1+\omega)}$ as $t \to +\infty$. Here $\omega > 0$ indicates the decay rate of the memory effects in our target functional family. The smaller its value, the slower the decay and the longer the system memory. For any $\omega > 0$, the system's memory vanishes more slowly than any exponential decay. Notice that $y^{(1)}(t) = \rho(t)$ and in this case there exists no $\beta > 0$ making (13) true, and no rate estimate can be deduced from it.

A natural way to circumvent this obstacle is to introduce a truncation in time. With $T \, (\gg 1)$ we can define $\tilde{\rho}(t) \in C^{(1)}(\mathbb{R})$ such that $\tilde{\rho}(t) \equiv \rho(t)$ for $t \leq T$, $\tilde{\rho}(t) \equiv 0$ for $t \geq T + 1$, and $\tilde{\rho}(t)$ is monotonically decreasing for $T \leq t \leq T + 1$. With the auxiliary linear functional $\tilde{H}_t(\boldsymbol{x}) := \int_0^t x_{t-s} \tilde{\rho}(s) ds$, we can have an error estimate (with technical details provided in Appendix B)

$$\sup_{t \in \mathbb{R}} \|H_t - \hat{H}_t\| \leq \sup_{t \in \mathbb{R}} \|H_t - \tilde{H}_t\| + \sup_{t \in \mathbb{R}} \|\tilde{H}_t - \hat{H}_t\| \leq C\left(T^{-\omega} + \frac{\omega}{m} T^{1-\omega}\right). \quad (15)$$

In order to achieve an error tolerance $\epsilon$, according to the first term above we require $T \sim \epsilon^{-\frac{1}{\omega}}$, and then according to the second term we have

$$m = \mathcal{O}\left(\omega T^{1-\omega}/\epsilon\right) = \mathcal{O}\left(\omega \epsilon^{-1/\omega}\right). \quad (16)$$

This estimate gives us a quantitative relationship between the degree of freedom needed and the decay speed. With $\rho(t) \sim t^{-(1+\omega)}$, the system has long memory when $\omega$ is small. Denote the minimum number of terms needed to achieve an $L^1$ error $\epsilon$ as $m(\omega, \epsilon)$. The above estimate shows

an upper-bound of $m(\omega, \epsilon)$ goes to infinity exponentially fast as $\omega \to 0^+$ with fixed $\epsilon$. This is akin to the curse of dimensionality, but this time on memory, which manifests itself even in the simplest linear settings. A stronger result would be that the lower bound of $m(\omega, \epsilon) \to \infty$ exponentially fast as $\omega \to 0^+$ with fixed $\epsilon$, and this is a point of future work. Note that this kind of estimates differ from the previous results in the literature (Kammler, 1979b; Braess & Hackbusch, 2005) regarding the order of $m(\omega, \epsilon) \sim \log(1/\epsilon)$ as $\epsilon \to 0$ with fixed $\omega = 1$ or 2 in the $L^\infty$ or $L^1$ sense. Note that the $L^1$ result has not been proved.

## 5 Fine-grained Analysis of Optimization Dynamics

According to Section 4, memory plays an important role in determining the approximation rates. The result therein only depends on the model architecture, and does not concern the actual training dynamics. In this section, we perform a fine-grained analysis on the latter, which again reveals an interesting interaction between memory and learning.

The loss function (for training) is defined as

$$\mathbb{E}_{\boldsymbol{x}} J_T(\boldsymbol{x}; c, W, U) := \mathbb{E}_{\boldsymbol{x}} |\hat{H}_T(\boldsymbol{x}) - H_T(\boldsymbol{x})|^2 = \mathbb{E}_{\boldsymbol{x}} \left| \int_0^T [c^\top e^{Wt} U - \rho(t)^\top] x_{T-t} dt \right|^2. \quad (17)$$

Without loss of generality, here the input time series $x$ is assumed to be finitely cut off at zero, i.e. $x_t = 0$ for any $t \leq 0$ almost surely. Training the RNN amounts to optimizing $\mathbb{E}_{\boldsymbol{x}} J_T$ with respect to the parameters $(c, W, U)$. The most commonly applied method is gradient descent (GD) or its stochastic variants (say SGD), which updates the parameters in the steepest descent direction.

We first show that the training dynamics of $\mathbb{E}_{\boldsymbol{x}} J_T$ exhibits very different behaviors depending on the form of target functionals. Take $d = 1$ and consider learning different target functionals with white noise $\boldsymbol{x}$. We first investigate two choices for $\rho$: a simple decaying exponential sum and a scaled Airy function. The Airy function target is defined as $\rho(t) = \text{Ai}(s_0[t - t_0])$, where $\text{Ai}(t)$ is the Airy function of the first kind, given by the improper integral $\text{Ai}(t) = \frac{1}{\pi} \lim_{\xi \to \infty} \int_0^\xi \cos\left(\frac{u^3}{3} + tu\right) du$. Note that the effective rate of decay is controlled by the parameter $t_0$: for $t \leq t_0$, the Airy function is oscillatory. Hence for large $t_0$, a large amount of memory is present in the target.

Observe from Figure 1 that the training proceeds efficiently for the exponential sum case. However, in the second Airy function case, there are interesting "plateauing" behaviors in the training loss, where the loss decrement slows down significantly after some time in training. The plateau is sustained for a long time before an eventual reduction is observed.

As a further demonstration of that this behavior may be generic, we also consider a nonlinear forced dynamical system, the Lorenz 96 system (Lorenz, 1996), where the similar plateauing behavior is observed even for a non-linear RNN model trained with the Adam optimizer (Kingma & Ba, 2015). All experimental details are found in Appendix C.3.1.

The results in Figure 1 hint at the fact that there are certainly some functionals that are much harder to learn than others, and it is the purpose of the remaining analyses to understand precisely when and why such difficulties occur. In particular, we will again relate this to the memory effects in the target functional, which shows yet another facet of the *curse of memory* when it comes to optimization.

**Dynamical analysis.** To make analysis amenable, we will make a series of simplifications to the loss function (17), by assuming that $\boldsymbol{x}$ is white noise, $d = 1$, $T \to \infty$, and the recurrent kernel $W$ is diagonal. This allows us to write (see Appendix C.1 for details) the optimization problem as

$$\min_{a \in \mathbb{R}^m, w \in \mathbb{R}^m_+} J(a, w) := \int_0^\infty \left( \sum_{i=1}^m a_i e^{-w_i t} - \rho(t) \right)^2 dt. \quad (18)$$

We will subsequently see that these simplifications do not lose the key features of the training dynamics, such as the plateauing behavior. We start with some informal discussion on a probable reason behind the plateauing. A straightforward computation shows that, for $k = 1, 2, \ldots, m$,

$$\frac{\partial J}{\partial w_k}(a, w) = 2a_k \int_0^\infty (-t) e^{-w_k t} \left( \sum_{i=1}^m a_i e^{-w_i t} - \rho(t) \right) dt. \quad (19)$$

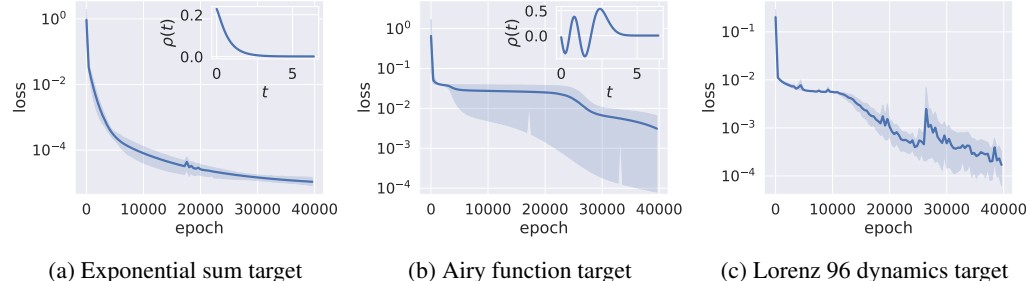

(a) Exponential sum target     (b) Airy function target     (c) Lorenz 96 dynamics target

Figure 1: Comparison of training dynamics on different types of functionals. (a) and (b): using the linear RNN model with the GD optimizer; (c): using the nonlinear RNN model (with $\tanh$ activation) with the Adam optimizer. The shaded region depicts the mean $\pm$ the standard deviation in 10 independent runs using randomized initialization. Observe that learning complex functionals (Airy, Lorenz) suffers from slow-downs in the form of long plateaus.

A similar expression holds for $\frac{\partial J}{\partial a_k}$. Write the (simplified) linear functional representation for linear RNNs as $\hat{\rho}(t; a, w) := \sum_{i=1}^{m} a_i e^{-w_i t}$, which serves to learn the target $\rho$. Observe that plateauing under the GD dynamics occurs if the gradient $\nabla J$ is small but the loss $J$ is large. A sufficient condition is that the residual $\hat{\rho}(t; a, w) - \rho(t)$ is large *only for* large $t$ (meaning the exponential multiplier to the residual is small). That is, the learned functional differs from the target functional only at large times. This again relates to the long-term memory in the target.

Based on this observation, we build this memory effect explicitly into the target functional by considering $\rho$ of the parametrized form

$$\rho_\omega(t) := \bar{\rho}(t) + \rho_{0,\omega}(t), \tag{20}$$

where $\bar{\rho}$ is the function which can be well-approximated by the model $\hat{\rho}$, e.g. the exponential sum $\bar{\rho}(t) = \sum_{j=1}^{m^*} a_j^* e^{-w_j^* t}$ (with $w_j^* > 0$, $j = 1, \cdots, m^*$). On the other hand, $\rho_{0,\omega}(t) := \rho_0(t - 1/\omega)$ controls the target memory, with $\rho_0$ as any bounded template function in $L^2(\mathbb{R}) \cap C^2(\mathbb{R})$ with sub-Gaussian tails. As $\omega \to 0^+$, the support of $\rho_{0,\omega}$ shifts towards large times, modelling the dominance of long-term memories. In this case, if the initialization satisfies $\hat{\rho} \approx \bar{\rho}$, the sufficient condition informally discussed above is satisfied as $\omega \to 0^+$, which heuristically leads to the plateauing.

A simple example of (20) can be $\rho_\omega(t) = a^* e^{-w^* t} + c_0 e^{-\frac{(t - 1/\omega)^2}{2\sigma^2}}$, where $a^*, c_0, \sigma \neq 0$ and $w^* > 0$ are fixed constants. This corresponds to the simple case that $m^* = 1$ and $\rho_0$ is the Gaussian density. Observe that as $\omega \to 0^+$, the memory of this sequence of functionals represented by $\rho_\omega$ increases. It can be numerically verified that this simple target functional gives rise to the plateauing behavior, which gets worse significantly as $\omega \to 0^+$ (see Figure 2 in Appendix C.3.2).

Our main result on training dynamics quantifies the plateauing behavior theoretically for general functionals possessing the decomposition (20). For rigorous statements and detailed proofs, see Theorem C.1 in Appendix C.2.

**Theorem 5.1.** *Define the loss function $J_\omega$ as in (18) with the target $\rho = \rho_\omega$ as defined in (20). Consider the gradient flow training dynamics*

$$\frac{d}{d\tau}\theta_\omega(\tau) = -\nabla J_\omega(\theta_\omega(\tau)), \quad \theta_\omega(0) = \theta_0, \tag{21}$$

*where $\theta_\omega(\tau) := (a_\omega(\tau), w_\omega(\tau)) \in \mathbb{R}^{2m}$ for any $\tau \geq 0$, and $\theta_0 := (a_0, w_0)$. For any $\omega > 0$, $m \in \mathbb{N}_+$ and $\theta_0 \in \mathbb{R}^m \times \mathbb{R}_+^m$, $0 < \delta \ll 1$, define the hitting time*

$$\tau_0 = \tau_0(\delta; \omega, m, \theta_0) := \inf\{\tau \geq 0 : |J_\omega(\theta_\omega(\tau)) - J_\omega(\theta_0)| > \delta\}. \tag{22}$$

*Assume that $m > m^*$, and the initialization is bounded and satisfies $\hat{\rho}(t; \theta_0) \approx \bar{\rho}(t)$. Then*

$$\tau_0(\delta; \omega, m, \theta_0) \gtrsim \omega^2 e^{c_0/\omega} \min\left\{\frac{\delta}{\sqrt{m}}, \ln(1 + \delta)\right\} \tag{23}$$

*for any $\omega > 0$ sufficiently small, where $c_0$ and $\gtrsim$ hide universal positive constants independent of $\omega$, $m$ and $\theta_0$.*

Let us sketch the intuition behind Theorem 5.1. Suppose that we currently have a good approximation $\hat{\rho}$ of the short-term memory part $\bar{\rho}$, then we can show that the loss is large ($J = \mathcal{O}(1)$) since the long-term memory part $\rho_{0,\omega}$ is not accounted for. However, the gradient now is small ($\nabla J = o(1)$), since the gradient corresponding to the long-term memory part is concentrated at large $t$, and thus modulated by exponentially decayed multipliers (see (19)). This implies slowdowns in the training dynamics in the region $\hat{\rho} \approx \bar{\rho}$. It remains to estimate a lower bound on the timescale of escaping from this region, which depends on the curvature of the loss function. In particular, we show that $\nabla^2 J$ is positive semi-definite when $\omega = 0$, but has $\mathcal{O}(1)$ positive eigenvalues and multiple $o(1)$ (can be exponentially small) eigenvalues for any $0 < \omega \ll 1$. Hence, a local linearization analysis implies an exponentially increasing escape timescale, as indicated in (23).

While the target form (20) may appear restrictive, we emphasize that some restrictions on the type of functionals is necessary, since plateauing does not always occur (see Figure 1). In fact, a goal of the preceding analysis is to establish a family of functionals for which exponential slowdowns in training *provably* occurs, and this can be related to memories of target functionals in a precise way.

**The curse of memory in optimization.** The timescale proved in Theorem 5.1 is verified numerically in Figure 3 in Appendix C.3.3, where we also show that the analytical setting here is representative of general cases, where plateauing occurs even for non-linear RNNs trained with accelerated optimizers, as long as the target functional has the memory structure imposed in (20).

Theorem 5.1 reveals another aspect of the *curse of memory*, this time in optimization. When $\omega \to 0^+$, the influence of target functional $H_t$ does not decay, much like the case considered in the curse of memory in approximation. However, different from the approximation case where an exponentially large number of hidden states is required to achieve approximation tolerance, here in optimization the adverse effect of memory comes with the exponentially pronounced slowdowns of the gradient flow training dynamics. While this is theoretically proved under sensible but restrictive settings, we show numerically in Appendix C.3.3 (Figure 4) that this is representative of general cases.

In the literature, a number of results have been obtained pertaining to the analysis of training dynamics of RNNs. A positive result for training by GD is established in Hardt et al. (2018), but this is in the setting of identifying hidden systems, i.e. the target functional comes from a linear dynamical system, hence it must possess good decay properties provided stablity. On the other hand, convergence can also be ensured if the RNN is sufficiently over-parameterized (large $m$; Allen-Zhu et al. (2019)). However, both of these settings may not be sufficient in reality. Here we provide an alternative analysis of a setting that is representative of the difficulties one may encounter in practice. In particular, the curse of memory that we established here is consistent with the difficulty in RNN training often observed in applications, where heuristic attributions to memory are often alluded to Hu et al. (2018); Campos et al. (2017); Talathi & Vartak (2015); Li et al. (2018). The analysis here makes the connection between memories and optimization difficulties precise, and may form a basis for future developments to overcome such difficulties in applications.

## 6 CONCLUSION

In this paper, we analyzed the basic approximation and optimization aspects of using RNNs to learn input-output relationships involving temporal sequences in the linear, continuous-time setting. In particular, we coined the concept *curse of memory*, and revealed two of its facets. That is, when the target relationship has the long-term memory, both approximation and optimization become exceedingly difficult. These analyses make concrete heuristic observations of the adverse effect of memory on learning RNNs. Moreover, it quantifies the interaction between the structure of the model (RNN functionals) and the structure of the data (target functionals). The latter is a much less studied topic. Here, we adopt a continuous-time approach in order gain access to more quantitative tools, including classical results in approximation theory and stochastic analysis, which help us derive precise results in approximation rates and optimization dynamics. The extension of these results to discrete time may be performed via numerical analysis in subsequent work. More broadly, this approach may be a basic starting point for understanding learning from partially observed time series data in general, including gated RNN variants (Hochreiter & Schmidhuber, 1997; Cho et al., 2014) and other methods such as transformers and convolution-based approaches (Vaswani et al., 2017; Oord et al., 2016). These are certainly worthy of future exploration.

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

## A  Universal Approximation Theorem of Linear Functionals by Linear RNNs

A key simplification of considering linear functionals is due to the classical representation result below, which allows us to pass from the approximation of functionals to the approximation of functions. In short, this theorem says that while for each measure $\mu$, $\boldsymbol{x} \mapsto \int_{\mathbb{R}} x_s^\top d\mu(s)$ defines a linear functional, this is in fact the *only* way to define a linear functional: for any linear functional $H$ there exists a unique measure $\mu$ such that $H(\boldsymbol{x}) = \int_{\mathbb{R}} x_s^\top d\mu(s)$.

**Theorem A.1** (Riesz-Markov-Kakutani representation theorem). *Let $H : \mathcal{X} \to \mathbb{R}$ be a continuous linear functional. Then, there exists a unique, vector-valued, regular, countably additive signed measure $\mu$ on $\mathbb{R}$ such that*

$$H(\boldsymbol{x}) = \int_{\mathbb{R}} x_s^\top d\mu(s) = \sum_{i=1}^{d} \int_{\mathbb{R}} x_{s,i} d\mu_i(s). \tag{24}$$

*Moreover, we have*

$$\|H\| := \sup_{\|\boldsymbol{x}\|_{\mathcal{X}} \le 1} |H(\boldsymbol{x})| = \|\mu\|_1(\mathbb{R}) := \sum_i |\mu_i|(\mathbb{R}). \tag{25}$$

*Proof.* Well-known, see e.g. Bogachev (2007), CH 7.10.4. □

We will use the representation theorem to prove Theorem 4.1. First, we prove some lemmas. The first shows that there is in fact a *common* representation of a sequence of linear functionals satisfying the assumptions of Theorem 4.1.

**Lemma A.1.** *Let $\{H_t\}$ be a family of continuous, linear, regular, causal and time homogeneous functionals on $\mathcal{X}$. Then, there exists a measurable function $\rho : [0, \infty) \to \mathbb{R}^d$ that is integrable, i.e.*

$$\|\rho\|_{L^1([0,\infty))} := \sum_{i=1}^{d} \int_0^\infty |\rho_i(s)| ds < \infty \tag{26}$$

*and*

$$H_t(\boldsymbol{x}) = \int_0^\infty x_{t-s}^\top \rho(s) ds, \qquad t \in \mathbb{R}. \tag{27}$$

*In particular, $\{H_t\}$ is uniformly bounded with $\sup_t \|H_t\| = \|\rho\|_{L^1([0,\infty))}$ and $t \mapsto H_t(\boldsymbol{x})$ is continuous for all $\boldsymbol{x} \in \mathcal{X}$.*

*Proof.* By the Riesz-Markov-Kakutani representation theorem (Theorem A.1), for each $t$ there is a unique regular signed Borel measure $\mu_t$ such that

$$H_t(\boldsymbol{x}) = \int_{\mathbb{R}} x_s^\top d\mu_t(s), \tag{28}$$

and $\sum_i |\mu_{t,i}|(\mathbb{R}) = \|H_t\|$. Since $\{H_t\}$ is causal, we must have $\int_t^\infty x_s^\top d\mu_t(s) = 0$ for any $\boldsymbol{x}$ and thus

$$H_t(\boldsymbol{x}) = \int_{-\infty}^{t} x_s^\top d\mu_t(s). \tag{29}$$

Now, by time homogeneity we have

$$\int_{-\infty}^{t} x_s^\top d\mu_t(s) = H_t(\boldsymbol{x}) = H_{t+\tau}(\boldsymbol{x}^{(\tau)}) = \int_{-\infty}^{t+\tau} x_{s-\tau}^\top d\mu_{t+\tau}(s). \tag{30}$$

Take $\tau = -t$ and set $\mu = -\mu_0$ to get

$$H_t(\boldsymbol{x}) = \int_0^\infty x_{t-s}^\top d\mu(s). \tag{31}$$

Note that we have $\|\mu\|_1([0,\infty)) = \|\mu_0\|_1([0,\infty)) = \|H_0\| = \|H_t\|$, and continuity follows from the fact that

$$
\begin{aligned}
|H_{t+\delta}(\boldsymbol{x}) - H_t(\boldsymbol{x})| &= \left| \int_0^\infty (x_{t+\delta-s} - x_{t-s})^\top d\mu(s) \right| \\
&\leq \sum_i \int_0^\infty \|x_{t+\delta-s} - x_{t-s}\|_\infty d|\mu_i|(s),
\end{aligned}
\tag{32}
$$

which converges to 0 as $\delta \to 0$ by dominated convergence theorem. Finally, we will show that each $\mu_i$ is absolutely continuous with respect to $\lambda$ (Lebesgue measure). Take a measurable $E \subset [0,\infty)$ such that $\lambda(E) = 0$ and set $E' = [0,\infty) \setminus E$. For each $n \geq 0$ set $K_n \subset E, K'_n \subset E'$ where $K_n, K'_n$ are closed and $\mu_i(E \setminus K_n) \leq 1/n$, $\mu_i(E' \setminus K'_n) \leq 1/n$. For a fixed $i \in \{1, \ldots, d\}$, define $\boldsymbol{x}^{(n)}$ to be such that $x_{t-s,j}^{(n)} = 0$ for all $j \neq i$ and all $s$. For $j = i$, we set $x_{t-s,i}^{(n)} = 1$ if $s \in K_n$ and 0 if $s \in K'_n$, which can then be continuously extended to $[0,\infty)$. Observe that by construction, $x_{t-s}^{(n)} \to 0$ for $\lambda$-a.e. $s$, thus by dominated convergence theorem

$$
0 = \lim_{n\to\infty} H_t(\boldsymbol{x}^{(n)}) = \mu_i(E).
\tag{33}
$$

This shows that $\mu_i$ is absolutely continuous with respect to $\lambda$, and by the Radon-Nikodym theorem there exists a measurable function $\rho_i : [0,\infty) \to \mathbb{R}$ such that for any measurable $A \subset \mathbb{R}$ we have

$$
\int_A d\mu_i(s) = \int_A \rho_i(s)ds,
\tag{34}
$$

for $i = 1, \ldots, d$. Hence, we have

$$
H_t(\boldsymbol{x}) = \int_0^\infty x_{t-s}^\top \rho(s)ds
\tag{35}
$$

with $\|\rho\|_{L^1([0,\infty))} = \sum_i \int_0^\infty |\rho_i(s)|ds = \|\mu\|_1([0,\infty)) < \infty$. $\qquad\square$

**Lemma A.2.** *Let $\rho : [0,\infty) \to \mathbb{R}$ a Lebesgue integrable function, i.e. $\|\rho\|_{L^1([0,\infty))} < \infty$. Then, for any $\epsilon > 0$, there exists a polynomial $p$ with $p(0) = 0$ such that*

$$
\left\| \rho - p(e^{-\cdot}) \right\|_{L^1([0,\infty))} = \int_0^\infty |\rho(t) - p(e^{-t})|dt \leq \epsilon.
\tag{36}
$$

*Proof.* The approach here is similar to that of the approximation of functions using exponential sums (Kammler, 1976; Braess, 1986). Alternatively, one may also appeal to the density of phase type distributions (He & Zhang, 2007; O'Cinneide, 1990) in the space of positive distributions, and generalizing them to signed measures.

Fix $\epsilon > 0$. Define

$$
R(u) = \begin{cases} \frac{1}{u}\rho(-\log u), & u \in (0,1], \\ 0, & u = 0. \end{cases}
\tag{37}
$$

Then, we can check that

$$
\|R\|_{L^1([0,1])} = \|\rho\|_{L^1([0,\infty))} < \infty.
\tag{38}
$$

By density of continuous functions in $L^1$ there exists a continuous function $\tilde{R}$ on $[0,1]$ with $\tilde{R}(0) = 0$ such that

$$
\|R - \tilde{R}\|_{L^1([0,1])} \leq \epsilon/2.
\tag{39}
$$

By Müntz-Szász theorem (Müntz, 1914; Szász, 1916), there exists a polynomial $p$ with $p(0) = 0$ such that

$$
\|q - \tilde{R}\|_{L^1([0,1])} \leq \epsilon/2,
\tag{40}
$$

and $q(u) := p(u)/u$ is also a polynomial. Therefore, we have

$$
\begin{aligned}
\left\| \rho - p(e^{-\cdot}) \right\|_{L^1([0,\infty))} &= \int_0^1 |R(u) - p(u)/u|du \\
&\leq \int_0^1 |R(u) - \tilde{R}(u)|du + \int_0^1 |\tilde{R}(u) - p(u)/u|du \leq \epsilon.
\end{aligned}
\tag{41}
$$

$\qquad\square$

We are now ready to present the proof of Theorem 4.1.

*Proof of Theorem 4.1.* By (7), for each $\{\hat{H}_t\} \in \hat{\mathcal{H}}$ we can write

$$\hat{H}_t(\boldsymbol{x}) = \int_0^\infty x_{t-s}^\top (U^\top [e^{Ws}]^\top c) ds. \tag{42}$$

By Lemma A.1, we can write

$$H_t(\boldsymbol{x}) = \int_0^\infty x_{t-s}^\top \rho(s) ds, \tag{43}$$

where $\rho$ is integrable. Thus, we can apply Lemma A.2 to conclude that there exists polynomials $p_i, i = 1, \ldots, d$ with $p_i(0) = 0$ such that

$$\sum_i \|\rho_i - p_i(e^{-\cdot})\|_{L^1([0,\infty))} \le \epsilon. \tag{44}$$

Notice that we can write each $p_i(u) = \sum_{j=1}^m \alpha_{ij} u^j$ for some $m$ equaling the maximal order of $\{p_i\}$. Taking $W = \text{diag}(-1, \ldots, -m)$, $c = (1, \ldots, 1)$ and $U_{ij} = \alpha_{ji}$, we have

$$(U^\top [e^{Ws}]^\top c)_i = p_i(e^{-s}), \qquad i = 1, \ldots, d. \tag{45}$$

Consequently, we have for any $\boldsymbol{x}$ with $\|\boldsymbol{x}\|_\infty \le 1$,

$$
\begin{aligned}
|H_t(\boldsymbol{x}) - \hat{H}_t(\boldsymbol{x})| &= \left| \int_0^\infty x_{t-s}^\top \rho(s) ds - \int_0^\infty x_{t-s}^\top p(e^{-s}) ds \right| \\
&\le \sum_i \int_0^\infty |x_{t-s,i}| \left| \rho_i(s) - p_i(e^{-s}) \right| ds \le \sum_i \|\rho_i - p_i(e^{-\cdot})\|_{L^1([0,\infty))} \\
&\le \epsilon.
\end{aligned}
\tag{46}
$$

$\square$

# B   APPROXIMATION RATES AND THE CURSE OF MEMORY IN APPROXIMATION

We first give the proof of Theorem 4.2.

*Proof of Theorem 4.2.* We fix $i \in \{1, \ldots, d\}$ below until the last part of the proof. By Lemma A.1, there exists $\rho_i(t) \in C'^\alpha[0, \infty)$ such that

$$y_i(t) = H_t(\boldsymbol{e}_i) = \int_0^t \rho_i(r) dr, \quad t \ge 0. \tag{47}$$

By the assumption,

$$\rho_i^{(k)}(t) = o(e^{-\beta t}) \text{ as } t \to \infty, \quad k = 0, \ldots, \alpha. \tag{48}$$

Consider the transform

$$q_i(s) = \begin{cases} 0 & s = 0, \\ \dfrac{\rho_i\left(\frac{-(\alpha+1)\log s}{\beta}\right)}{s} & s \in (0, 1]. \end{cases} \tag{49}$$

For $k = 0, \ldots, \alpha$, one can prove by induction that

$$q_i^{(k)}(s) = (-1)^k \sum_{j=0}^k c(j, k) \left( \frac{\alpha + 1}{\beta} \right)^j \frac{\rho_i^{(j)}\left( \frac{-(\alpha+1)\log s}{\beta} \right)}{s^{k+1}}, \tag{50}$$

where $c(j,k)$ are some integer constants. Together with the assumption, we have

$$\left| q_i^{(k)}(e^{-\frac{\beta}{\alpha+1}t}) \right| = \left| \sum_{j=0}^{k} c(j,k) \left( \frac{\alpha+1}{\beta} \right)^j \frac{\rho_i^{(j)}(t)}{e^{-\frac{(k+1)\beta}{\alpha+1}t}} \right| \le \sum_{j=0}^{k} \left| c(j,k) \right| (\alpha+1)^j \left| \gamma \le C(\alpha)\gamma, \quad (51)$$

where $C(\alpha)$ is a universal constant only depending on $\alpha$. Note that for $j = 0, \ldots, \alpha$,

$$\lim_{s \to 0^+} \frac{\rho_i^{(j)} \left( \frac{-(\alpha+1)\log s}{\beta} \right)}{s^{k+1}} = \lim_{t \to \infty} \frac{\rho_i^{(j)}(t)}{e^{-\frac{(k+1)\beta}{\alpha+1}t}} = \lim_{t \to \infty} \frac{\rho_i^{(j)}(t)}{e^{-\beta t}} e^{-\frac{(\alpha-k)\beta}{\alpha+1}t} = 0, \quad (52)$$

hence $q_i(s) \in C^\alpha[0,1]$ with $q_i(0) = q_i^{(1)}(0) = \cdots = q_i^{(\alpha)}(0) = 0$. By Jackson's theorem (Jackson, 1930), for $m = 1, 2, \ldots$, there exists a polynomial $Q_{i,m}$ of degree $m-1$ such that

$$\|q_i - Q_{i,m}\|_{L^\infty([0,1])} \le \frac{C(\alpha)\gamma}{m^\alpha}. \quad (53)$$

Denote the polynomial $Q_{i,m}$ as

$$Q_{i,m}(s) = \sum_{j=0}^{m-1} \alpha_{i,j} s^j, \quad (54)$$

and define

$$\phi_{i,m}(t) = e^{-\frac{\beta}{\alpha+1}t} Q_{i,m}(e^{-\frac{\beta}{\alpha+1}t}). \quad (55)$$

Then we have

$$\phi_{i,m}(t) = c^\top e^{Wt} u_i, \quad (56)$$

where

$$c = (1, 1, \ldots, 1), \quad (57)$$

$$W = \begin{bmatrix} -\frac{\beta}{\alpha+1} & & & \\ & -\frac{2\beta}{\alpha+1} & & \\ & & \ddots & \\ & & & -\frac{m\beta}{\alpha+1} \end{bmatrix}, \quad (58)$$

$$u_i = (\alpha_{i,0}, \alpha_{i,1}, \ldots, \alpha_{i,m-1}). \quad (59)$$

With a change of variable $s = e^{-\frac{\beta}{\alpha+1}t}$, we have the estimate

$$\begin{aligned}
\|\rho_i - \phi_{i,m}\|_{L^1([0,\infty))} &= \int_0^\infty |\rho_i(t) - \phi_{i,m}(t)| dt \\
&= \int_0^1 \left| \rho_i \left( \frac{-(\alpha+1)\log s}{\beta} \right) - s Q_{i,m}(s) \right| \frac{\alpha+1}{\beta s} ds \\
&= \frac{\alpha+1}{\beta} \int_0^1 |q_i(s) - Q_{i,m}(s)| ds \\
&\le \frac{C(\alpha)\gamma}{\beta m^\alpha}.
\end{aligned} \quad (60)$$

Finally we define $U = [u_1, \ldots, u_d] \in \mathbb{R}^{m \times d}$ and have

$$c^\top e^{Wt} U = (\phi_{1,m}(t), \ldots, \phi_{d,m}(t)). \quad (61)$$

Recall the dynamical system (6)

$$\frac{d}{dt} h_t = \sigma(W h_t + U x_t), \qquad \hat{y}_t = c^\top h_t, \quad (62)$$

which is together determined by the parameters $c, W, U$. Similar to the argument in the proof of Theorem 4.1, for any $\boldsymbol{x}$ with $\|\boldsymbol{x}\|_\infty \le 1$ and $t$, we have

$$|H_t(\boldsymbol{x}) - \hat{H}_t(\boldsymbol{x})| \le \sum_i \|\rho_i - \phi_{i,m}\|_{L^1([0,\infty))} \le \frac{C(\alpha)\gamma d}{\beta m^\alpha}. \quad (63)$$

$\square$

**The curse of memory in approximation.** Now we explain more technical details of why Theorem 4.2 implies the curse of memory in approximation, as pointed out in the main text. We assume $d = 1$ and consider an example

$$H_t(\boldsymbol{x}) \coloneqq \int_0^t x_{t-s}\rho(s)ds, \ t \geq 0, \tag{64}$$

in which the density satisfies $\rho(t) \in C^{(1)}(\mathbb{R})$ and

$$\rho(t) \sim t^{-(1+\omega)} \text{ as } t \to +\infty. \tag{65}$$

Here $\omega > 0$ indicates the decay rate of the memory effects in our target functional family $\{H_t\}$. The smaller its value, the slower the decay and the longer the system memory. Notice that $y^{(1)}(t) = \rho(t)$, and in this case there exists no $\beta > 0$ making the following condition ((13) in the main text) true:

$$e^{\beta t}y_i^{(k)}(t) = o(1) \text{ as } t \to +\infty, \qquad \text{and} \qquad \sup_{t \geq 0} \beta^{-k}|e^{\beta t}y_i^{(k)}(t)| \leq \gamma, \tag{66}$$

and no rate estimate can be deduced from it.

A natural way to circumvent this obstacle is to introduce a truncation in time. With $T \ (\gg 1)$ we can define $\tilde{\rho}(t) \in C^{(1)}(\mathbb{R})$ such that $\tilde{\rho}(t) \equiv \rho(t)$ for $t \leq T$, $\tilde{\rho}(t) \equiv 0$ for $t \geq T + 1$, and $\tilde{\rho}(t)$ is monotonically decreasing for $T \leq t \leq T + 1$. Considering the linear functional $\tilde{H}_t(\boldsymbol{x}) \coloneqq \int_0^t x_{t-s}\tilde{\rho}(s)ds$, we have the truncation error estimate

$$|H_t(\boldsymbol{x}) - \tilde{H}_t(\boldsymbol{x})| \leq \|\boldsymbol{x}\|_{\mathcal{X}} \left( \int_T^\infty |\rho(s)|ds \right) \sim \|\boldsymbol{x}\|_{\mathcal{X}} T^{-\omega}. \tag{67}$$

Now the conclusion of Theorem 4.2 (i.e. (63)) is applicable to the truncated $\{\tilde{H}_t\}$ (with $\alpha = 1$), and we have for $\forall \beta > 0$, there is a linear RNN $(U, W, c)$ such that the associated functionals $\{\hat{H}_t\} \in \hat{\mathcal{H}}_m$ satisfy

$$\sup_{t \in \mathbb{R}} \|\tilde{H}_t - \hat{H}_t\| \leq \frac{C\gamma}{\beta m} \coloneqq \frac{C}{\beta m} \sup_{t \geq 0} \frac{|e^{\beta t}y^{(1)}(t)|}{\beta} = \frac{C\omega}{m} \frac{e^{\beta T}}{\beta^2 T^{\omega+1}}. \tag{68}$$

It is straightforward to verify that when $\beta = 2/T$, the right-hand side of (68) achieves the minimum, which gives us

$$\sup_{t \in \mathbb{R}} \|\tilde{H}_t - \hat{H}_t\| \leq \frac{C\omega}{m} T^{1-\omega}. \tag{69}$$

Combining (67) and (69) gives us

$$\sup_{t \in \mathbb{R}} \|H_t - \hat{H}_t\| \leq C \left( T^{-\omega} + \frac{\omega}{m} T^{1-\omega} \right). \tag{70}$$

In order to achieve an error tolerance $\epsilon$, according to the first term above we require $T \sim \epsilon^{-\frac{1}{\omega}}$, and then according to second term we have

$$m = \mathcal{O} \left( \frac{\omega T^{1-\omega}}{\epsilon} \right) = \mathcal{O} \left( \omega \epsilon^{-\frac{1}{\omega}} \right). \tag{71}$$

This estimate gives us a quantitative relationship between the degree of freedom needed and the decay speed. When $\omega$ is small, i.e., the system has long memory, the size of the RNN required grows exponentially.

## C  FINE-GRAINED ANALYSIS OF OPTIMIZATION DYNAMICS

### C.1  SIMPLIFICATIONS ON THE LOSS FUNCTION

Recall the loss function

$$\mathbb{E}_{\boldsymbol{x}} J_T(\boldsymbol{x}; c, W, U) = \mathbb{E}_{\boldsymbol{x}} \left| \int_0^T [c^\top e^{Wt}U - \rho(t)^\top]x_{T-t}dt \right|^2. \tag{72}$$

The simplifications on $\mathbb{E}_{\boldsymbol{x}} J_T$ are listed as follows.

1. Take the input data $x$ to be the white noise, so that

$$x_{T-t}dt \overset{\text{in distribution}}{=} dB_t, \tag{73}$$

where $B_t$ is the standard $d$-dimensional Wiener process. As a consequence, simplifying (72) via Itô's isometry gives

$$J_T(c, W, U) := \mathbb{E}_{\boldsymbol{x}} J_T(\boldsymbol{x}; c, W, U) = \int_0^T \left\| c^\top e^{Wt} U - \rho(t)^\top \right\|_2^2 dt. \tag{74}$$

2. We focus on the temporal dimension and take the spatial dimension $d = 1$ in (74).[1] Moreover, to investigate the effect of long-term memory, it is necessary to consider the training on large time horizons. Hence, we take $T \to \infty$ to get

$$J_\infty(c, W, b) := \int_0^\infty \left( c^\top e^{Wt} b - \rho(t) \right)^2 dt, \tag{75}$$

where $b$ is the sole column of $U$ in (74) and $\rho(t)$ becomes a scalar-valued target. This corresponds to the so-called single-input-single-output (SISO) system.

3. Due to the difficulty of directly analyzing $\nabla_W e^{Wt}$ and $\nabla_W^2 e^{Wt}$, we consider a further simplified ansatz. Assume that $W$ is a diagonal matrix with negative entries (to guarantee the stability of the model). That is, $W = -\text{diag}(w)$ with $w \in \mathbb{R}_+^m$. Then we can combine $a = b \circ c$ (where $\circ$ denotes the Hadamard product) and rewrite the model as

$$\hat{\rho}(t; c, W, b) := c^\top e^{Wt} b = \sum_{i=1}^m a_i e^{-w_i t} \triangleq a^\top e^{-wt} \triangleq \hat{\rho}(t; a, w). \tag{76}$$

The optimization problem (75) becomes

$$\min_{a \in \mathbb{R}^m, w \in \mathbb{R}_+^m} J(a, w) := \int_0^\infty \left( \sum_{i=1}^m a_i e^{-w_i t} - \rho(t) \right)^2 dt. \tag{77}$$

Here we omit the subscript $\infty$. [2]

4. We apply a continuous-time idealization of the gradient descent dynamics by considering the gradient flow with respect to $J(a, w)$. That is,

$$\begin{cases} a'(\tau) = -\nabla_a J(a(\tau), w(\tau)), \\ w'(\tau) = -\nabla_w J(a(\tau), w(\tau)), \end{cases} \tag{78}$$

with some initial value $a(0) = a_0 \in \mathbb{R}^m$, $w(0) = w_0 \in \mathbb{R}_+^m$.

As we will show later, applying the training dynamics (78) to optimize (77) is able to serve as a starting point in the fine-grained dynamical analysis, since it still preserves the plateauing behavior observed in the optimization process (see Figure 1), provided additional structures related to memories as discussed next.

## C.2 CONCRETE DYNAMICAL ANALYSIS AND THE CURSE OF MEMORY IN OPTIMIZATION

We prove Theorem 5.1 in this section. The basic insight is, by adding long-term memories in targets, one can increase the loss with little effect on the gradient and Hessian, which leads to a significant slow down of the training dynamics near the short-term memory parts of the targets. Therefore, Theorem 5.1 is proved subsequently in the following procedure.

1. We prove that $J_\omega$ has a large value but small gradient when $\hat{\rho}(t; a, w) \equiv \bar{\rho}(t)$.

---

[1]One can see that the spatial dimension plays little role in the previous approximation analysis (see the proof of Theorem 4.2), since each spatial dimension can be handled separately.

[2]The time horizon is always taken as $\infty$ in the whole analysis. Note that here we also omit an index $m$ (width of the network, relating to the model capacity), since it remains unchanged in the following content if not specified.

2. We prove that when $\hat{\rho}(t; a, w) \equiv \bar{\rho}(t)$, the Hessian $\nabla^2 J_\omega$ is positive semi-definite for $\omega = 0$, but for finite, small $\omega > 0$, $\nabla^2 J_\omega$ has $\mathcal{O}(1)$ positive eigenvalues and multiple $o(1)$ eigenvalues.

3. Based on these results, we perform a local linearization analysis on the gradient flow (21) initialized by $\hat{\rho}(t; a_0, w_0) \equiv \bar{\rho}(t)$, from which and by continuity the timescale of plateauing is derived.

### (1) Preliminary Results

As discussed around (20), we consider the target functional with a parametrized representation

$$\rho_\omega(t) = \bar{\rho}(t) + \rho_{0,\omega}(t) = \sum_{j=1}^{m^*} a_j^* e^{-w_j^* t} + \rho_0(t - 1/\omega).$$

Here, $a_j^* := a^*(w_j^*) \neq 0$, $w_j^* > 0$ and $w_i^* \neq w_j^*$ for any $i \neq j$, $i, j \in [m^*]$, [3] and $m^* < m$. The former requirements are just non-degenerate conditions, and the last requirement ensures that the model can perfectly represent the well-approximated part of the target, $\bar{\rho}(t)$. The memory in the target is controlled by $\rho_{0,\omega}(t) = \rho_0(t - 1/\omega)$, with $\rho_0$ as a fixed template function which satisfies the following assumptions.

*Assumptions on $\rho_0$.* (i) $\rho_0(t) \not\equiv 0$; (ii) $\rho_0 \in L^2(\mathbb{R}) \cap C^2(\mathbb{R})$; (iii) $\rho_0$ is bounded on $\mathbb{R}$, i.e. $\|\rho_0\|_{L^\infty(\mathbb{R})} < \infty$; (iv) $\lim_{t \to -\infty} \rho_0(t) = 0$.

**Remark C.1.** *The above assumptions (i)(ii)(iii) are rather natural, and (iv) only restricts the single side tail of $\rho_0$ to be zero. In the following analysis, we further focus on $\rho_0$ with light tails, e.g. the sub-Gaussian tail*

$$|\rho_0(t)| \leq c_0 e^{-c_1 t^2}, \quad \forall t : |t| \geq t_0 \tag{79}$$

*for some fixed positive constants $c_0, c_1, t_0$. Obviously, Gaussian densities and continuous functions with compact supports are sub-Gaussian functions.*

We begin by the following preliminary estimate that is used throughout the subsequent analysis.

**Lemma C.1.** *For any $n \in \mathbb{N}$, $\omega > 0$ and $w > 0$, let*

$$\Delta_{n,\omega}(w) := \int_0^\infty t^n e^{-wt} |\rho_{0,\omega}(t)| dt. \tag{80}$$

*Then*

- *$\Delta_{n,\omega}(w)$ is monotonically decreasing on $(0, \infty)$;*

- *$\lim_{\omega \to 0^+} \Delta_{n,\omega}(w) = 0$;*

- *In particular, if $\rho_0$ is sub-Gaussian, we further have*

$$\Delta_{n,\omega}(w) \lesssim \omega^{-n} e^{-w/\omega} \left( c_2^{w^2} + c_3^w \right), \quad \omega \in (0, \min\{1/2, 1/t_0, 2c_1/w\}). \tag{81}$$

*Here $c_2 = e^{\frac{1}{4c_1}} > 1$, $c_3 = e^{t_0} > 1$, and $\lesssim$ hides universal constants only depending on $n$ and $\rho_0, t_0, c_0, c_1$.*

*Proof.* (i) Obviously $\Delta_{n,\omega}(w_1) \leq \Delta_{n,\omega}(w_2)$ for any $w_1 > w_2 > 0$.

(ii) By the assumptions on $\rho_0$, we get

$$\lim_{\omega \to 0^+} |\rho_{0,\omega}(t)| = \lim_{\omega \to 0^+} |\rho_0(t - 1/\omega)| = \lim_{s \to -\infty} |\rho_0(s)| = 0, \quad \forall t \geq 0,$$

and $M_0 := \|\rho_0\|_{L^\infty(\mathbb{R})} < \infty$, which gives $t^n e^{-wt} |\rho_{0,\omega}(t)| \leq M_0 t^n e^{-wt} \in L^1([0, \infty))$ for any $n \in \mathbb{N}$, $\omega > 0$ and $w > 0$. By Lebesgue's dominant convergence theorem, we have

$$\lim_{\omega \to 0^+} \Delta_{n,\omega}(w) = \int_0^\infty t^n e^{-wt} \cdot \lim_{\omega \to 0^+} |\rho_{0,\omega}(t)| dt = 0, \quad \forall n \in \mathbb{N}, \forall w > 0. \tag{82}$$

---

[3] For any $n \in \mathbb{N}_+$, $[n] := \{1, 2, \cdots, n\}$.

(iii) Now we estimate $\Delta_{n,\omega}(w)$ under the sub-Gaussian condition (79). Suppose $0 < \omega < 1/t_0$, we have

$$
\begin{aligned}
|\Delta_{n,\omega}(w)| &= \int_0^\infty t^n e^{-wt} |\rho_0(t - 1/\omega)| dt \\
&= \int_{1/\omega - t_0}^{1/\omega + t_0} t^n e^{-wt} |\rho_0(t - 1/\omega)| dt + \int_0^{1/\omega - t_0} t^n e^{-wt} |\rho_0(t - 1/\omega)| dt \\
&\quad + \int_{1/\omega + t_0}^\infty t^n e^{-wt} |\rho_0(t - 1/\omega)| dt \triangleq I_1 + I_2 + I_3.
\end{aligned}
$$

Then we bound $I_1$, $I_2$ and $I_3$ respectively:

$$
\begin{aligned}
I_1 &\leq M_0 \int_{1/\omega - t_0}^{1/\omega + t_0} t^n e^{-wt} dt \leq M_0 e^{-w(1/\omega - t_0)} \int_{1/\omega - t_0}^{1/\omega + t_0} t^n dt \\
&= M_0 e^{wt_0} \cdot e^{-w/\omega} \cdot \frac{(1 + \omega t_0)^{n+1} - (1 - \omega t_0)^{n+1}}{(n+1)\omega^{n+1}} \\
&\lesssim M_0 e^{wt_0} \omega^{-n} e^{-w/\omega} (t_0 + \omega),
\end{aligned}
$$

where $\omega \in (0, 1/2)$, and $\lesssim$ hides universal constants only related to $n$ and $t_0$. Let $1/c_1 := 2\sigma^2$, we have

$$
I_2 = e^{-w/\omega} \int_{-1/\omega}^{-t_0} (s + 1/\omega)^n e^{-ws} |\rho_0(s)| ds \leq e^{-w/\omega} \int_{-1/\omega}^{-t_0} (s + 1/\omega)^n e^{-ws} \cdot c_0 e^{-c_1 s^2} ds,
$$

where

$$
\begin{aligned}
\int_{-1/\omega}^{-t_0} (s + 1/\omega)^n e^{-ws} e^{-c_1 s^2} ds &= e^{\frac{w^2}{4c_1}} \int_{-1/\omega}^{-t_0} (s + 1/\omega)^n e^{-c_1 (s + \frac{w}{2c_1})^2} ds \\
&\leq e^{\sigma^2 w^2/2} \int_{\mathbb{R}} (|t| + |1/\omega - \sigma^2 w|)^n \cdot e^{-\frac{t^2}{2\sigma^2}} dt \\
&= e^{\sigma^2 w^2/2} \sum_{k=0}^n C_n^k (1/\omega - \sigma^2 w)^{n-k} \cdot 2 \int_0^\infty t^k e^{-\frac{t^2}{2\sigma^2}} dt \\
&\leq e^{\sigma^2 w^2/2} \sum_{k=0}^n C_n^k (1/\omega)^{n-k} (\sqrt{2}\sigma)^{k+1} \Gamma\left(\frac{k+1}{2}\right)
\end{aligned}
$$

holds for any $\omega \in (0, 2c_1/w)$. Here the last inequality is due to the Mellin Transform of absolute moments of the Gaussian density (see Proposition 1 in Li et al. (2005)). The argument is similar for $I_3$, which gives the same bound as $I_2$.

Combining all the estimates gives the desired conclusion. The proof is completed. $\qquad\square$

The main idea to analyze plateauing behaviors is to investigate the local dynamics of the gradient flow (21) when $\hat{\rho} = \bar{\rho}$, then extend the results to the setting $\hat{\rho} \approx \bar{\rho}$ by continuity. Recall that both of them are exponential sums, we can obtain the relation of parameters between $\hat{\rho}$ and $\bar{\rho}$, according to the following lemma.

**Lemma C.2.** *For any $m \in \mathbb{N}_+$, let $\lambda = (\lambda_1, \cdots, \lambda_m)$ with $\lambda_i \neq \lambda_j$ for any $i \neq j$, $i, j \in [m]$. Then the series of functions $\{e^{\lambda_i t}\}_{i=1}^m$ is linear independent on any interval $I \subset \mathbb{R}$.*

*Proof.* The aim is to show

$$
\sum_{i=1}^m c_i e^{\lambda_i t} = 0, \ t \in I \Rightarrow c_i = 0, \ \forall i \in [m]. \tag{83}
$$

(83) holds trivially for $m = 1$. Assume that (83) holds for $m - 1$, then

$$
\sum_{i=1}^m c_i e^{\lambda_i t} = 0, \ t \in I \Rightarrow \sum_{i=1}^{m-1} c_i e^{(\lambda_i - \lambda_m)t} + c_m = 0, \ t \in I \tag{84}
$$

$$
\Rightarrow \sum_{i=1}^{m-1} c_i (\lambda_i - \lambda_m) e^{(\lambda_i - \lambda_m)t} = 0, \ t \in I.
$$

By induction, we get $c_i(\lambda_i - \lambda_m) = 0$ for any $i = 1, \cdots, m-1$. Since $\lambda_1, \cdots, \lambda_m$ are distinct, we have $c_i = 0$ for any $i = 1, \cdots, m-1$. Together with (84), we get $c_m = 0$, which completes the proof. $\square$

**Definition C.1.** *Let $m \geq m^*$. For any partition $\mathcal{P}$: $[m] = \cup_{j=0}^{m^*}\mathcal{I}_j$ with $\mathcal{I}_{j_1} \cap \mathcal{I}_{j_2} = \varnothing$ for any $j_1 \neq j_2$, $j_1, j_2 \in \{0\} \cup [m^*]$, and $\mathcal{I}_0 = \cup_{r=1}^{i_0}\mathcal{I}_{0,r}$ with $\mathcal{I}_{0,r_1} \cap \mathcal{I}_{0,r_2} = \varnothing$ for any $r_1 \neq r_2$, $r_1, r_2 \in [i_0]$, where $\mathcal{I}_j \neq \varnothing$ for any $j \in [m^*]$ and $\mathcal{I}_{0,r} \neq \varnothing$ for any $r \in [i_0]$ (if $\mathcal{I}_0 \neq \varnothing$), define the affine space (with respect to $\mathcal{P}$):*

$$\mathcal{M}_{\mathcal{P}}^* := \left\{ (a, w) \in \mathbb{R}^m \times \mathbb{R}_+^m : \sum_{i \in \mathcal{I}_j} a_i = a_j^*, \ w_i = w_j^* \text{ for any } i \in \mathcal{I}_j, \ j \in [m^*]; \right.$$

$$\left. \sum_{i \in \mathcal{I}_{0,r}} a_i = 0, \ w_i = v_r \neq w_j^* \text{ for any } i \in \mathcal{I}_{0,r}, \ r \in [i_0] \text{ and } j \in [m^*] \right\}.$$

*Denote the collection of all such affine spaces by $\mathcal{M}^* := \bigcup_{\mathcal{P}} \mathcal{M}_{\mathcal{P}}^*$.*

The following lemma characterizes the relation of parameters, by showing that $\mathcal{M}^*$ is exactly the set of equivalent points to $(a^*, w^*)$ for the purpose of representation via exponential sums.

**Lemma C.3.** *For any $(a, w) \in \mathbb{R}^m \times \mathbb{R}_+^m$, $\hat{\rho}(t; a, w) \equiv \bar{\rho}(t) \Leftrightarrow (a, w) \in \mathcal{M}^*$.*

*Proof.* (i) ($\Leftarrow$) Since $(a, w) \in \mathcal{M}^*$, there exists $\mathcal{P}$ such that $(a, w) \in \mathcal{M}_{\mathcal{P}}^*$. Then for any $t \geq 0$,

$$\hat{\rho}(t; a, w) = \sum_{i=1}^m a_i e^{-w_i t} = \sum_{j=0}^{m^*} \sum_{i \in \mathcal{I}_j} a_i e^{-w_i t} = \sum_{r=1}^{i_0} \sum_{i \in \mathcal{I}_{0,r}} a_i e^{-w_i t} + \sum_{j=1}^{m^*} \sum_{i \in \mathcal{I}_j} a_i e^{-w_i t}$$

$$= \sum_{r=1}^{i_0} \left( \sum_{i \in \mathcal{I}_{0,r}} a_i \right) e^{-v_r t} + \sum_{j=1}^{m^*} \left( \sum_{i \in \mathcal{I}_j} a_i \right) e^{-w_j^* t} = \sum_{j=1}^{m^*} a_j^* e^{-w_j^* t} = \bar{\rho}(t).$$

(ii) ($\Rightarrow$) Let $\mathcal{I}_j = \left\{ i \in [m] : w_i = w_j^* \right\}$ for any $j \in [m^*]$, and $\mathcal{I}_0 = \left\{ i \in [m] : w_i \neq w_j^* \text{ for any } j \in [m^*] \right\}$. Recall that $\bar{\rho}(t) = \sum_{j=1}^{m^*} a_j^* e^{-w_j^* t}$ is non-degenerate: $a_j^* \neq 0$, $w_j^* > 0$ and $w_i^* \neq w_j^*$ for any $i \neq j$, $i, j \in [m^*]$, we get $[m] = \cup_{j=0}^{m^*}\mathcal{I}_j$, $\mathcal{I}_{j_1} \cap \mathcal{I}_{j_2} = \varnothing$ for any $j_1 \neq j_2$, $j_1, j_2 \in \{0\} \cup [m^*]$. Combining Lemma C.2 and the non-degeneracy of $\bar{\rho}$, $\mathcal{I}_j \neq \varnothing$ for any $j \in [m^*]$. Assume that there are $i_0$ different components in $(w_i)_{i \in \mathcal{I}_0}$, say $v_1, \cdots, v_{i_0}$, then $v_r \neq w_j^*$ for any $r \in [i_0]$ and $j \in [m^*]$. Let $\mathcal{I}_{0,r} = \left\{ i \in \mathcal{I}_0 : w_i = v_r \right\}$ for any $r \in [i_0]$, we get $\mathcal{I}_{0,r} \neq \varnothing$ for any $r \in [i_0]$ (if $\mathcal{I}_0 \neq \varnothing$), and $\mathcal{I}_0 = \cup_{r=1}^{i_0}\mathcal{I}_{0,r}$, and $\mathcal{I}_{0,r_1} \cap \mathcal{I}_{0,r_2} = \varnothing$ for any $r_1 \neq r_2$, $r_1, r_2 \in [i_0]$. Hence $[m] = \cup_{j=0}^{m^*}\mathcal{I}_j$ with $\mathcal{I}_0 = \cup_{r=1}^{i_0}\mathcal{I}_{0,r}$ forms a $\mathcal{P}$ defined in Definition C.1, and

$$0 \equiv \hat{\rho}(t; a, w) - \bar{\rho}(t) = \sum_{i=1}^m a_i e^{-w_i t} - \sum_{j=1}^{m^*} a_j^* e^{-w_j^* t}$$

$$= \sum_{j=0}^{m^*} \sum_{i \in \mathcal{I}_j} a_i e^{-w_i t} - \sum_{j=1}^{m^*} a_j^* e^{-w_j^* t}$$

$$= \sum_{r=1}^{i_0} \sum_{i \in \mathcal{I}_{0,r}} a_i e^{-w_i t} + \left( \sum_{j=1}^{m^*} \sum_{i \in \mathcal{I}_j} a_i e^{-w_i t} - \sum_{j=1}^{m^*} a_j^* e^{-w_j^* t} \right)$$

$$= \sum_{r=1}^{i_0} \left( \sum_{i \in \mathcal{I}_{0,r}} a_i \right) e^{-v_r t} + \sum_{j=1}^{m^*} \left( \sum_{i \in \mathcal{I}_j} a_i - a_j^* \right) e^{-w_j^* t}.$$

Again by Lemma C.2, we have $\sum_{i \in \mathcal{I}_j} a_i = a_j^*$ for any $j \in [m^*]$ and $\sum_{i \in \mathcal{I}_{0,r}} a_i = 0$ for any $r \in [i_0]$, which gives $(a, w) \in \mathcal{M}_{\mathcal{P}}^*$. The proof is completed. $\square$

**Remark C.2.** *Let $\mathcal{I}_0 = \varnothing$.* [4] *It is straightforward to check that for any partition $\mathcal{P}$, the dimension of $\mathcal{M}_{\mathcal{P}}^*$ is $\sum_{j=1}^{m^*}(|\mathcal{I}_j| - 1) = m - m^*$. In addition, it can be verified that the cardinality of $\mathcal{M}^*$ is $m^*! \begin{Bmatrix} m \\ m^* \end{Bmatrix}$, where $\begin{Bmatrix} m \\ m^* \end{Bmatrix}$ is the Stirling number of the second kind.* [5]

### (2) Loss and Gradient

Now we show that as $\omega \to 0^+$, the loss $J_\omega$ remains lower bounded, while $\|\nabla J_\omega\|_2$ converges to 0. This implies that the loss will saturate at a high value when learning long-term memories.

**Proposition C.1.** *For any $(a, w) \in \mathbb{R}^m \times \mathbb{R}_+^m$ satisfying $\hat{\rho}(t; a, w) \equiv \bar{\rho}(t)$, we have*

$$J_\omega(a, w) \geq C(\rho_0) > 0, \quad \forall \omega \in (0, C'(\rho_0)), \tag{85}$$

*where $C(\rho_0), C'(\rho_0) > 0$ are constants only depending on $\rho_0$. That is, the loss is lower bounded uniformly in $\omega$.*

*Proof.* Recall the assumptions on $\rho_0$, we have $\rho_0(t) \not\equiv 0$ and $\rho_0 \in C(\mathbb{R})$. Let $t_1 \in \mathbb{R}$ such that $\rho_0(t_1) \neq 0$. By continuity, there exists $\delta_0 > 0$ such that $|\rho_0(t)| \geq |\rho_0(t_1)|/2$ for any $t \in [t_1 - \delta_0, t_1 + \delta_0]$. Hence, for any $\omega > 0$ sufficiently small such that $-1/\omega < t_1 - \delta_0$, we have

$$\|\rho_{0,\omega}\|_{L^2[0,\infty)}^2 = \int_0^\infty \rho_0^2(t - 1/\omega)dt = \int_{-\frac{1}{\omega}}^\infty \rho_0^2(t)dt \geq \int_{t_1-\delta_0}^{t_1+\delta_0} \rho_0^2(t)dt \geq \frac{1}{2}\delta_0|\rho_0(t_1)|^2 > 0.$$

Fix any $(\hat{a}, \hat{w}) \in \mathbb{R}^m \times \mathbb{R}_+^m$ such that $\hat{\rho}(t; \hat{a}, \hat{w}) \equiv \bar{\rho}(t)$. Then

$$J_\omega(\hat{a}, \hat{w}) = \|\hat{\rho}(t; \hat{a}, \hat{w}) - \bar{\rho}(t) - \rho_{0,\omega}(t)\|_{L^2[0,\infty)}^2 = \|\rho_{0,\omega}\|_{L^2[0,\infty)}^2 \geq \frac{1}{2}\delta_0|\rho_0(t_1)|^2 > 0,$$

which completes the proof. □

**Proposition C.2.** *For any $(a, w) \in \mathbb{R}^m \times \mathbb{R}_+^m$ satisfying $\hat{\rho}(t; a, w) \equiv \bar{\rho}(t)$, we have*

$$\lim_{\omega \to 0^+} \|\nabla J_\omega(a, w)\|_2 = 0. \tag{86}$$

*In particular, if $\rho_0$ has the sub-Gaussian tail (79), the estimate*

$$\|\nabla J_\omega(a, w)\|_2 \lesssim \sqrt{m}\omega^{-1}e^{-w_{min}/\omega}\left(c_2^{w_{min}^2} + c_3^{w_{min}}\right)(1 + \|a\|_\infty) \tag{87}$$

*holds for any $\omega \in (0, \min\{1/2, 1/t_0, 2c_1/w_{min}\})$. Here $w_{min} := \min_{i \in [m]} w_i > 0$, $c_2, c_3 > 1$ are constants only related to $c_1, t_0$, and $\lesssim$ hides universal constants only depending on $\rho_0$, $t_0$, $c_0$, $c_1$.*

*Proof.* A straightforward computation shows, for $k = 1, 2, \cdots, m$,

$$\frac{\partial J_\omega}{\partial a_k}(a, w) = 2\left[\sum_{i=1}^m \frac{a_i}{w_k + w_i} - \sum_{j=1}^{m^*} \frac{a_j^*}{w_k + w_j^*}\right] - 2\Delta_{0,\omega}(w_k), \tag{88}$$

$$\frac{\partial J_\omega}{\partial w_k}(a, w) = -2a_k\left[\sum_{i=1}^m \frac{a_i}{(w_k + w_i)^2} - \sum_{j=1}^{m^*} \frac{a_j^*}{(w_k + w_j^*)^2}\right] + 2a_k\Delta_{1,\omega}(w_k). \tag{89}$$

Fix any $(\hat{a}, \hat{w}) \in \mathbb{R}^m \times \mathbb{R}_+^m$ satisfying $\hat{\rho}(t; \hat{a}, \hat{w}) \equiv \bar{\rho}(t)$. By Lemma C.3, we have $(\hat{a}, \hat{w}) \in \mathcal{M}^*$. Recall Definition C.1, there exists a partition $\mathcal{P}$: $[m] = \cup_{j=0}^{m^*}\mathcal{I}_j$ with $\mathcal{I}_0 = \cup_{r=1}^{i_0}\mathcal{I}_{0,r}$, where $\mathcal{I}_j \neq \varnothing$ for any $j \in [m^*]$ and $\mathcal{I}_{0,r} \neq \varnothing$ for any $r \in [i_0]$ (if $\mathcal{I}_0 \neq \varnothing$), such that $(\hat{a}, \hat{w}) \in \mathcal{M}_{\mathcal{P}}^*$, which gives

---

[4]That is, the non-degenerate case. Obviously, $\mathcal{I}_0 \neq \varnothing$ implies an uncountable $\mathcal{M}_{\mathcal{P}}^*$, but they are all degenerate.

[5]The result follows from basic knowledge of combinatorics. See details in the proof of Theorem D.1.

that $\sum_{i \in \mathcal{I}_j} \hat{a}_i = a_j^*$, $\hat{w}_i = w_j^*$ for any $i \in \mathcal{I}_j$ and $j \in [m^*]$; $\sum_{i \in \mathcal{I}_{0,r}} \hat{a}_i = 0$, $\hat{w}_i = v_r \neq w_j^*$ for any $i \in \mathcal{I}_{0,r}$, $r \in [i_0]$ and $j \in [m^*]$. Therefore, for any $n \in \mathbb{N}_+$, we have

$$
\begin{aligned}
&\sum_{i=1}^{m} \frac{\hat{a}_i}{(\hat{w}_k + \hat{w}_i)^n} - \sum_{j=1}^{m^*} \frac{a_j^*}{(\hat{w}_k + w_j^*)^n} \\
&= \sum_{r=1}^{i_0} \sum_{i \in \mathcal{I}_{0,r}} \frac{\hat{a}_i}{(\hat{w}_k + \hat{w}_i)^n} + \sum_{j=1}^{m^*} \sum_{i \in \mathcal{I}_j} \frac{\hat{a}_i}{(\hat{w}_k + \hat{w}_i)^n} - \sum_{j=1}^{m^*} \frac{a_j^*}{(\hat{w}_k + w_j^*)^n} \\
&= \sum_{r=1}^{i_0} \frac{\sum_{i \in \mathcal{I}_{0,r}} \hat{a}_i}{(\hat{w}_k + v_r)^n} + \sum_{j=1}^{m^*} \frac{\sum_{i \in \mathcal{I}_j} \hat{a}_i}{(\hat{w}_k + w_j^*)^n} - \sum_{j=1}^{m^*} \frac{a_j^*}{(\hat{w}_k + w_j^*)^n} = 0. \quad (90)
\end{aligned}
$$

This yields

$$
\frac{\partial J_\omega}{\partial a_k}(\hat{a}, \hat{w}) = -2\Delta_{0,\omega}(\hat{w}_k), \qquad \frac{\partial J_\omega}{\partial w_k}(\hat{a}, \hat{w}) = 2\hat{a}_k \Delta_{1,\omega}(\hat{w}_k),
$$

and hence

$$
\|\nabla J_\omega(\hat{a}, \hat{w})\|_2^2 = 4 \sum_{k=1}^{m} \left[ \Delta_{0,\omega}^2(\hat{w}_k) + \hat{a}_k^2 \Delta_{1,\omega}^2(\hat{w}_k) \right].
$$

By Lemma C.1, we get $\lim_{\omega \to 0^+} \|\nabla J_\omega(\hat{a}, \hat{w})\|_2 = 0$.

If $\rho_0$ has the sub-Gaussian tail (79), again by Lemma C.1, the estimate

$$
\Delta_{n,\omega}(\hat{w}_k) \leq \Delta_{n,\omega}(\hat{w}_{\min}) \lesssim \omega^{-n} e^{-\hat{w}_{\min}^2/\omega} \left( c_2^{\hat{w}_{\min}^2} + c_3^{\hat{w}_{\min}} \right) \quad (91)
$$

holds for any $n \in \mathbb{N}$, $\omega \in (0, \min\{1/2, 1/t_0, 2c_1/\hat{w}_{\min}\})$ and $k \in [m]$. Here $\hat{w}_{\min} := \min_{i \in [m]} \hat{w}_i > 0$, $c_2, c_3 > 1$ are constants only related to $c_1, t_0$, and $\lesssim$ hides universal constants only depending on $n$ and $\rho_0, t_0, c_0, c_1$. Therefore

$$
\|\nabla J_\omega(\hat{a}, \hat{w})\|_2 \lesssim \sqrt{m}\omega^{-1} e^{-\hat{w}_{\min}^2/\omega} \left( c_2^{\hat{w}_{\min}^2} + c_3^{\hat{w}_{\min}} \right)(1 + \|\hat{a}\|_\infty), \quad \omega \in (0,1].
$$

The proof is completed. $\qquad\qquad\qquad\qquad\qquad\qquad\qquad\qquad\qquad\qquad\qquad\qquad\qquad\qquad\square$

### (3) Eigenvalues of Hessian

Now we show that minimal eigenvalues of $\nabla^2 J_\omega$ also converges to 0 as $\omega \to 0^+$.

**Proposition C.3.** *For any $(a, w) \in \mathbb{R}^m \times \mathbb{R}_+^m$ satisfying $\hat{\rho}(t; a, w) \equiv \bar{\rho}(t)$, denote the eigenvalues of $\nabla^2 J_\omega(a, w)$ by $\lambda_1(\omega) \geq \lambda_2(\omega) \geq \cdots \geq \lambda_{2m}(\omega)$. If $m > m^*$, we have*

$$
\lambda_k(\omega) > 0, \quad k = 1, 2, \cdots, m', \quad (92)
$$

$$
\lim_{\omega \to 0^+} \lambda_k(\omega) = 0, \quad k = m'+1, m'+2, \cdots, 2m \quad (93)
$$

*for $\omega > 0$ sufficiently small, where $m' \leq 2m^* + |\mathcal{I}_0| \leq m + m^*$. In particular, if $\rho_0$ has the sub-Gaussian tail (79), the estimate*

$$
|\lambda_k(\omega)| \lesssim \omega^{-2} e^{-w_{min}/\omega} \left( c_2^{w_{min}^2} + c_3^{w_{min}} \right)(1 + \|a\|_\infty) \quad k = m'+1, m'+2, \cdots, 2m \quad (94)
$$

*holds for any $\omega \in (0, \min\{1/2, 1/t_0, 2c_1/w_{min}\})$. Here $w_{min} := \min_{i \in [m]} w_i > 0$, $c_2, c_3 > 1$ are constants only related to $c_1, t_0$, and $\lesssim$ hides universal constants only depending on $\rho_0, t_0, c_0, c_1$.*

*Proof.* A straightforward computation shows, for $k, j = 1, 2, \cdots, m$,

$$\frac{\partial^2 J_\omega}{\partial a_k \partial a_j}(a, w) = \frac{2}{w_k + w_j}, \tag{95}$$

$$\frac{\partial^2 J_\omega}{\partial a_k \partial w_j}(a, w) = \frac{-2a_j}{(w_k + w_j)^2}, \quad k \neq j, \tag{96}$$

$$\frac{\partial^2 J_\omega}{\partial a_k \partial w_k}(a, w) = -2 \left[ \sum_{i=1}^m \frac{a_i}{(w_k + w_i)^2} - \sum_{j'=1}^{m^*} \frac{a_{j'}^*}{(w_k + w_{j'}^*)^2} \right] - \frac{a_k}{2w_k^2} + 2\Delta_{1,\omega}(w_k), \tag{97}$$

$$\frac{\partial^2 J_\omega}{\partial w_k \partial w_j}(a, w) = \frac{4a_k a_j}{(w_k + w_j)^3}, \quad k \neq j, \tag{98}$$

$$\frac{\partial^2 J_\omega}{\partial w_k \partial w_k}(a, w) = 4a_k \left[ \sum_{i=1}^m \frac{a_i}{(w_k + w_i)^3} - \sum_{j'=1}^{m^*} \frac{a_{j'}^*}{(w_k + w_{j'}^*)^3} \right] + \frac{a_k^2}{2w_k^3} - 2a_k \Delta_{2,\omega}(w_k). \tag{99}$$

Fix any $(\hat{a}, \hat{w}) \in \mathbb{R}^m \times \mathbb{R}_+^m$ satisfying $\hat{\rho}(t; \hat{a}, \hat{w}) \equiv \bar{\rho}(t)$. By (90), we have

$$\frac{\partial^2 J_\omega}{\partial a_k \partial a_j}(\hat{a}, \hat{w}) = \frac{2}{\hat{w}_k + \hat{w}_j},$$

$$\frac{\partial^2 J_\omega}{\partial a_k \partial w_j}(\hat{a}, \hat{w}) = \frac{-2\hat{a}_j}{(\hat{w}_k + \hat{w}_j)^2} \quad (k \neq j), \qquad \frac{\partial^2 J_\omega}{\partial a_k \partial w_k}(\hat{a}, \hat{w}) = -\frac{\hat{a}_k}{2\hat{w}_k^2} + 2\Delta_{1,\omega}(\hat{w}_k),$$

$$\frac{\partial^2 J_\omega}{\partial w_k \partial w_j}(\hat{a}, \hat{w}) = \frac{4\hat{a}_k \hat{a}_j}{(\hat{w}_k + \hat{w}_j)^3} \quad (k \neq j), \qquad \frac{\partial^2 J_\omega}{\partial w_k \partial w_k}(\hat{a}, \hat{w}) = \frac{\hat{a}_k^2}{2\hat{w}_k^3} - 2\hat{a}_k \Delta_{2,\omega}(\hat{w}_k).$$

Let

$$\bar{J}(a, w) := \|\hat{\rho}(t; a, w) - \bar{\rho}(t)\|_{L^2[0,\infty)}^2 \tag{100}$$

$$= \left\| \sum_{i=1}^m a_i e^{-w_i t} - \sum_{j=1}^{m^*} a_j^* e^{-w_j^* t} \right\|_{L^2[0,\infty)}^2,$$

and

$$\mathcal{E}_\omega(a, w) := \begin{bmatrix} O_{m \times m} & \mathrm{Diag}(\Delta_{1,\omega}(w)) \\ \mathrm{Diag}(\Delta_{1,\omega}(w)) & -\mathrm{Diag}(a \circ \Delta_{2,\omega}(w)) \end{bmatrix},$$

where $\Delta_{n,\omega}(w)$ $(n = 1, 2)$ is performed element-wisely. One can verify that

$$\nabla^2 J_\omega(a, w) = \nabla^2 \bar{J}(a, w) + 2\mathcal{E}_\omega(a, w). \tag{101}$$

Then we analyze $\nabla^2 \bar{J}(\hat{a}, \hat{w})$ and $\mathcal{E}_\omega(\hat{a}, \hat{w})$ respectively.

(i) $\nabla^2 \bar{J}(\hat{a}, \hat{w})$. Obviously $(\hat{a}, \hat{w})$ is a global minimizer of $\bar{J}(a, w)$ due to $\bar{J}(\hat{a}, \hat{w}) = 0$. Hence $\nabla \bar{J}(\hat{a}, \hat{w}) = 0$ and $\nabla^2 \bar{J}(\hat{a}, \hat{w})$ is positive semi-definite. We further show that $\nabla^2 \bar{J}(\hat{a}, \hat{w})$ has multiple zero eigenvalues when $m > m^*$. In fact, since

$$\frac{\partial^2 \bar{J}}{\partial a_k \partial a_j}(\hat{a}, \hat{w}) = \frac{2}{\hat{w}_k + \hat{w}_j}, \quad \frac{\partial^2 \bar{J}}{\partial a_k \partial w_j}(\hat{a}, \hat{w}) = \frac{-2\hat{a}_j}{(\hat{w}_k + \hat{w}_j)^2}, \quad \frac{\partial^2 \bar{J}}{\partial w_k \partial w_j}(\hat{a}, \hat{w}) = \frac{4\hat{a}_k \hat{a}_j}{(\hat{w}_k + \hat{w}_j)^3},$$

it is straightforward to verify that for any $i, j \in \mathcal{I}_p, p \in [m^*]$ and any $i, j \in \mathcal{I}_{0,r}, r \in [i_0]$,

$$\nabla^2 \bar{J}(\hat{a}, \hat{w})^{i,:} = \nabla^2 \bar{J}(\hat{a}, \hat{w})^{j,:}, \ \hat{a}_j \cdot \nabla^2 \bar{J}(\hat{a}, \hat{w})^{m+i,:} = \hat{a}_i \cdot \nabla^2 \bar{J}(\hat{a}, \hat{w})^{m+j,:},$$

where $A^{i,:}$ denotes the $i$-th row of matrix $A$. Notice that $\sum_{i \in \mathcal{I}_{0,r}} \nabla^2 \bar{J}(\hat{a}, \hat{w})^{m+i,:} = 0$ for any $r \in [i_0]$, we conclude that the Hessian $\nabla^2 \bar{J}(\hat{a}, \hat{w})$ has at most $2m^* + i_0 + i_2 \leq 2m^* + |\mathcal{I}_0| \leq m + m^*$ different rows, [6] which yields $\mathrm{rank}(\nabla^2 \bar{J}(\hat{a}, \hat{w})) \leq 2m^* + |\mathcal{I}_0| \leq m + m^*$. Therefore,

---

[6]Here $i_2 := |\{r \in [i_0] : |\mathcal{I}_{0,r}| \geq 2\}|$. When $\mathcal{I}_0 = \varnothing$, the upper bound is $2m^*$; when $\mathcal{I}_0 \neq \varnothing$, since $\mathcal{I}_{0,r} \neq \varnothing$ for any $r \in [i_0]$, let $i_1 := |\{r \in [i_0] : |\mathcal{I}_{0,r}| = 1\}|$ and $i_2$ defined as before. Then $i_0 = i_1 + i_2$, $|\mathcal{I}_0| = \sum_{r=1}^{i_0} |\mathcal{I}_{0,r}| \geq i_1 + 2i_2 = i_0 + i_2$. The last inequality follows from $|\mathcal{I}_0| = m - \sum_{j=1}^{m^*} |\mathcal{I}_j| \leq m - m^*$.

the number of zero eigenvalues of $\nabla^2 \bar{J}(\hat{a}, \hat{w}) \geq \dim \left\{ x \in \mathbb{R}^{2m} : \nabla^2 \bar{J}(\hat{a}, \hat{w}) \cdot x = 0 \right\} = 2m - \text{rank}(\nabla^2 \bar{J}(\hat{a}, \hat{w})) \geq 2(m - m^*) - |\mathcal{I}_0| \geq m - m^*$. Since $\nabla^2 \bar{J}(\hat{a}, \hat{w})$ is positive semi-definite, all the non-zero eigenvalues must be positive.

(ii) $\mathcal{E}_\omega(\hat{a}, \hat{w})$. Let

$$
\begin{aligned}
G_k^{(1)} &:= \{ y \in \mathbb{R} : |y| \leq |\Delta_{1,\omega}(\hat{w}_k)| \}, \\
G_k^{(2)} &:= \{ y \in \mathbb{R} : |y + \hat{a}_k \Delta_{2,\omega}(\hat{w}_k)| \leq |\Delta_{1,\omega}(\hat{w}_k)| \}.
\end{aligned}
$$

By Gershgorin's circle theorem, for any eigenvalue of $\mathcal{E}_\omega(\hat{a}, \hat{w})$, say $\lambda(\omega)$, we have $\lambda(\omega) \in \bigcup_{k=1}^m (G_k^{(1)} \cup G_k^{(2)})$. Combining with Lemma C.1, we get

$$
|\lambda(\omega)| \leq \max_{k \in [m]} \left( |\hat{a}_k| |\Delta_{2,\omega}(\hat{w}_k)| + |\Delta_{1,\omega}(\hat{w}_k)| \right) \to 0, \quad \omega \to 0^+. \tag{102}
$$

If $\rho_0$ has the sub-Gaussian tail (79), by (91), we further have

$$
\begin{aligned}
|\lambda(\omega)| &\leq \max_{k \in [m]} \left( |\hat{a}_k| |\Delta_{2,\omega}(\hat{w}_k)| + |\Delta_{1,\omega}(\hat{w}_k)| \right) \\
&\lesssim \omega^{-2} e^{-\hat{w}_{\min}/\omega} \left( c_2^{\hat{w}_{\min}^2} + c_3^{\hat{w}_{\min}} \right) (1 + \|\hat{a}\|_\infty), \quad \omega \in (0, 1], \tag{103}
\end{aligned}
$$

where $\omega \in (0, \min\{1/2, 1/t_0, 2c_1/\hat{w}_{\min}\})$, $\hat{w}_{\min} := \min_{i \in [m]} \hat{w}_i > 0$, $c_2, c_3 > 1$ are constants only related to $c_1, t_0$, and $\lesssim$ hides universal constants only depending on $\rho_0, t_0, c_0, c_1$.

Combining (i), (ii) and applying Weyl's theorem gives the desired result. $\square$

### (4) Local Linearization Analysis

The previous analysis can now be tied directly to a quantitative dynamics via linearization arguments. It is shown that under mild assumptions, the gradient flow (21) can become trapped in plateaus with an *exponentially* large timescale. That is, *the curse of memory* occurs, this time in optimization dynamics instead of approximation rates.

**Theorem C.1** (Restatement of Theorem 5.1)**.** *For any $\omega > 0$, $m \in \mathbb{N}_+$ and $\theta_0 = (a_0, w_0) \in \mathbb{R}^m \times \mathbb{R}_+^m$, $0 < \delta \ll 1$, define the hitting time*

$$
\tau_0 = \tau_0(\delta; \omega, m, \theta_0) := \inf \{ \tau \geq 0 : \|\theta_\omega(\tau) - \theta_0\|_2 > \delta \}, \tag{104}
$$

$$
\tau_0' = \tau_0'(\delta; \omega, m, \theta_0) := \inf \{ \tau \geq 0 : |J_\omega(\theta_\omega(\tau)) - J_\omega(\theta_0)| > \delta \}. \tag{105}
$$

*Assume that $m > m^*$, and the initialization satisfies $\hat{\rho}(t; \theta_0) \approx \bar{\rho}(t)$. Then we have*

$$
\lim_{\omega \to 0^+} \tau_0(\delta; \omega, m, \theta_0) = \lim_{\omega \to 0^+} \tau_0'(\delta; \omega, m, \theta_0) = +\infty. \tag{106}
$$

*In particular, if $\rho_0$ has the sub-Gaussian tail (79), and the initialization is bounded as $(a_0, w_0) \in [a_l^0, a_r^0]^m \times [w_l^0, w_r^0]^m$ with constants $a_l^0 < a_r^0$, $0 < w_l^0 < w_r^0$, we further have*

$$
\tau_0'(\delta; \omega, m, \theta_0) \geq \tau_0(\delta; \omega, m, \theta_0) \gtrsim \omega^2 e^{w_l^0/\omega} \min \left\{ \frac{\delta}{\sqrt{m}}, \ln(1 + \delta) \right\} \tag{107}
$$

*for any $\omega \in (0, \min\{1/2, 1/t_0, 2c_1/w_r^0\})$ sufficiently small, where $\gtrsim$ hides universal constants only depending on $\rho_0, t_0, c_0, c_1, w_r^0, a_l^0$ and $a_r^0$.*

*Proof.* Consider the asymptotic expansion with the form

$$
\theta_\omega(\tau) = \theta_\omega^0(\tau) + \sum_{i=1}^\infty \delta^i \theta_\omega^i(\tau) = \theta_\omega^0(\tau) + \delta \theta_\omega^1(\tau) + \delta^2 \theta_\omega^2(\tau) + o(\delta^2), \tag{108}
$$

for some $\delta \in (0, 1)$ (with $\delta \ll 1$) and $\theta_\omega^i(\tau) = \mathcal{O}(1)$ ($\tau \geq 0$, $i = 0, 1, \cdots$). [7] For consistency, we have $\theta_\omega^0(0) = \theta_0$ and $\theta_\omega^i(0) = 0$ for $i = 1, 2, \cdots$. By continuity, $\tau_0 > 0$ and $\|\theta_\omega(\tau) - \theta_0\|_2 \leq \delta$ for any $\tau \in [0, \tau_0]$. The aim is to quantify the scale of $\tau_0$.

---

[7]Here $\theta_\omega^i(\tau)$ denotes the $i$-th term in the asymptotic expansion of $\theta_\omega(\tau)$, not the $i$-th power.

Let $g_0 := \nabla J_\omega(\theta_0)$ and $H_0 := \nabla^2 J_\omega(\theta_0)$. The local linearization on (21) shows

$$\frac{d}{d\tau}\theta_\omega(\tau) = -g_0 - H_0(\theta_\omega(\tau) - \theta_0) + \mathcal{O}(\delta^2), \quad \tau \in [0, \tau_0].$$

Combining with (108), we have

$$\frac{d}{d\tau}\theta_\omega^0(\tau) = -g_0 - H_0(\theta_\omega^0(\tau) - \theta_0), \quad \theta_\omega^0(0) = \theta_0, \qquad \text{at } \mathcal{O}(1) \text{ scale,}$$

$$\frac{d}{d\tau}\theta_\omega^1(\tau) = -H_0\theta_\omega^1(\tau), \quad \theta_\omega^1(0) = 0, \qquad \text{at } \mathcal{O}(\delta) \text{ scale,}$$

$$\frac{d}{d\tau}\theta_\omega^2(\tau) = -H_0\theta_\omega^2(\tau) + \mathcal{O}(1), \quad \theta_\omega^2(0) = 0, \qquad \text{at } \mathcal{O}(\delta^2) \text{ scale.}$$

Therefore

$$\theta_\omega^0(\tau) = \theta_0 - \left(\int_0^\tau e^{-H_0 s} ds\right) g_0,$$

$$\theta_\omega^1(\tau) = e^{-H_0\tau}\theta_\omega^1(0) = 0,$$

which gives

$$\theta_\omega(\tau) = \theta_0 - \left(\int_0^\tau e^{-H_0 s} ds\right) g_0 + \mathcal{O}(\delta^2), \quad \tau \in [0, \tau_0]. \tag{109}$$

To achieve a parameter separation gap $\delta_0$, i.e. $\|\theta_\omega(\tau) - \theta_0\|_2 = \delta_0$ with $\delta_0 = c\delta, c \in (0, 1]$, we need to take $\tau$ such that

$$\left\|\left(\int_0^\tau e^{-H_0 s} ds\right) g_0\right\|_2 \geq \frac{\delta_0}{2}. \tag{110}$$

Let $H_0 = P^\top \Lambda P$ be the eigenvalue decomposition with $P$ orthogonal and $\Lambda = \text{diag}(\lambda_1, \cdots \lambda_{2m})$ consisting of the eigenvalues of $H_0$ with $\lambda_1 \geq \cdots \geq \lambda_{2m}$. Then

$$\left\|\left(\int_0^\tau e^{-H_0 s} ds\right) g_0\right\|_2 = \left\|P^\top \left(\int_0^\tau e^{-\Lambda s} ds\right) P g_0\right\|_2 \leq \left\|\int_0^\tau e^{-\Lambda s} ds\right\|_2 \|g_0\|_2$$

$$\leq \|g_0\|_2 \cdot \max\left\{\max_{i \in [2m], \lambda_i \neq 0} \frac{1}{|\lambda_i|}|e^{-\lambda_i \tau} - 1|, \ \tau\right\}.$$

It is straightforward to verify that $h(\tau; \lambda) := \frac{1}{|\lambda|}|e^{-\lambda\tau} - 1|, \ \tau \geq 0$ monotonically decreases on $\lambda \in \mathbb{R}$ for any $\tau \geq 0$. [8] Hence

$$\left\|\left(\int_0^\tau e^{-H_0 s} ds\right) g_0\right\|_2 \leq \|g_0\|_2 \cdot \begin{cases} \frac{1}{-\lambda_{2m}}(e^{-\lambda_{2m}\tau} - 1), & \lambda_{2m} < 0, \\ \tau, & \lambda_{2m} \geq 0, \end{cases} \tag{111}$$

and the right hand side monotonically increases on $\tau \geq 0$. Combining (110), (111) gives

$$\frac{\delta_0}{2} \leq \|g_0\|_2 \cdot \begin{cases} \frac{1}{-\lambda_{2m}}(e^{-\lambda_{2m}\tau} - 1), & \lambda_{2m} < 0, \\ \tau, & \lambda_{2m} \geq 0. \end{cases} \tag{112}$$

We discuss for different cases:

(i) $\|g_0\|_2 = 0$. Obviously the inequality (112) fails since (110) fails for any $\tau \geq 0$, which gives $\tau_0 = +\infty$;

(ii) $\|g_0\|_2 \neq 0$ and $\lambda_{2m} \geq 0$. By (112), we get $\tau \geq \frac{\delta_0}{2\|g_0\|_2}$;

(iii) $\|g_0\|_2 \neq 0$ and $\lambda_{2m} < 0$. By (112), we get

$$\tau \geq \frac{1}{-\lambda_{2m}} \ln\left(1 + \delta_0 \frac{-\lambda_{2m}}{2\|g_0\|_2}\right).$$

---

[8] With the convention that $h(\tau; 0) = \tau$ for any $\tau > 0$, and $h(0; \lambda) \equiv 0$ for any $\lambda \in \mathbb{R}$.

If $-\lambda_{2m} \le 2\|g_0\|_2$, we have

$$\tau \ge \frac{1}{-\lambda_{2m}} \cdot \delta_0 \frac{-\lambda_{2m}}{2\|g_0\|_2} \cdot \frac{\ln\left(1 + \delta_0 \frac{-\lambda_{2m}}{2\|g_0\|_2}\right)}{\delta_0 \frac{-\lambda_{2m}}{2\|g_0\|_2}}$$

$$= \frac{\delta_0}{2\|g_0\|_2}\left(1 + \mathcal{O}\left(\delta_0 \frac{-\lambda_{2m}}{2\|g_0\|_2}\right)\right) = \frac{\delta_0}{2\|g_0\|_2}(1 + \mathcal{O}(\delta_0));$$

if $-\lambda_{2m} > 2\|g_0\|_2$, we have $\tau \ge \frac{\ln(1+\delta_0)}{-\lambda_{2m}}$.

Combining (i), (ii), (iii) gives

$$\tau_0 = \tau_0(\delta; \omega, m, \theta_0) \gtrsim \min\left\{\frac{\delta}{\|g_0\|_2}, \frac{\ln(1+\delta)}{|\lambda_{2m}|}\right\}. \tag{113}$$

Let the initialization satisfy $\hat{\rho}(t; \theta_0) \equiv \bar{\rho}(t)$, and assume $m > m^*$. According to Propositions C.2 and C.3, we have

$$\lim_{\omega\to0^+} \|g_0\|_2 = 0, \quad \lim_{\omega\to0^+} \lambda_{2m} = 0 \Rightarrow \lim_{\omega\to0^+} \tau_0(\delta; \omega, m, \theta_0) = +\infty. \tag{114}$$

If $\rho_0$ has the sub-Gaussian tail (79), again by Propositions C.2 and C.3, we further have

$$\tau_0(\delta; \omega, m, \theta_0) \gtrsim \omega^2 e^{w_{0,\min}/\omega} \min\left\{\frac{\delta}{\sqrt{m}}, \ln(1+\delta)\right\} \frac{1}{\left(c_2^{w_{0,\min}^2} + c_3^{w_{0,\min}}\right)(1 + \|a_0\|_\infty)} \tag{115}$$

for any $\omega \in (0, \min\{1/2, 1/t_0, 2c_1/w_{0,\min}\})$, where $w_{0,\min} := \min_{i\in[m]} w_{0,i} > 0$, $c_2, c_3 > 1$ are constants only related to $c_1, t_0$, and $\gtrsim$ hides universal constants only depending on $\rho_0, t_0, c_0, c_1$. Since the initialization is bounded as $(a_0, w_0) \in [a_l^0, a_r^0]^m \times [w_l^0, w_r^0]^m$ with $a_l^0 < a_r^0, 0 < w_l^0 < w_r^0$, let $c_a^0 = \max\{|a_l^0|, |a_r^0|\}$, we get

$$\tau_0(\delta; \omega, m, \theta_0) \gtrsim \omega^2 e^{w_l^0/\omega} \min\left\{\frac{\delta}{\sqrt{m}}, \ln(1+\delta)\right\} \frac{1}{\left(c_2^{(w_r^0)^2} + c_3^{w_r^0}\right)(1 + c_a^0)}$$

$$\gtrsim \omega^2 e^{w_l^0/\omega} \min\left\{\frac{\delta}{\sqrt{m}}, \ln(1+\delta)\right\}, \tag{116}$$

where $\gtrsim$ hides universal constants only related to $w_r^0, a_l^0$ and $a_r^0$.

The last task is to show the dynamics of the loss is much slower than the parameter separation when there is plateauing. The argument is trivial since for any $\tau \in [0, \tau_0]$,

$$J_\omega(\theta_\omega(\tau)) - J_\omega(\theta_0) = g_0^\top(\theta_\omega(\tau) - \theta_0) + (\theta_\omega(\tau) - \theta_0)^\top H_0(\theta_\omega(\tau) - \theta_0) + o(\delta^2)$$

$$\ge -\|g_0\|_2\|\theta_\omega(\tau) - \theta_0\|_2 + \lambda_{2m}\|\theta_\omega(\tau) - \theta_0\|_2^2 + o(\delta^2)$$

$$= o(1)\mathcal{O}(\delta) + o(1)\mathcal{O}(\delta^2) + o(\delta^2)$$

$$= o(\delta^2), \quad \omega \to 0^+.$$

By continuity, the proof is completed. $\qquad\square$

**Remark C.3.** *The estimate in Theorem C.1 shows a lower bound on the escape time, hence it does not appear to preclude the situation that the plateauing lasts forever. However, in the proof above, if one supposes $\tau_0 = +\infty$ in (104), i.e. the hypothetical situation where the parameters are trapped forever, and write $\tilde{g}_0 := Pg_0 = (\tilde{g}_{0,1}, \cdots, \tilde{g}_{0,2m})$, we have*

$$\left\|\left(\int_0^\tau e^{-H_0 s} ds\right) g_0\right\|_2^2 = \tilde{g}_0^\top\left(\int_0^\tau e^{-\Lambda s} ds\right)^2 \tilde{g}_0 = \sum_{i=1}^{2m}(\tilde{g}_{0,i})^2(h(\tau; \lambda_i))^2 \ge (\tilde{g}_{0,j})^2(h(\tau; \lambda_j))^2$$

*for any $j$ such that $\lambda_j < 0$. If $\tilde{g}_{0,j} \ne 0$, (109) gives*

$$\|\theta_\omega(\tau) - \theta_0\|_2 \ge \left\|\left(\int_0^\tau e^{-H_0 s} ds\right) g_0\right\|_2 + \mathcal{O}(\delta^2)$$

$$\ge \frac{|\tilde{g}_{0,j}|}{-\lambda_j}(e^{-\lambda_j \tau} - 1) + \mathcal{O}(\delta^2) \to +\infty, \quad \tau \to \infty,$$

*which is a contradiction. That is to say, the parameter separation has to achieve the gap $\delta$ within a finite time, even if it is exponentially large.*

**Remark C.4.** *Recall Lemma C.3, $\hat{\rho}(t; a_0, w_0) \equiv \bar{\rho}(t)$ if and only if $(a_0, w_0) \in \mathcal{M}^* = \bigcup_{\mathcal{P}} \mathcal{M}^*_{\mathcal{P}}$, where $\mathcal{P}$ is a partition over $[m]$ as defined in Definition C.1. That is, as a union of affine spaces, $\mathcal{M}^*$ is in fact an equivalent set for qualified initializations. As discussed in Remark C.2, when there is no degeneracy, the cardinality of $\mathcal{M}^*$ is $m^*! \begin{Bmatrix} m \\ m^* \end{Bmatrix}$ (i.e. the number of $\mathcal{P}$), with each $\mathcal{M}^*_{\mathcal{P}}$ an $(m - m^*)$-dimensional affine space; when there is degeneracy in some $\mathcal{M}^*_{\mathcal{P}}$, it then becoms an uncountable set. Certainly, initializations sufficiently near $\mathcal{M}^*$ are also qualified by continuity.*

**Remark C.5.** *Motivated by the idea of weights degeneracy (see Definition C.1), we can further apply similar methods to a (global) landscape analysis on the loss function $J_\omega$. The results there show that the plateaus are all over the landscape, even provided general targets (without memory structures). See details in Appendix D.*

## C.3 NUMERICAL EXPERIMENTS

### C.3.1 MOTIVATING TESTS

In this section we give details of Figure 1.

**(1) Learning linear functionals using linear RNNs (with GD optimizer)**

The target functional is $H_T(\boldsymbol{x}) = \int_0^T \rho(t) x_{T-t} dt$ with white noise $\boldsymbol{x}$, while the representation $\rho$ is selected as the exponential sum or the scaled Airy function:

1. Exponential sum: $\rho(t) = [c^*]^\top e^{W^* t} b^*$, where $c^*, b^*$ are standard normal random vectors of $m^*$ dimensions, and $W = -I - Z^\top Z$ with $Z \in \mathbb{R}^{m^* \times m^*}$ is a Gaussian random matrix with i.i.d. entries having variance $1/m^*$.

2. Airy function: $\rho(t) = \mathrm{Ai}(s_0[t - t_0])$, where $\mathrm{Ai}(t)$ is the Airy function of the first kind, given by the improper integral

$$\mathrm{Ai}(t) = \frac{1}{\pi} \lim_{\xi \to \infty} \int_0^\xi \cos\left(\frac{u^3}{3} + tu\right) du. \tag{117}$$

Note that in the first example, the memory of target functional decays quickly. However, for the second example, the effective rate of decay is controlled by the parameter $t_0$. For $t \leq t_0$, the Airy function is oscillatory. Hence for large $t_0$, a large amount of memory is present in the target. In Figure 1 we set $m^* = 8$ for exponential sums, and $t_0 = 3$, $s_0 = 2.25$ for Airy functions.

In Figure 1 (a) and (b), we plot the gradient descent dynamics on training the linear RNNs (discretized using Euler method, hence equivalent to residual RNNs). We observe an efficient training process for the exponential sum case, while "plateauing" behaviors in the Airy function case. This causes a severe slow down of training. In addition, we also find that the plateauing effect gets worse as $t_0$ (or $s_0$) is increased, which corresponds to more complex Airy functions in the sense of more memory effects. That is, the long-term memory adversely affects the optimization process via gradient descent.

**(2) Learning nonlinear functionals using nonlinear RNNs (with Adam optimizer)**

To show that the plateauing behavior may be generic, we also consider a nonlinear forced dynamical system target, the Lorenz 96 system (Lorenz, 1996):

$$\begin{aligned} \dot{y} &= -y + x + \sum_{k=1}^{K} z_k / K, \\ \dot{z}_k &= 2[z_{k+1}(z_{k-1} - z_{k+2}) - z_k + y], \quad k = 1, \cdots, K, \end{aligned} \tag{118}$$

with cyclic indices $z_{k+K} = z_k$, and $x$ is an external stochastic noise. When the unresolved variables $z_k$ are unknown, the dynamics of the resolved variable $y$ driven by $x$ is a nonlinear dynamical system with memory effects (but not a linear functional). We use a standard nonlinear RNN (with the $\tanh$ activation) to learn the sequence-to-sequence mapping $\boldsymbol{x}_{0:T} \mapsto \boldsymbol{y}_{0:T}$ with the Adam optimizer. Figure 1 (c) shows that the training of the Lorenz 96 system with the presence of memory also exhibits the plateauing phenomenon.

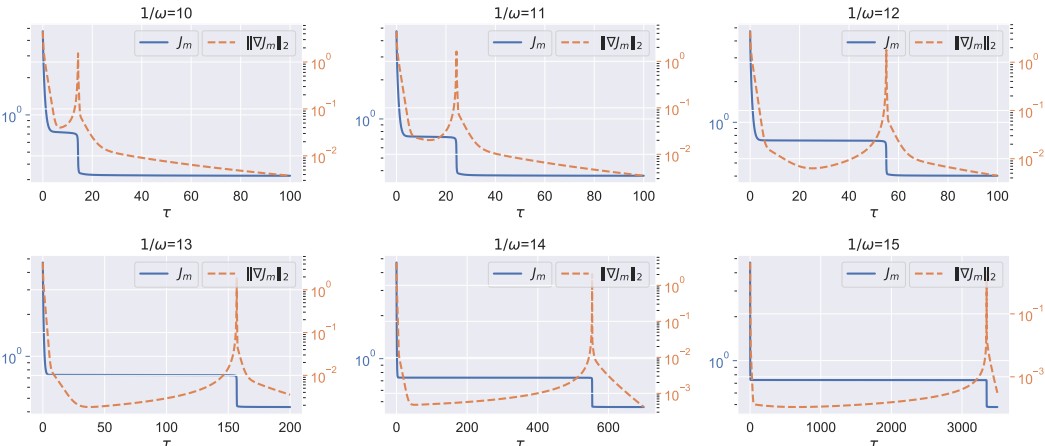

Figure 2: The training dynamics of the target functional defined by (119) using the model (120). Here we take the width $m = 2$ in $\hat{\rho}$. The corresponding gradient flow (21) is numerically solved by the Adams-Bashforth-Moulton method. Observe that the plateauing time increases rapidly as the memory becomes longer ($\omega$ decreases).

In all cases, the trained RNN model has a hidden dimension of 16 and the total length of the path is $T = 6.4$. The continuous-time RNN is discretized using the Euler method with step size $0.1$.

### C.3.2  Long-term Memory Significantly Contributes to Plateaus

Recall the simple example of target with memory

$$\rho(t) = a^* e^{-w^* t} + c_0 e^{-\frac{(t - 1/\omega)^2}{2\sigma^2}}, \qquad w^* > 0. \tag{119}$$

We aim to learn $\rho$ with the exponential sum

$$\hat{\rho}_m(t; a, w) = \sum_{i=1}^{m} a_i e^{-w_i t}, \qquad a \in \mathbb{R}^m, \quad w \in \mathbb{R}_+^m. \tag{120}$$

That is, the simplified linear RNN model with a diagonal recurrent kernel (see (76) and Appendix C.1). The optimization is performed by the gradient flow (21), i.e. a continuous-time idealization of the gradient descent dynamics. The ODE (21) is numerically solved by the Adams-Bashforth-Moulton method. The results are illustrated in Figure 2. It is shown that the plateauing time increases rapidly as the memory $1/\omega$ becomes longer.

### C.3.3  Numerical Verifications

#### (1) The timescale estimate

We first numerically verify the timescale proved in Theorem 5.1 (or Theorem C.1). That is, the time of plateauing (and also parameter separation $\|\theta(\tau) - \theta(0)\|_2$) is exponentially large as the memory $1/\omega \to +\infty$. The results are shown in Figure 3, where we observe good agreement with the predicted scaling.

#### (2) General cases

To facilitate mathematical tractability, the analysis so far is done on the restrictive cases of the diagonal recurrent kernel $W$ with negative entries, linear activations and the gradient flow training dynamics. However, we show here that the plateauing behavior - which we now understood as a generic feature of long-term memory of the target functional and its interaction with the optimization dynamics - is present even for general cases, and hence our simplified analytical setting is representative of the general situation.

In Figure 4, we still take the target functional as defined in (119), but apply more general models to learn it, including using full (non-diagonal) recurrent kernels of the RNN with no restrictions on

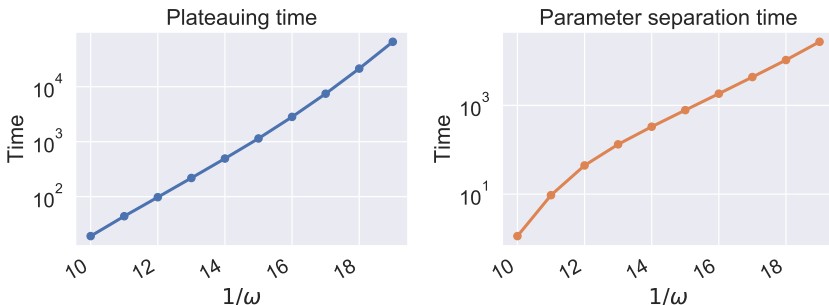

Figure 3: The timescale of plateauing and parameter separation. Here the model and target are both selected the same as Figure 2, but with a larger width $m = 10$. We see the logarithm of time of plateauing and parameter separation is almost linear to the memory $1/\omega$.

entries, using non-linear ($\texttt{tanh}$) activations and using the Adam optimizer. Furthermore, we do not take the Itô isometry simplification and instead use actual input sample paths of finite time horizons, just as one would do in training RNNs in practice. We observe that the plateauing behavior is present in all cases. Moreover, in the last case of Adam (which can be thought of as a momentum-based optimization method), the plateauing behavior is somewhat alleviated, although the separation of timescales is still present. This is consistent with our supplemental analysis in Appendix E, where we show that the momentum-based methods will speed up training based on our dynamical (Appendix C.2) and landscape analysis (Appendix D) of plateauing.

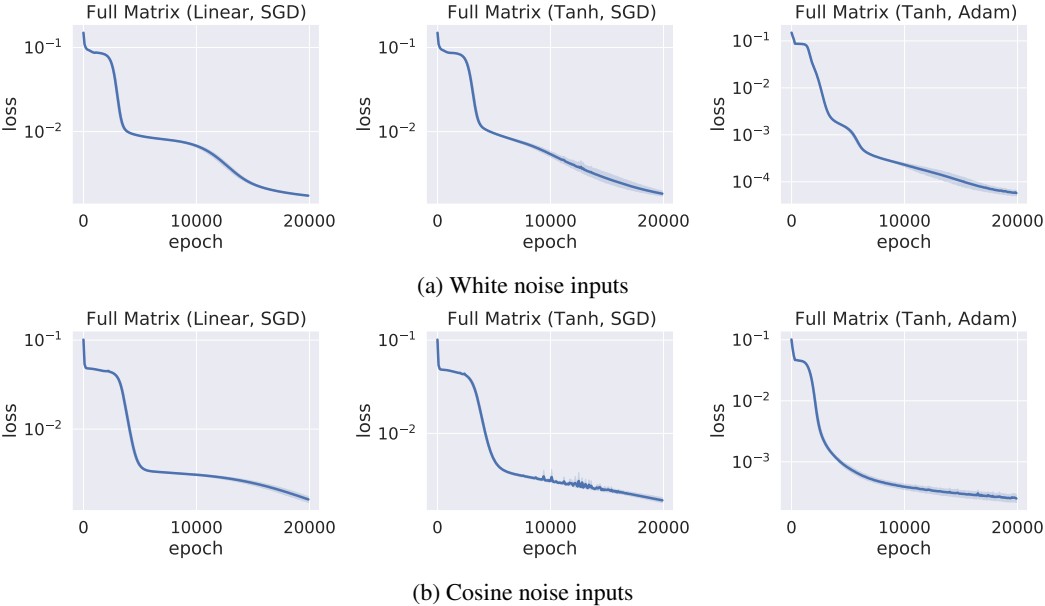

Figure 4: Numerical verifications of the plateauing behavior under general settings, with non-diagonal recurrent kernels, the non-linear activation ($\texttt{tanh}$), and the Adam (momentum-based) optimizer. Here we use the target functional the same as Figure 2 with the memory $1/\omega = 20$. The time horizon is chosen as $T = 32$, and 128 input samples are generated from a standard white noise. The learning rate is 1.0 for GD and 0.001 for Adam. 10 initializations are sampled and trained for each experiment. We consider two possible input distributions: a) white noise inputs; b) inputs of the form $x_t = \sum_{j=1}^{J} \alpha_j \cos(\lambda_j t)$, where $\lambda_j \sim U[0, 10]$ and $\alpha_j \sim \mathcal{N}(0, 1)$. We observe that plateaus occur in all cases and the momentum generally improves the situation but still not resolve the difficulty.

# D  LANDSCAPE ANALYSIS

As mentioned in Remark C.5, we can perform a global landscape analysis on the loss function based on the idea of weights degeneracy, which arises from Definition C.1. Recall that the loss function reads

$$\min_{a\in\mathbb{R}^m, w\in\mathbb{R}^m_+} J_m(a,w) := \int_0^\infty \left( \sum_{i=1}^m a_i e^{-w_i t} - \rho(t) \right)^2 dt. \tag{121}$$

The main results of the appendix are summarized as follows.

- In Theorem D.1, we prove that the loss function has infinitely many critical points, which form a factorial number of affine spaces;
- In Theorem D.2, we prove that such (critical) affine spaces are much more than global minimizers provided the target being an exponential sum; [9]
- In Theorem D.3, we prove that on such (critical) affine spaces, the Hessian is singular in the sense of processing multiple zero eigenvalues;
- In Proposition D.1, we prove that the (critical) affine spaces contain both saddles and degenerate stable points which are not global optimal.

Instead of a local dynamical analysis in Appendix C.2, we generalize similar methods to a *global landscape analysis* here, and the results hold for the loss function associated with *general targets*. More specifically, these results complement our main results (see Theorem 5.1 or Theorem C.1) in the following aspects.

- It is shown that the weights degeneracy is quite common in the whole landscape of the loss function. Unfortunately, the weights degeneracy often worsens the landscape to a large extent;
- It is shown that the weights degeneracy leads to a large number of stable areas (i.e. critical affine spaces), but most of them contribute to non-global minimizers;
- It is shown that these stable areas can also be quite flat, which often connect with local plateaus;
- For the structure of these stable areas, there are both saddles and degenerate critical points (not global optimal). In certain regimes, even saddles can be rather difficult to escape from (see Theorem 5.1 or Theorem C.1).

As a consequence, the optimization problem of linear RNNs is globally and essentially difficult to solve.

## D.1  SYMMETRY ANALYSIS ON THE LANDSCAPE

This subsection consists of two parts: in Appendix D.1.1, we give main results provided the existence of weights degeneracy; in Appendix D.1.2, we give sufficient conditions to guarantee the existence. Since the key observation to utilize weights degeneracy is to notice the permutation symmetry of coordinates of gradients, we also called it "symmetry analysis".

### D.1.1  GENERIC THEORY

We begin with the following definition, which describes the concept of weights degeneracy in a natural and rigorous way.

**Definition D.1.** *(coincided critical solutions and affine spaces) Let $d \in \mathbb{N}_+$ and $1 \leq d \leq m$. We say $(a, w)$ is a d-coincided critical solution of $J_m$, if $\nabla J_m(a, w) = 0$, and $w = (w_i) \in \mathbb{R}^m_+$ has d different components. The coincided critical affine spaces are defined as coincided critical solutions that form affine spaces.*

---

[9]The global minimizers are distinct when the target is an exponential sum. Here we compare the number of (critical) affine spaces with the number of global minimizers (both of them are finite). When the target is not an exponential sum, the same conclusion holds if there are still finite number of global minimizers. See Remark D.3 in Appendix D.1.1 for details.

To guarantee the existence of such solutions, it is necessary to have the following definition.

**Definition D.2.** $(\hat{a}, \hat{w}) \in \mathbb{R}^m \otimes \mathbb{R}^m_+$ *is called the non-degenerate global minimizer of $J_m$, if and only if*

$$J_m(\hat{a}, \hat{w}) = \inf_{a \in \mathbb{R}^m, w \in \mathbb{R}^m_+} J_m(a, w), \tag{122}$$

*and $(\hat{a}, \hat{w})$ takes a non-degenerate form*

$$\hat{a}_i \neq 0, \ \hat{w}_i \neq \hat{w}_j \text{ for } i \neq j, \quad i, j = 1, 2, \cdots, m. \tag{123}$$

For convenience, we also define an index set

$$\mathcal{N} := \{n \in \mathbb{N}_+ : \ J_m \text{ has non} - \text{degenerate global minimizers for any } m \leq n\}, \tag{124}$$

which is used frequently in the following analysis. For any $f \in L^2[0, \infty)$, let $\mathcal{L}[f]$ be the Laplace transform of $f$, i.e. $\mathcal{L}[f](s) = \int_0^\infty e^{-st} f(t) dt, \ s > 0$. We begin with the following lemma.

**Lemma D.1.** *Assume that $\rho$ is smooth and $\sqrt{w} \, |\mathcal{L}[\rho](w)| \to 0$ as $w \to 0^+$ and $w \to \infty$. Then we have $1 \in \mathcal{N}$ and thus $\mathcal{N} \neq \varnothing$.*

*Proof.* We aim to show that there exists $\hat{a} \neq 0$ and $\hat{w} > 0$, such that

$$J_1(\hat{a}, \hat{w}) = \inf_{a \in \mathbb{R}, w \in \mathbb{R}_+} J_1(a, w). \tag{125}$$

The basic idea is to limit the unbounded domain $a \in \mathbb{R}, w \in \mathbb{R}_+$ to a compact set without effecting the minimization of $J_1(a, w)$. We have

$$
\begin{aligned}
\min_{a, w > 0} J_1(a, w) &= \min_{w > 0} \min_a \ \left\{ \frac{1}{2w} \cdot a^2 - 2\mathcal{L}[\rho](w) \cdot a + \|\rho\|^2_{L^2[0,\infty)} \right\} \\
&= \min_{w > 0} \min_a \ \left\{ \frac{1}{2w} \big(a - 2w\mathcal{L}[\rho](w)\big)^2 + \left[ \|\rho\|^2_{L^2[0,\infty)} - 2w(\mathcal{L}[\rho](w))^2 \right] \right\} \\
&= \min_{w > 0} \ \left\{ \|\rho\|^2_{L^2[0,\infty)} - 2w(\mathcal{L}[\rho](w))^2 \right\} = J_1(a(w), w),
\end{aligned}
$$

where $a(w) := 2w\mathcal{L}[\rho](w)$. Write $h(w) := J_1(a(w), w)$, then $h(0^+) = h(\infty) = \|\rho\|^2_{L^2[0,\infty)}$. Obviously $h(w) < \|\rho\|^2_{L^2[0,\infty)}$ for any $w > 0$, hence

$$\min_{w > 0} h(w) = \min_{w \in [w_{lb}, w_{ub}]} h(w), \quad 0 < w_{lb} < w_{ub} < \infty,$$

which implies

$$\min_{a, w > 0} J_1(a, w) = \min_{w > 0} J_1(a(w), w) = \min_{w \in [w_{lb}, w_{ub}]} J_1(a(w), w).$$

That is to say, the minimization of $J_1(a, w)$ can be equivalently performed on a 2-dimensional smooth curve
$(w, a(w))_{w \in [w_{lb}, w_{ub}]}$, which is certainly a compact set. By continuity, $J_1(a, w)$ has global minimizers, say $(\hat{a}, \hat{w})$. Obviously $\hat{w} > 0$ and $\hat{a} = a(\hat{w}) \neq 0$ (since $\hat{a} = 0$ implies $J_1(\hat{a}, w) = \|\rho\|^2_{L^2[0,\infty)}$, certainly not a minimum), which completes the proof. $\square$

**Remark D.1.** *If the target is an exponential sum, i.e. $\rho(t) = \sum_{j=1}^{m^*} a_j^* e^{-w_j^* t}$, we know $\rho$ is smooth and $\sqrt{w} \, |\mathcal{L}[\rho](w)| \to 0$ as $w \to 0^+$ and $w \to +\infty$; hence $1 \in \mathcal{N}$ by Lemma D.1, and thus $\mathcal{N} \neq \varnothing$. In fact, $\mathcal{L}[\rho](w) = \sum_{j=1}^{m^*} \frac{a_j^*}{w + w_j^*}$ implies that $\mathcal{L}[\rho](w) = O(1)$ when $w \to 0^+$, and $\mathcal{L}[\rho](w) = O(1/w)$ when $w \to +\infty$.*

**Theorem D.1.** *Assume that $\mathcal{N} \neq \varnothing$ with $\mathcal{N}$ defined as (124). Let $M := \sup \mathcal{N}$. Then for any $m, d \in \mathbb{N}_+, 1 \leq d \leq \min\{m, M\}$, there exists at least $d! \begin{Bmatrix} m \\ d \end{Bmatrix}$ d-coincided critical affine spaces of $J_m$, [10] where $\begin{Bmatrix} m \\ d \end{Bmatrix} \in \mathbb{N}_+$ is called the Stirling number of the second kind.*

---

[10] The affine spaces degenerate to distinct points when $d = m$. For sufficient conditions to guarantee $M > 1$ (to avoid vacuous results), see Theorem D.6 and Remark D.7 in Appendix D.1.2.

*Proof.* (i) Existence. The key observation is the permutation symmetry of $\nabla J_m$: by (19), if $a_i = a_j$ and $w_i = w_j$ for some $i \neq j$, then $\frac{\partial J_m}{\partial a_i} = \frac{\partial J_m}{\partial a_j}$ and $\frac{\partial J_m}{\partial w_i} = \frac{\partial J_m}{\partial w_j}$.

For any $m, d \in \mathbb{N}_+$, $1 \leq d \leq m$, suppose that $w = (w_i) \in \mathbb{R}_+^m$ has $d$ different components. Then for any partition $\mathcal{P}$: $\{1, \cdots, m\} = \cup_{j=1}^d \mathcal{I}_j$ with $\mathcal{I}_{j_1} \cap \mathcal{I}_{j_2} = \varnothing$ for any $j_1 \neq j_2$, $j_1, j_2 = 1, \cdots, d$, define the affine space

$$\mathcal{M}_{\mathcal{P},(b,v),(m,d)} := \left\{ (a,w) \in \mathbb{R}^m \otimes \mathbb{R}_+^m : w_i = v_j \text{ for any } i \in \mathcal{I}_j, \sum_{i \in \mathcal{I}_j} a_i = b_j, \quad j = 1, \cdots, d \right\}$$

for some $(b, v) \in \mathbb{R}^d \otimes \mathbb{R}_+^d$, where $v$ has exactly $d$ different components. Therefore, for any $(a, w) \in \mathcal{M}_{\mathcal{P},(b,v),(m,d)}$, we have

$$J_m(a,w) = \left\| \sum_{j=1}^d \sum_{i \in \mathcal{I}_j} a_i e^{-w_i t} - \rho(t) \right\|_{L^2[0,\infty)}^2 = \left\| \sum_{j=1}^d b_j e^{-v_j t} - \rho(t) \right\|_{L^2[0,\infty)}^2 = J_d(b,v),$$

and similarly

$$\frac{\partial J_m}{\partial a_k}(a,w) = 2 \int_0^\infty e^{-v_s t} \left( \sum_{j=1}^d b_j e^{-v_j t} - \rho(t) \right) dt, \quad k \in \mathcal{I}_s, \ s = 1, 2, \cdots, d,$$

$$\frac{\partial J_m}{\partial w_k}(a,w) = 2a_k \int_0^\infty (-t) e^{-v_s t} \left( \sum_{j=1}^d b_j e^{-v_j t} - \rho(t) \right) dt, \quad k \in \mathcal{I}_s, \ s = 1, 2, \cdots, d.$$

Notice that

$$\frac{\partial J_d}{\partial b_s}(b,v) = 2 \int_0^\infty e^{-v_s t} \left( \sum_{j=1}^d b_j e^{-v_j t} - \rho(t) \right) dt, \quad s = 1, 2, \cdots, d,$$

$$\frac{\partial J_d}{\partial v_s}(b,v) = 2b_s \int_0^\infty (-t) e^{-v_s t} \left( \sum_{j=1}^d b_j e^{-v_j t} - \rho(t) \right) dt, \quad s = 1, 2, \cdots, d,$$

we have

$$\frac{\partial J_m}{\partial a_k}(a,w) = \frac{\partial J_d}{\partial b_s}(b,v), \quad b_s \frac{\partial J_m}{\partial w_k}(a,w) = a_k \frac{\partial J_d}{\partial v_s}(b,v), \quad k \in \mathcal{I}_s, \ s = 1, 2, \cdots, d. \quad (126)$$

[11] Since $d \leq \min\{m, M\}$, $d \in \mathcal{N}$. In fact, for any $k \in \mathbb{N}_+$, if $k \notin \mathcal{N}$, there exists $i \leq k$ such that $J_{(i)}$ has no non-degenerate global minimizers, we have $j \notin \mathcal{N}$ for any $j \geq i$, hence $M \leq i - 1 \leq k - 1$. Hence $M = \infty$ implies $\mathcal{N} = \mathbb{N}_+$ and $M < \infty$ implies $M \in \mathcal{N}$, and both of them lead to $d \in \mathcal{N}$. Therefore, $J_d$ has non-degenerate global minimizers, i.e. there exists $(\hat{b}, \hat{v}) \in \mathbb{R}^d \otimes \mathbb{R}_+^d$ such that

$$J_d(\hat{b}, \hat{v}) = \inf_{b \in \mathbb{R}^d, v \in \mathbb{R}_+^d} J_d(b, v), \quad (127)$$

and $(\hat{b}, \hat{v})$ takes a non-degenerate form

$$\hat{b}_i \neq 0, \ \hat{v}_i \neq \hat{v}_j \text{ for any } i \neq j, \quad i, j = 1, 2, \cdots, d. \quad (128)$$

By (127), we get $\nabla J_d(\hat{b}, \hat{v}) = 0$. Combining with (126) and (128), we obtain $\nabla J_m(\hat{a}, \hat{w}) = 0$ for any $(\hat{a}, \hat{w}) \in \mathcal{M}_{\mathcal{P},(\hat{b},\hat{v}),(m,d)}$, i.e. $(\hat{a}, \hat{w})$ belongs to a $d$-coincided critical affine space. Note that the affine space is with the dimension $\sum_{j=1}^d (|\mathcal{I}_j| - 1) = m - d$, since there are $d$ linear equality constrains on the $m$-dimensional vector $a$.

---

[11] By considering the gradient flow dynamic of $J_d$ instead of $J_m$, a model reduction (from $m$-dimensional to $d$-dimensional) is almost completed on $\mathcal{M}_{\mathcal{P},(b,v),(m,d)}$, except for some trivial degenerate cases (e.g. $a_k = 0$ or $b_s = 0$).

(ii) Counting. By the structure of affine spaces discussed above, we can identify different affine spaces with respect to the partition $\mathcal{P}$. For counting the number of different partitions $\mathcal{P}$: $\{1, \cdots, m\} = \cup_{j=1}^{d} \mathcal{I}_j$, it can be decomposed into the following two steps. First, partitioning a set of $m$ labelled objects into $d$ nonempty unlabelled subsets. By definition, the answer is the Stirling number of the second kind $\begin{Bmatrix} m \\ d \end{Bmatrix}$. Second, assign each partition to $\mathcal{I}_1, \cdots, \mathcal{I}_d$ accordingly. There are $d!$ ways in total. Therefore, the number of $d$-coincided critical affine spaces is at least $d! \begin{Bmatrix} m \\ d \end{Bmatrix}$. The proof is completed. $\square$

Combining Lemma D.1, Remark D.1 and Theorem D.1 gives the following theorem, which states that there are much more saddles and degenerate stable points which are not global optimal than global minimizers in the landscape (provided the target being an exponential sum).

**Theorem D.2.** *Fix any $m \in \mathbb{N}_+$ relatively large. Consider the loss $J_m$ associated with the target being a non-degenerate exponential sum, i.e. $\rho(t) = \sum_{j=1}^{m} a_j^* e^{-w_j^* t}$, where $a_j^* \neq 0$ and $w_i^* \neq w_j^*$ for any $i \neq j$, $i, j = 1, \cdots, m$. Assume that $m \in \mathcal{N}$ [12] with $\mathcal{N}$ defined in (124). Then in the landscape of $J_m$, the number of coincided critical affine spaces is at least $Poly(m)$ times larger than the number of global minimizers.*

*Proof.* (i) Global minimizers. Since the target is an exponential sum, we have $J_m(a, w) \geq 0$ and $J_m(\bar{a}^*, \bar{w}^*) = 0$, where $\bar{a}^* = Pa^*$ and $\bar{w}^* = Pw^*$ with $P \in \mathbb{R}^{m \times m}$ to be some permutation matrix. Next we show $J_m$ has no other global minimizers.

Suppose $J_m(a, w) = 0$, we have

$$\sum_{i=1}^{m} a_i e^{-w_i t} - \sum_{j=1}^{m} a_j^* e^{-w_j^* t} = 0, \quad t \geq 0. \tag{129}$$

It is easy to see that for any $j = 1, \cdots, m$, there exists $i(j)$ such that $w_{i(j)} = w_j^*$. Otherwise, if $w_i \neq w_j^*$, $i = 1, \cdots, m$, by (83) or Lemma C.2, we have $a_j^* = 0$, which is a contradiction. Notice that $w_i^* \neq w_j^*$ for any $i \neq j$, different $w_j^*$'s will correspond to different $w_i$'s, hence the correspondence is one-to-one. Therefore, let $w_i = w_{j(i)}^*$, (129) can be rewritten as

$$0 = \sum_{i=1}^{m} a_i e^{-w_{j(i)}^* t} - \sum_{i=1}^{m} a_{j(i)}^* e^{-w_{j(i)}^* t} = \sum_{i=1}^{m} (a_i - a_{j(i)}^*) e^{-w_{j(i)}^* t}, \quad t \geq 0.$$

Again by Lemma C.2, we have $a_i = a_{j(i)}^*$. That is to say, $J_m(a, w) = 0$ implies $a = Pa^*$ and $w = Pw^*$ with $P \in \mathbb{R}^{m \times m}$ to be some permutation matrix. This gives $m!$ global minimizers.

(ii) Coincided critical affine spaces. Obviously $\mathcal{N} \neq \varnothing$, and $M = \sup \mathcal{N} \geq m$. According to Theorem D.1, for any $d \in \mathbb{N}_+$, $1 \leq d \leq \min\{m, M\} = m$, we have at least $d! \begin{Bmatrix} m \\ d \end{Bmatrix}$ $d$-coincided critical affine spaces of $J_m$. By (i), for any $d \leq m-1$, there are no global minimizers in these affine spaces. Counting the total number

$$\sum_{d=1}^{m-1} d! \begin{Bmatrix} m \\ d \end{Bmatrix}. \tag{130}$$

(iii) Comparison. To give a bound between (130) and $m!$, we need an elementary recurrence

$$\begin{Bmatrix} m \\ d \end{Bmatrix} = d \begin{Bmatrix} m-1 \\ d \end{Bmatrix} + \begin{Bmatrix} m-1 \\ d-1 \end{Bmatrix}.$$

- For $d = m-1$, let $p_m := \begin{Bmatrix} m \\ m-1 \end{Bmatrix}$, then

$$p_m = (m-1) \begin{Bmatrix} m-1 \\ m-1 \end{Bmatrix} + \begin{Bmatrix} m-1 \\ m-2 \end{Bmatrix} = (m-1) + p_{m-1} = \cdots = \frac{m(m-1)}{2}.$$

---

[12] Although the assumption $m \in \mathcal{N}$ seems strong, we will provide sufficient conditions to guarantee its validity in Appendix D.1.2. See an complement in Theorem D.7.

- For $d = m - 2$, let $q_m := \left\{ {m \atop m-2} \right\}$, then

$$q_m = (m-2) \left\{ {m-1 \atop m-2} \right\} + \left\{ {m-1 \atop m-3} \right\} = (m-2)p_{m-1} + q_{m-1} = \cdots$$

$$= \frac{1}{24}[2(m-2)(m-1)(2m-3) + 3(m-2)^2(m-1)^2].$$

Combining above gives

$$\frac{1}{m!} \sum_{d=1}^{m-1} d! \left\{ {m \atop d} \right\} > \frac{1}{m!}[(m-1)!p_m + (m-2)!q_m] = \frac{(m+1)(3m-2)}{24},$$

which is a quadratic polynominal on $m$. The proof is completed. $\square$

**Remark D.2.** *We only take the last two terms of (130) for a lower bound, which is obviously rather loose. In principle, a $\mathrm{Poly}(m)$ bound with higher degrees can be similarly obtained. That is to say, on one hand, there are infinitely many critical points forming affine spaces in the landscape of $J_m$; on the other hand, we see that even if only counting the number of affine spaces, there are still much less global minimizers (provided the width $m$ relatively large).*

**Remark D.3.** *When the target $\rho$ is not an exponential sum, it is straightforward to see Theorem D.2 still holds if there is a finite number (with the scale of no more than factorial) of global minimizers.*

Now we get down to investigate $\nabla^2 J_m$ on the above coincided critical affine spaces. It is shown that $\nabla^2 J_m$ is singular and can have multiple zero eigenvalues.

**Theorem D.3.** *Fix any $m, d \in \mathbb{N}_+$, $1 \le d \le m$. On the $d$-coincided critical affine spaces (induced by non-degenerate global minimizers of $J_d$[13]) of $J_m$, $\nabla^2 J_m$ is with rank at most $m + d$, and hence has at least $m - d$ zero eigenvalues.*

*Proof.* A straightforward computation shows that, for $k, l = 1, 2, \cdots, m$,

$$\frac{\partial^2 J_m}{\partial a_k \partial a_l}(a, w) = \frac{2}{w_k + w_l}, \tag{131}$$

$$\frac{\partial^2 J_m}{\partial a_k \partial w_l}(a, w) = \frac{-2a_l}{(w_k + w_l)^2}, \quad k \ne l, \tag{132}$$

$$\frac{\partial^2 J_m}{\partial a_k \partial w_k}(a, w) = \frac{-a_k}{2w_k^2} + 2 \int_0^\infty (-t)e^{-w_k t} \left( \sum_{i=1}^m a_i e^{-w_i t} - \rho(t) \right) dt. \tag{133}$$

Let the induced $d$-coincided critical affine space be $\mathcal{M}_{\mathcal{P}, (\hat{b}, \hat{v}), (m, d)}$, as is derived in the proof of Theorem D.1. Since $(\hat{b}, \hat{v})$ is the non-degenerate global minimizer of $J_d$, we have

$$\int_0^\infty (-t)e^{-\hat{w}_k t} \left( \sum_{i=1}^m \hat{a}_i e^{-\hat{w}_i t} - \rho(t) \right) dt = \int_0^\infty (-t)e^{-\hat{v}_s t} \left( \sum_{j=1}^d \hat{b}_j e^{-\hat{v}_j t} - \rho(t) \right) dt$$

$$= \frac{1}{2\hat{b}_s} \frac{\partial J_d}{\partial v_s}(\hat{b}, \hat{v}) = 0, \quad k \in \mathcal{I}_s, \ s = 1, 2, \cdots, d$$

for any $(\hat{a}, \hat{w}) \in \mathcal{M}_{\mathcal{P}, (\hat{b}, \hat{v}), (m, d)}$. This gives

$$\frac{\partial^2 J_m}{\partial a_k \partial w_k}(\hat{a}, \hat{w}) = \frac{-\hat{a}_k}{2\hat{w}_k^2}. \tag{134}$$

Now we show that, for any $i, j \in \mathcal{I}_s$, $i \ne j$, $s = 1, 2, \cdots, d$, $\nabla^2 J_m(\hat{a}, \hat{w})_{i,:} = \nabla^2 J_m(\hat{a}, \hat{w})_{j,:}$. In fact, for any $k = 1, \cdots, m$, let $k \in \mathcal{I}_{s'}$, then by (131),

$$\frac{\partial^2 J_m}{\partial a_i \partial a_k}(\hat{a}, \hat{w}) = \frac{2}{\hat{w}_i + \hat{w}_k} = \frac{2}{\hat{v}_s + \hat{v}_{s'}}, \qquad \frac{\partial^2 J_m}{\partial a_j \partial a_k}(\hat{a}, \hat{w}) = \frac{2}{\hat{w}_j + \hat{w}_k} = \frac{2}{\hat{v}_s + \hat{v}_{s'}}.$$

---

[13]That is, the affine space $\mathcal{M}_{\mathcal{P}, (\hat{b}, \hat{v}), (m, d)}$. See details in the proof of Theorem D.1

For $k \neq i$ and $k \neq j$, (132) gives

$$\frac{\partial^2 J_m}{\partial a_i \partial w_k}(\hat{a}, \hat{w}) = \frac{-2\hat{a}_k}{(\hat{w}_i + \hat{w}_k)^2} = \frac{-2\hat{a}_k}{(\hat{v}_s + \hat{v}_{s'})^2}, \qquad \frac{\partial^2 J_m}{\partial a_j \partial w_k}(\hat{a}, \hat{w}) = \frac{-2\hat{a}_k}{(\hat{w}_j + \hat{w}_k)^2} = \frac{-2\hat{a}_k}{(\hat{v}_s + \hat{v}_{s'})^2}.$$

By (134), for $k = i \neq j$,

$$\frac{\partial^2 J_m}{\partial a_i \partial w_k}(\hat{a}, \hat{w}) = \frac{-\hat{a}_i}{2\hat{w}_i^2} = \frac{-\hat{a}_i}{2\hat{v}_s^2}, \qquad \frac{\partial^2 J_m}{\partial a_j \partial w_k}(\hat{a}, \hat{w}) = \frac{-2\hat{a}_i}{(\hat{w}_j + \hat{w}_i)^2} = \frac{-\hat{a}_i}{2\hat{v}_s^2},$$

and similarly for $k = j \neq i$,

$$\frac{\partial^2 J_m}{\partial a_i \partial w_k}(\hat{a}, \hat{w}) = \frac{-2\hat{a}_j}{(\hat{w}_i + \hat{w}_j)^2} = \frac{-\hat{a}_j}{2\hat{v}_s^2}, \qquad \frac{\partial^2 J_m}{\partial a_j \partial w_k}(\hat{a}, \hat{w}) = \frac{-\hat{a}_j}{2\hat{w}_j^2} = \frac{-\hat{a}_j}{2\hat{v}_s^2}.$$

That is to say, there are at most $m + d$ different rows in the symmetric matrix $\nabla^2 J_m(\hat{a}, \hat{w}) \in \mathbb{R}^{2m \times 2m}$, hence $\mathrm{rank}\big(\nabla^2 J_m(\hat{a}, \hat{w})\big) \leq m + d$. Therefore, the number of zero eigenvalues of $\nabla^2 J_m(\hat{a}, \hat{w}) \geq \dim\big\{x \in \mathbb{R}^{2m} : \nabla^2 J_m(\hat{a}, \hat{w}) \cdot x = 0\big\} = 2m - \mathrm{rank}(\nabla^2 J_m(\hat{a}, \hat{w})) \geq m - d$. The proof is completed. $\qquad \square$

**Remark D.4.** *The bound in Theorem D.3 is not sharp, since the estimate on $\mathrm{rank}\big(\nabla^2 J_m\big)$ here is loose as only rows with the same elements are considered. In practice (numerical tests), it is often observed that there are more zero eigenvalues of $\mathrm{rank}\big(\nabla^2 J_m\big)$ on the coincided critical affine space $\mathcal{M}_{\mathcal{P},(\hat{b},\hat{v}),(m,d)}$.*

**Remark D.5.** *Theorem D.3 shows that, there are local plateaus around the $d$-coincided critical affine spaces $\mathcal{M}_{\mathcal{P},(\hat{b},\hat{v}),(m,d)}$ for $d \leq m - 1$. In addition, the $0$-eigenspace of $J_m$ is higher-dimensional for smaller $d$, which may suggest that one can stuck on plateaus more easily.*

### D.1.2 SUFFICIENT CONDITIONS

There is still a gap when connecting Theorem D.1 and Theorem D.2. That is, it is necessary to guarantee $\sup \mathcal{N}$ relatively large, i.e. $J_1, J_2, \cdots, J_d$ all have non-degenerate global minimizers for $d$ as large as possible. Motivated by Kammler (1979a), we can give some sufficient conditions by limiting the target $\rho$ within a smaller function space, the so-called completely monotonic functions.

**Definition D.3.** $F \in C[0, \infty] \cap C^\infty(0, \infty)$ *is called completely monotonic, if and only if*

$$(-1)^n F^{(n)}(t) \geq 0, \quad 0 < t < \infty, \ n = 0, 1, \cdots,$$

*and $F(\infty) = 0$.*

**Remark D.6.** *Several examples of completely monotonic functions:*

- *$\rho(t) = 1/(1 + t)^\alpha$ for any $\alpha > 0$;*

- *The non-degenerate exponential sum with positive coefficients*

$$\rho(t) = \sum_{j=1}^{m^*} a_j^* e^{-w_j^* t}, \quad 0 \leq w_1^* < \cdots < w_{m^*}^*, \ a_j^* > 0, \ j = 1, 2, \cdots, m^*.$$

Since the space of exponential sums is not close, we turn to consider the problem of finding a best approximation to a given $\rho \in L^2[0, \infty)$ from the set

$$V_d(\mathbb{R}_+) := \big\{\hat{\rho} \in C^d[0, \infty) : [(D + w_1) \cdots (D + w_d)]\hat{\rho} = 0 \text{ for some } w_1, \cdots w_d \in \mathbb{R}_+\big\} \quad (135)$$

with respect to the common $L^2$-norm, i.e. $\inf_{\hat{\rho} \in V_d(\mathbb{R}_+)} \|\hat{\rho} - \rho\|_{L^2[0,\infty)}$, where $D$ denotes the common differential operator. Obviously $V_d(\mathbb{R}_+) \subset L^2[0, \infty)$ and $V_d(\mathbb{R}_+) \subsetneq V_{d+1}(\mathbb{R}_+)$ for any $d \in \mathbb{N}_+$.

Kammler (1979a) proves the following theorem.

**Theorem D.4.** *Assume $\rho \in L^2[0, \infty)$ to be completely monotonic. Then there exists a best approximation $\hat{\rho}^0$ to $\rho$ in $V_d(\mathbb{R}_+)$, i.e.*

$$\|\hat{\rho}^0 - \rho\|_{L^2[0,\infty)} = \inf_{\hat{\rho} \in V_d(\mathbb{R}_+)} \|\hat{\rho} - \rho\|_{L^2[0,\infty)}. \quad (136)$$

When $\rho \notin V_d(\mathbb{R}_+)$, any such best approximation admits a non-degenerate form

$$\hat{\rho}^0(t) = \sum_{j=1}^{d} \hat{b}_j e^{-\hat{v}_j t}, \quad 0 < \hat{v}_1 < \cdots < \hat{v}_d, \ \hat{b}_j > 0, \ j = 1, 2, \cdots, d, \tag{137}$$

and satisfies the generalized Aigrain-Williams equations

$$\mathcal{L}[\hat{\rho}^0](\hat{v}_j) = \mathcal{L}[\rho](\hat{v}_j), \quad j = 1, 2, \cdots, d, \tag{138}$$

$$\frac{d}{ds}\mathcal{L}[\hat{\rho}^0](s)\Big|_{s=\hat{v}_j} = \frac{d}{ds}\mathcal{L}[\rho](s)\Big|_{s=\hat{v}_j}, \quad j = 1, 2, \cdots, d. \tag{139}$$

Note that (136) and (137) are pretty similar to Definition D.2, except for a different choice of hypothesis function space. Now we show a connection between these two problems.

**Theorem D.5.** *Assume $\rho \in L^2[0, \infty)$ to be completely monotonic, and $\rho \notin V_d(\mathbb{R}_+)$ for some $d \in \mathbb{N}_+$. Then $J_d$ has non-degenerate global minimizers $(\hat{b}, \hat{v}) \in \mathbb{R}^d \otimes \mathbb{R}_+^d$.*

*Proof.* According to Theorem D.4, there exists a non-degenerate best approximation $\hat{\rho}^0$ to $\rho$ from $V_d(\mathbb{R}_+)$, i.e.

$$\|\hat{\rho}^0 - \rho\|_{L^2[0,\infty)} = \inf_{\hat{\rho} \in V_d(\mathbb{R}_+)} \|\hat{\rho} - \rho\|_{L^2[0,\infty)}, \tag{140}$$

$$\hat{\rho}^0(t) = \sum_{j=1}^{d} \hat{b}_j e^{-\hat{v}_j t}, \quad 0 < \hat{v}_1 < \cdots < \hat{v}_d, \ \hat{b}_j > 0, \ j = 1, 2, \cdots, d. \tag{141}$$

We aim to prove $J_d(\hat{b}, \hat{v}) = \inf_{b \in \mathbb{R}^d, v \in \mathbb{R}_+^d} J_d(b, v)$. Define the following subsets of exponential sums

$$\mathcal{V}_d(\mathbb{R}_+) := \left\{ \hat{\rho} : \hat{\rho}(t) = \sum_{i=1}^{d} a_i e^{-w_i t}, a_i \in \mathbb{R}, w_i > 0 \right\},$$

$$\mathcal{V}_{d,k}(\mathbb{R}_+) := \left\{ \hat{\rho} \in \mathcal{V}_d(\mathbb{R}_+) : w = (w_i) \text{ has } k \text{ different components} \right\}, \quad 1 \le k \le d,$$

then we have $\inf_{b \in \mathbb{R}^d, v \in \mathbb{R}_+^d} J_d(b, v) = \inf_{\hat{\rho} \in \mathcal{V}_d(\mathbb{R}_+)} \|\hat{\rho} - \rho\|_{L^2[0,\infty)}^2$. It is straightforward to verify that $\mathcal{V}_d(\mathbb{R}_+) = \bigcup_{k=1}^{d} \mathcal{V}_{d,k}(\mathbb{R}_+)$, and $\mathcal{V}_{d,k}(\mathbb{R}_+) = \mathcal{V}_{k,k}(\mathbb{R}_+) \subsetneq V_k(\mathbb{R}_+)$ for $k = 1, \cdots, d$. By (140), we get

$$\|\hat{\rho}^0 - \rho\|_{L^2[0,\infty)}^2 = \inf_{\hat{\rho} \in V_d(\mathbb{R}_+)} \|\hat{\rho} - \rho\|_{L^2[0,\infty)}^2 \le \inf_{\hat{\rho} \in \mathcal{V}_{d,d}(\mathbb{R}_+)} \|\hat{\rho} - \rho\|_{L^2[0,\infty)}^2.$$

Since $\hat{\rho}^0 \in \mathcal{V}_{d,d}(\mathbb{R}_+)$, we have

$$J_d(\hat{b}, \hat{v}) = \|\hat{\rho}^0 - \rho\|_{L^2[0,\infty)}^2 = \inf_{\hat{\rho} \in \mathcal{V}_{d,d}(\mathbb{R}_+)} \|\hat{\rho} - \rho\|_{L^2[0,\infty)}^2.$$

The last task is to show $\inf_{\hat{\rho} \in \mathcal{V}_d(\mathbb{R}_+)} \|\hat{\rho} - \rho\|_{L^2[0,\infty)} = \inf_{\hat{\rho} \in \mathcal{V}_{d,d}(\mathbb{R}_+)} \|\hat{\rho} - \rho\|_{L^2[0,\infty)}$. In fact, for any $\hat{\rho} \in \mathcal{V}_{k,k}$, $\hat{\rho}(t) = \sum_{i=1}^{k} a_i e^{-w_i t}$, let $\tilde{a} := (a_1, \cdots, a_k, 0)$, $\tilde{w} := (w_1, \cdots, w_k, 1 + \max_{1 \le i \le k} w_i)$, we get $\hat{\rho}(t) := \sum_{i=1}^{k+1} \tilde{a}_i e^{-\tilde{w}_i t} \in \mathcal{V}_{k+1,k+1}$, which implies $\mathcal{V}_{k,k} \subset \mathcal{V}_{k+1,k+1}$. Therefore,

$$\inf_{\hat{\rho} \in \mathcal{V}_d(\mathbb{R}_+)} \|\hat{\rho} - \rho\|_{L^2[0,\infty)} = \inf_{\hat{\rho} \in \bigcup_{k=1}^{d} \mathcal{V}_{d,k}(\mathbb{R}_+)} \|\hat{\rho} - \rho\|_{L^2[0,\infty)} = \min_{1 \le k \le d} \left\{ \inf_{\hat{\rho} \in \mathcal{V}_{d,k}(\mathbb{R}_+)} \|\hat{\rho} - \rho\|_{L^2[0,\infty)} \right\}$$

$$= \min_{1 \le k \le d} \left\{ \inf_{\hat{\rho} \in \mathcal{V}_{k,k}(\mathbb{R}_+)} \|\hat{\rho} - \rho\|_{L^2[0,\infty)} \right\} \ge \inf_{\hat{\rho} \in \mathcal{V}_{d,d}(\mathbb{R}_+)} \|\hat{\rho} - \rho\|_{L^2[0,\infty)},$$

which completes the proof. □

Combining Theorem D.1 and Theorem D.5 immediately gives the following result.

**Theorem D.6.** *Assume $\rho \in L^2[0,\infty)$ to be completely monotonic, and $\rho \notin V_1(\mathbb{R}_+)$. Let $\mathcal{D} := \{d \in \mathbb{N}_+ : \rho \notin V_d(\mathbb{R}_+)\}$, $D_0 := \sup \mathcal{D}$ and write $m' := \min\{m, D_0\}$. Then the total number of coincided critical affine spaces of $J_m$ is at least $\sum_{d=1}^{m'} d! \begin{Bmatrix} m \\ d \end{Bmatrix}$.*

*Proof.* We have $1 \in \mathcal{D}$ and thus $\mathcal{D} \neq \varnothing$, $D_0 \geq 1$. Since $V_d(\mathbb{R}_+) \subsetneq V_{d+1}(\mathbb{R}_+)$ for any $d \in \mathbb{N}_+$, we have $\mathcal{D} = \{1, 2, \cdots, D_0\}$ if $D_0 < \infty$, and $\mathcal{D} = \mathbb{N}_+$ if $D_0 = \infty$.[14] Both of them gives $\{1, 2, \cdots, m'\} \subset \mathcal{D}$, i.e. $\rho \notin V_k(\mathbb{R}_+)$ for any $k \leq m'$. By Theorem D.5, $J_{(k)}$ has non-degenerate global minimizers for any $k \leq m'$, i.e. $m' \in \mathcal{N}$. According to Theorem D.1, for any $d \in \mathbb{N}_+$, $1 \leq d \leq m' = \min\{m, m'\} \leq \min\{m, M\}$, there exists at least $d! \begin{Bmatrix} m \\ d \end{Bmatrix}$ $d$-coincided critical affine spaces of $J_m$. Sum over $d$ gives the total number $\sum_{d=1}^{m'} d! \begin{Bmatrix} m \\ d \end{Bmatrix}$. $\square$

**Remark D.7.** *Examples:*

- *Suppose the target is $\rho(t) = 1/(1+t)^\alpha$, $\alpha > 0$, then $\mathcal{D} = \mathbb{N}_+$ and $D_0 = \infty$. The total number of coincided critical affine spaces of the corresponding $J_m$ is at least $\sum_{d=1}^{m} d! \begin{Bmatrix} m \\ d \end{Bmatrix}$.*

- *Suppose the target is an non-degenerate exponential sum with positive coefficients: $\rho(t) = \sum_{j=1}^{m} a_j^* e^{-w_j^* t}$, where $a_j^* > 0$ and $w_i^* \neq w_j^*$ for any $i \neq j$, $i, j = 1, \cdots, m$. Then $\mathcal{D} = \{1, 2, \cdots, m-1\}$ and $D_0 = m - 1$. The total number of coincided critical affine spaces of the corresponding $J_m$ is at least $\sum_{d=1}^{m-1} d! \begin{Bmatrix} m \\ d \end{Bmatrix}$, which is exactly (130).*

An complement for Theorem D.2 is as follows.

**Theorem D.7.** *Fix any $m \in \mathbb{N}_+$ relatively large. Consider the loss $J_m$ associated with the target being a non-degenerate exponential sum with positive coefficients, i.e. $\rho(t) = \sum_{j=1}^{m} a_j^* e^{-w_j^* t}$, where $a_j^* > 0$ and $w_i^* \neq w_j^*$ for any $i \neq j$, $i, j = 1, \cdots, m$. Then in the landscape of $J_m$, the number of coincided critical affine spaces is at least $Poly(m)$ times larger than the number of global minimizers.*

*Proof.* By Theorem D.2, we only need to show is $m \in \mathcal{N}$. Since $\rho \in L^2[0,\infty)$ is completely monotonic, and $\rho \notin V_k(\mathbb{R}_+)$ for any $k \leq m - 1$, then by Theorem D.5, $J_{(k)}$ has non-degenerate global minimizers for any $k \leq m - 1$, i.e. $m - 1 \in \mathcal{N}$. The proof is completed by noticing that $J_m$ obviously has non-degenerate global minimizers, e.g. $(a^*, w^*)$. $\square$

### D.1.3 A Low-dimensional Example

To further understand the structure of coincided critical affine spaces, we focus on a specific low-dimensional example here. That is

$$\min_{a \in \mathbb{R}^2, w \in \mathbb{R}_+^2} J_2(a, w) = \left\| \sum_{i=1}^{2} a_i e^{-w_i t} - \rho(t) \right\|_{L^2[0,\infty)}^2,$$

with the target to be a non-degenerate exponential sum $\rho(t) = \sum_{j=1}^{m^*} a_j^* e^{-w_j^* t}$, where $a_j^* \neq 0$ and $w_i^* \neq w_j^*$ for any $i \neq j$, $i, j = 1, \cdots, m^*$. As we will show later, the coincided critical affine spaces of $J_2$ contain both saddles and degenerate stable points which are not global optimal.

By Lemma D.1, Remark D.1 and Theorem D.1, we know the 1-coincided critical affine space of $J_2$ exists, and it can be constructed by taking the non-degenerate global minimizer of $J_1$, say $(\hat{a}, \hat{w})$

---

[14]In fact, $V_d(\mathbb{R}_+) \subsetneq V_{d+1}(\mathbb{R}_+)$ for any $d \in \mathbb{N}_+$ implies if $\rho \notin V_d(\mathbb{R}_+)$, $\rho \notin V_k(\mathbb{R}_+)$ for any $k \leq d$, i.e. $d \in \mathcal{D} \Rightarrow k \in \mathcal{D}$ for any $k \leq d$; otherwise, if $\rho \in V_d(\mathbb{R}_+)$, $\rho \in V_l(\mathbb{R}_+)$ for any $l \geq d$, i.e. $d \notin \mathcal{D} \Rightarrow l \notin \mathcal{D}$ for any $l \geq d$.

with $\hat{a} \neq 0$ and $\hat{w} > 0$. Then $\mathcal{M}_{(\hat{a},\hat{w}),(2,1)} := \{(a_1, \hat{a} - a_1, \hat{w}, \hat{w}) : a_1 \in \mathbb{R}\} \in \mathbb{R}^4$ is a line [15], and $\nabla J_2(a_1, \hat{a} - a_1, \hat{w}, \hat{w}) = 0$ for any $a_1 \in \mathbb{R}$. Denote the Hessian of $J_2$ on the line $\mathcal{M}_{(\hat{a},\hat{w}),(2,1)}$ by $\mathcal{A}_{(\hat{a},\hat{w})}(a_1)$, i.e. $\mathcal{A}_{(\hat{a},\hat{w})}(a_1) := \nabla^2 J_2(a_1, \hat{a} - a_1, \hat{w}, \hat{w})$. We investigate the landscape of $J_2$ on the line $\mathcal{M}_{(\hat{a},\hat{w}),(2,1)}$ by analyzing the eigenvalue distribution of $\mathcal{A}_{(\hat{a},\hat{w})}(a_1)$.

**Proposition D.1.** *Suppose $m = m^* = 2$, and $0 < w_1^* < w_2^*$. Let $I_1 := [0, \hat{a}]$ and $I_2 := (-\infty, 0) \cup (\hat{a}, +\infty)$ [16]. Then*

1. *If $a_1^* a_2^* < 0$, the minimal eigenvalue of $\mathcal{A}_{(\hat{a},\hat{w})}(a_1)$ is 0 for any $a_1 \in I_1$, and negative for any $a_1 \in I_2$;*

2. *If $a_1^* a_2^* > 0$ and $w_2^*/w_1^* < 2 + \sqrt{3}$, the minimal eigenvalue of $\mathcal{A}_{(\hat{a},\hat{w})}(a_1)$ is negative for any $a_1 \in I_1$, and 0 for any $a_1 \in I_2$.*

*Proof.* Write $c(w) := \sum_{j=1}^{m^*} a_j^* \left[ \frac{1}{2w(w+w_j^*)^2} - \frac{1}{(w+w_j^*)^3} \right]$, and $a_2 := \hat{a} - a_1$. A straightforward computation shows that

$$\mathcal{A}_{(\hat{a},\hat{w})}(a_1) = \begin{bmatrix} \frac{1}{\hat{w}} & \frac{1}{\hat{w}} & \frac{-a_1}{2\hat{w}^2} & \frac{-a_2}{2\hat{w}^2} \\ \frac{1}{\hat{w}} & \frac{1}{\hat{w}} & \frac{-a_1}{2\hat{w}^2} & \frac{-a_2}{2\hat{w}^2} \\ \frac{-a_1}{2\hat{w}^2} & \frac{-a_1}{2\hat{w}^2} & \frac{a_1^2}{2\hat{w}^3} + 4c(\hat{w})a_1 & \frac{a_1 a_2}{2\hat{w}^3} \\ \frac{-a_2}{2\hat{w}^2} & \frac{-a_2}{2\hat{w}^2} & \frac{a_1 a_2}{2\hat{w}^3} & \frac{a_2^2}{2\hat{w}^3} + 4c(\hat{w})a_2 \end{bmatrix}.$$

Considering the congruent transformation of $\mathcal{A}_{(\hat{a},\hat{w})}(a_1)$, which does not affect the index of inertia:

$$\mathcal{A}_{(\hat{a},\hat{w})}(a_1) = \begin{bmatrix} \frac{1}{\hat{w}} & \frac{1}{\hat{w}} & \frac{-a_1}{2\hat{w}^2} & \frac{-a_2}{2\hat{w}^2} \\ \frac{1}{\hat{w}} & \frac{1}{\hat{w}} & \frac{-a_1}{2\hat{w}^2} & \frac{-a_2}{2\hat{w}^2} \\ \frac{-a_1}{2\hat{w}^2} & \frac{-a_1}{2\hat{w}^2} & \frac{a_1^2}{2\hat{w}^3} + 4c(\hat{w})a_1 & \frac{a_1 a_2}{2\hat{w}^3} \\ \frac{-a_2}{2\hat{w}^2} & \frac{-a_2}{2\hat{w}^2} & \frac{a_1 a_2}{2\hat{w}^3} & \frac{a_2^2}{2\hat{w}^3} + 4c(\hat{w})a_2 \end{bmatrix}$$

$$\rightarrow \begin{bmatrix} \frac{1}{\hat{w}} & 0 & 0 & 0 \\ 0 & 0 & 0 & 0 \\ 0 & 0 & \frac{a_1^2}{4\hat{w}^3} + 4c(\hat{w})a_1 & \frac{a_1 a_2}{4\hat{w}^3} \\ 0 & 0 & \frac{a_1 a_2}{4\hat{w}^3} & \frac{a_2^2}{4\hat{w}^3} + 4c(\hat{w})a_2 \end{bmatrix},$$

we see that $\mathcal{A}_{(\hat{a},\hat{w})}(a_1)$ has one positive eigenvalue $1/\hat{w}$ and one eigenvalue 0. What remains are the eigenvalues of $\mathcal{A}'_{(\hat{a},\hat{w})}(a_1) := \begin{bmatrix} \frac{a_1^2}{4\hat{w}^3} + 4c(\hat{w})a_1 & \frac{a_1 a_2}{4\hat{w}^3} \\ \frac{a_1 a_2}{4\hat{w}^3} & \frac{a_2^2}{4\hat{w}^3} + 4c(\hat{w})a_2 \end{bmatrix}$. To determine their signs, we compute

$$\det(\mathcal{A}'_{(\hat{a},\hat{w})}(a_1)) = a_1(\hat{a} - a_1) \cdot 4c(\hat{w}) \left( \frac{\hat{a}}{4\hat{w}^3} + 4c(\hat{w}) \right) \tag{142}$$

$$= \frac{1}{\hat{a}^2 \hat{w}^3} a_1(\hat{a} - a_1) \cdot \hat{a}c(\hat{w}) \cdot \left( \hat{a}^2 + 16\hat{w}^3 \hat{a}c(\hat{w}) \right). \tag{143}$$

So we need to analyze the sign of $\hat{a}c(\hat{w})$ and $\hat{a}^2 + 16\hat{w}^3 \hat{a}c(\hat{w})$ under different assumptions on $(a^*, w^*)$.

(i) $a_1^* a_2^* < 0$. By the optimality condition of $(\hat{a}, \hat{w})$ for $J_1$, we have

$$\hat{a} = 2\hat{w} \sum_{j=1}^{m^*} \frac{a_j^*}{\hat{w} + w_j^*} = 4\hat{w}^2 \sum_{j=1}^{m^*} \frac{a_j^*}{(\hat{w} + w_j^*)^2}, \tag{144}$$

and therefore

$$c(\hat{w}) = \frac{\hat{a}}{8\hat{w}^3} - \sum_{j=1}^{m^*} \frac{a_j^*}{(\hat{w} + w_j^*)^3}.$$

---

[15] Here we omit the corresponding partition $\mathcal{P}$ since it is unique.

[16] Suppose $\hat{a} > 0$ here without loss of generality. If $\hat{a} < 0$, we let $I_1 := [\hat{a}, 0]$ and $I_2 := (-\infty, \hat{a}) \cup (0, +\infty)$ and the same conclusions hold.

Write $v_j := w_j^*/\hat{w}$, $j = 1, 2$, we get $0 < v_1 < v_2$, and

$$\hat{a} = 2\sum_{j=1}^{m^*} \frac{a_j^*}{1+v_j} = 4\sum_{j=1}^{m^*} \frac{a_j^*}{(1+v_j)^2}, \quad \hat{w}^3 c(\hat{w}) = \frac{\hat{a}}{8} - \sum_{j=1}^{m^*} \frac{a_j^*}{(1+v_j)^3}.$$

Therefore

$$8\hat{w}^3\hat{a}c(\hat{w}) = \hat{a}^2 - 8\hat{a}\sum_{j=1}^{m^*} \frac{a_j^*}{(1+v_j)^3}$$

$$= 16\left[\sum_{j=1}^{m^*} \frac{a_j^*}{(1+v_j)^2}\right]^2 - 16\sum_{j=1}^{m^*} \frac{a_j^*}{1+v_j} \cdot \sum_{j=1}^{m^*} \frac{a_j^*}{(1+v_j)^3}$$

$$= \frac{-16a_1^* a_2^* (v_1 - v_2)^2}{(1+v_1)^3(1+v_2)^3} \tag{145}$$

$$> 0,$$

which gives $\hat{a}c(\hat{w}) > 0$ and $\hat{a}^2 + 16\hat{w}^3\hat{a}c(\hat{w}) > 8\hat{w}^3\hat{a}c(\hat{w}) > 0$.

(ii) $a_1^* a_2^* > 0$, $w_2^*/w_1^* < 2 + \sqrt{3}$. By (145), $\hat{a}c(\hat{w}) < 0$.

$$\hat{a}^2 + 16\hat{w}^3\hat{a}c(\hat{w}) = 3\hat{a}^2 - 16\hat{a}\sum_{j=1}^{m^*} \frac{a_j^*}{(1+v_j)^3}$$

$$= 16\left\{3\left[\sum_{j=1}^{m^*} \frac{a_j^*}{(1+v_j)^2}\right]^2 - 2\sum_{j=1}^{m^*} \frac{a_j^*}{1+v_j} \cdot \sum_{j=1}^{m^*} \frac{a_j^*}{(1+v_j)^3}\right\}$$

$$= \frac{a_1^{*2}}{u_1^4 u_2^4}\left[u_2^4 + c^2 u_1^4 + 6cu_1^2 u_2^2 - 2cu_1^3 u_2 - 2cu_1 u_2^3\right] \quad (c = a_2^*/a_1^*, u_j := 1 + v_j > 1)$$

$$= \frac{a_1^{*2}}{u_2^4}\left(s^4 + c^2 + 6cs^2 - 2cs - 2cs^3\right) \quad (s = u_2/u_1 > 1)$$

$$= \frac{a_1^{*2}}{u_2^4}\left[c^2 - 2s(s^2 - 3s + 1)c + s^4\right].$$

Since $4s^2(s^2 - 3s + 1)^2 - 4s^4 = 4s^2(s-1)^2[(s-2)^2 - 3]$, and $1 < s = u_2/u_1 = (\hat{w} + w_2^*)/(\hat{w} + w_1^*) < w_2^*/w_1^* < 2 + \sqrt{3}$, we get $\Delta_c < 0$. This implies $c^2 - 2s(s^2 - 3s + 1)c + s^4 > 0$ and $\hat{a}^2 + 16\hat{w}^3\hat{a}c(\hat{w}) > 0$.

In both (i) and (ii), $\hat{a}^2 + 16\hat{w}^3\hat{a}c(\hat{w}) > 0$, which implies that there is at least one positive diagonal element of $\mathcal{A}'_{(\hat{a},\hat{w})}(a_1)$ in a sufficiently small neighborhood of $a_1 = 0$ and $a_1 = \hat{a}$. By the Rayleigh-Ritz Theorem and Weyl's Theorem, $\mathcal{A}'_{(\hat{a},\hat{w})}(a_1)$ has at least one positive eigenvalue in this neighborhood. However, by (142), $\det(\mathcal{A}'_{(\hat{a},\hat{w})}(a_1))$ only changes the sign at $a_1 = 0$ and $a_1 = \hat{a}$. This implies another eigenvalue of $\mathcal{A}'_{(\hat{a},\hat{w})}(a_1)$ changes the sign at $a_1 = 0$ and $a_1 = \hat{a}$ accordingly. By different signs of $\hat{a}c(\hat{w})$ derived in (i) and (ii), and (142), the proof is completed. $\square$

**Remark D.8.** *From Proposition D.1, we see that there are both saddles and degenerate stable points of $J_2$ on the critical affine spaces (line) $\mathcal{M}_{(\hat{a},\hat{w}),(2,1)}$, and each of them in fact forms affine spaces (lines) respectively, but they are certainly not global minimizers. Therefore, the gradient-based algorithms can get stuck around this affine space, except that it meets saddles with negative eigenvalues of large magnitude.*

## E    MOMENTUM HELPS TRAINING: QUADRATIC EXAMPLES

In practice, it is often the case that training is trapped in some very flat regions (plateaus), where the loss function has rather small gradients and negative eigenvalues of Hessian. Now we illustrate the escape dynamics (from plateaus) via a simple quadratic example.

Consider the loss function $f(x) = (x_1^2 - \epsilon x_2^2)/2$ with $0 < \epsilon \ll 1$. We check the escaping performance for continuous-in-time analogs of two optimization algorithms: gradient decent (GD) and momentum (heavy ball) method.

**(1) Gradient decent**

Consider the gradient flow of $f(x)$ with an initial value $x_0 = (\delta, 1)^\top$, where $0 < \delta \ll 1$ and $\delta = O(\epsilon)$. Thus $\|\nabla f(x_0)\|_2 = O(\epsilon)$, and

$$\begin{cases} x_1'(\tau) = -x_1(\tau), & x_1(0) = \delta \\ x_2'(\tau) = \epsilon x_2(\tau), & x_2(0) = 1 \end{cases} \Rightarrow \begin{cases} x_1(\tau) = \delta e^{-\tau} \\ x_2(\tau) = e^{\epsilon \tau} \end{cases}$$

$$\Rightarrow f(x(\tau)) = (\delta^2 e^{-2\tau} - \epsilon e^{2\epsilon\tau})/2 =: \ell_1(\tau).$$

It is easy to show that there are different timescales of $\ell_1(\tau)$. In fact, when $\tau = O(1/\epsilon)$, $\ell_1(\tau) = O(\epsilon^2)e^{-|O(1/\epsilon)|} - \epsilon e^{|O(1)|} = O(\epsilon)$. However, when $\tau$ continuous to increase, say

$$\tau \geq \frac{1}{2\epsilon} \ln \frac{\delta_0}{\epsilon} =: \tau_1^\epsilon, \tag{146}$$

where $\delta_0 > 0$ denotes the gap satisfying $\epsilon = o(\delta_0)$, we get $\ell_1(\tau) \leq \ell_1(\tau_1^\epsilon) = O(\epsilon^2) - \epsilon e^{2\epsilon \cdot \frac{1}{2\epsilon} \ln \frac{\delta_0}{\epsilon}}/2 = O(\epsilon^2) - \delta_0/2 < -\delta_0/4$ for any $\tau \geq \tau_1^\epsilon$.

**(2) Momentum**

The momentum algorithm has the update rule

$$x_{k+1} = x_k - \eta \nabla f(x_k) + \rho(x_k - x_{k-1}), \tag{147}$$

where $\rho \in \mathbb{R}$, $\eta > 0$ is the learning rate. The continuous-in-time analog can be derived as (similar to the arguments in Su et al. (2014))

$$0 = \rho \frac{x_{k+1} - 2x_k + x_{k-1}}{\eta} + \frac{(1-\rho)}{\sqrt{\eta}} \frac{x_{k+1} - x_k}{\sqrt{\eta}} + \nabla f(x_k)$$

$$\approx \rho x''(t) + \frac{(1-\rho)}{\sqrt{\eta}} x'(t) + \nabla f(x(t)),$$

with $x_k := x(k\sqrt{\eta})$ and the step size $\sqrt{\eta}$ of the simple finite differences [17]. Let $x_1 = x_0 - \eta \nabla f(x_0)$, we also get $x'(0) = -\sqrt{\eta} \nabla f(x(0))$.

To facilitate a comparison to GD, we take $\eta = 1$ [18] and $\rho = 1$ [19]. Plugging the expression of $f$, we can solve the ODE

$$x''(t) + \nabla f(x(t)) = 0 \Leftrightarrow \begin{cases} x_1''(\tau) + x_1(\tau) = 0, & x_1(0) = \delta, \quad x_1'(0) = -\delta \\ x_2''(\tau) - \epsilon x_2(\tau) = 0, & x_2(0) = 1, \quad x_2'(0) = \epsilon \end{cases}$$

$$\Rightarrow \begin{cases} x_1(\tau) = \delta(\cos\tau - \sin\tau) \\ x_2(\tau) = \frac{1+\sqrt{\epsilon}}{2} e^{\sqrt{\epsilon}\tau} + \frac{1-\sqrt{\epsilon}}{2} e^{-\sqrt{\epsilon}\tau} \end{cases}$$

$$\Rightarrow f(x(\tau)) = \frac{1}{2}\left[ \delta^2(\cos\tau - \sin\tau)^2 - \epsilon \left( \frac{1+\sqrt{\epsilon}}{2} e^{\sqrt{\epsilon}\tau} + \frac{1-\sqrt{\epsilon}}{2} e^{-\sqrt{\epsilon}\tau} \right)^2 \right].$$

Write $\ell_2(\tau) := f(x(\tau))$. It is not hard to show that there are still different timescales of $\ell_2(\tau)$. In fact, when $\tau = O(1/\sqrt{\epsilon})$, $\ell_2(\tau) = O(\epsilon^2)|O(1)| - \epsilon |O(1)|(e^{|O(1)|} + e^{-|O(1)|})^2 = O(\epsilon)$. However, when $\tau$ continuous to increase, say

$$\tau \geq \frac{1}{2\sqrt{\epsilon}} \ln \frac{4\delta_0}{\epsilon} =: \tau_2^\epsilon, \tag{148}$$

we get $\ell_2(\tau_2^\epsilon) = O(\epsilon^2) - \epsilon(\frac{1+\sqrt{\epsilon}}{2})^2 e^{2\sqrt{\epsilon}\tau}/2 + O(\epsilon) = O(\epsilon) - \delta_0 < -\delta_0/2$, hence $\ell_2(\tau) \leq O(\epsilon) - \delta_0 < -\delta_0/2$ for any $\tau \geq \tau_2^\epsilon$.

Combining **(1)** and **(2)**, we have the following conclusions.

---

[17] It is easy to check that the error of discretization is of order $O(\sqrt{\eta})$.

[18] In the continuous-in-time analog of GD, i.e. the gradient flow, the step size is taken as 1.

[19] As is seen later, $\rho = 1$ not only simplifies the analysis, but also helps to obtain the best acceleration.

- For both training dynamics, there are different timescales in the loss function. That is to say, relatively long time is needed to escape from the plateaus;

- Comparing (146) and (148), we get different timescales of escaping: $O\left(1/\epsilon \cdot \ln(1/\epsilon)\right)$ for GD and $O\left(1/\sqrt{\epsilon} \cdot \ln(1/\epsilon)\right)$ for momentum. Just like the convex case, where momentum improves the convergence rate by weakening the dependence on condition number, we see momentum can also help to escape rather flat saddles.

