# OpenReview forum: "On the Curse of Memory in Recurrent Neural Networks: Approximation and Optimization Analysis"
_ICLR.cc/2021/Conference — ICLR 2021 Poster_

### Official Review · AnonReviewer4 · 2020-10-28
**Official Blind Review #4**

**Rating:** 8
**Confidence:** 4

**Review:**

The paper provides a theoretical examination of the challenge of fitting recurrent neural networks (RNNs) to fit processes with long memory (or long-range dependence). Dubbed the “curse of memory”, the author(s) restrict to the case of linear activation functions, and show that for processes with increased spatial dependence: (a) the width of the layers must increase exponentially to _guarantee_ accurate approximations under a provided bound, and (b) a gradient-based optimization algorithm will take exponentially more time to converge. Sufficient details for reproducing the experiments are provided.

I like the paper and believe the contributions to be both substantive and of wide interest in the RNN community. The analysis presented here is rigorous and comprehensive, and while the discussion is limited to linear RNNs, it provides a good starting point for further studies regarding long-range dependence with RNNs. The paper is reasonably well-written, and aside from a few minor typos — see the minor comments below — I did not encounter any serious errors. I do have some criticisms, however:

1. Perhaps my biggest disappointment is that while the authors do a good summary of the relevant RNN literature, they have failed to mention any analogous ideas from the time series literature. For example, any discussion regarding Hurst parameters would be welcome, since this is precisely the type of “long-range dependence” discussed here. The lack of references to this literature is disheartening, since these concepts have played a key role in time series analysis for many decades, and it would have been nice to see this acknowledged, or a connection made.
2. The takeaway from Theorem 4.2 regarding the exponential increase in the required width for fitting a linear RNN is nice, but is undercut by the fact that Theorem 4.2 is an upper bound. I'm satisfied with the result, but a lower bound for approximating a particular functional would be more convincing.
3. I was disappointed with the “informal” presentation here. I can appreciate the attempt to simplify the full statement, but I still had to go to the supplementary material to understand the statement, which I should hope most readers would not have to do. I think a more precise statement than “trapped in a plateau with timescale” would be better. Perhaps something involving the hitting time, or even more simply, $||\theta'(t)|| = O(...)$ as $\omega \to 0^+$ for sufficiently large $t$?

It is also worth noting that there is an extraordinary amount of supplementary material here, much of which, unfortunately, does not get the attention it probably deserves in this format. There are aspects of the paper, including the precise statement of Theorem 5.1 and the definition of the Airy target, that require the reader to visit this material. The proofs of Theorems 4.1 and 4.2 seem fine to me. Unfortunately, the proof of Theorem 5.1 is especially lengthy, so I did not get the opportunity to check each step in detail. However, the general argument appears sound.

Overall, I am impressed with this paper and enjoyed reading it. The missing connections to classical ideas in time series analysis are unfortunate, and I would be willing to give the paper an 8 with a wider literature review and improved presentation of Theorem 5.1. An additional lower bound for Theorem 4.2 would bring it to a 9 at least. But, even in its current form, I recommend this paper for acceptance to ICLR.

Other comments:
- I believe Theorem 4.1 also follows from the fact that matrix exponential distributions are dense (see [1]), together with the Riesz-Markov-Kakutani representation theorem used here.
- “We first show that the training dynamics of $\mathbb{E}_x J_m$ exhibits very interesting behaviors depending on the form of target functionals.” — This is a little too vague, maybe consider rewording?
- In multiple places: “x being/is the white noise” -> “white noise x” or “x is white noise”.
- Check capitalization in the statement under Theorem A.1.
- “Airy” should be capitalized in each appearance.

[1] He, Qi-Ming, and Hanqin Zhang. "On matrix exponential distributions." Advances in Applied Probability 39.1 (2007): 271-292.

---

> ### Author Response · Authors · 2020-11-17
> **AnonReviewer4**
>
> 1. "Perhaps my biggest disappointment is that while the authors do a good summary of the relevant RNN literature, they have failed to mention any analogous ideas from the time series literature. For example, any discussion regarding Hurst parameters would be welcome, since this is precisely the type of “long-range dependence” discussed here. The lack of references to this literature is disheartening, since these concepts have played a key role in time series analysis for many decades, and it would have been nice to see this acknowledged, or a connection made."
>    * We thank the reviewer for the suggestion. We have added some literature review on time series analysis pertaining to memory. We feel that the key difference here is we consider memory of input-output relationship and their interaction with RNN approximation/optimization. On the other hand, these works mainly focus on characterizing the memory structures in the input temporal data themselves. **We have added discussion on these prior works at the end of section 2.**
> 2. "The takeaway from Theorem 4.2 regarding the exponential increase in the required width for fitting a linear RNN is nice, but is undercut by the fact that Theorem 4.2 is an upper bound. I'm satisfied with the result, but a lower bound for approximating a particular functional would be more convincing."
>    * We fully agree with the reviewer's comment that Theorem 4.2 gives an upper-bound of the approximation error.
>    If we denote the minimum number of terms needed to achieve an $L^1$ error $\epsilon$ to the target density $1/t^{(1+\omega)}$ as $m(\omega,\epsilon)$, the estimate derived from Theorem 4.2 shows an upper-bound of  $m(\omega,\epsilon)$ goes to infinity exponentially fast as $\omega\rightarrow 0^+$ with fixed $\epsilon$. A stronger result would be that the lower bound of  $m(\omega,\epsilon) \rightarrow \infty$ exponentially fast as $\omega\rightarrow 0^+$ with fixed $\epsilon$. The divergence of $m(\omega,\epsilon)$ is somewhat natural, but the exponential divergence of its lower-bound is not obvious. We have some thoughts on it, but we have not proved it in general so far. This is a point of our future work. **We have made the point about upper/lower bounds clearer at the end of Section 4 on Page 6 when discussing our estimate.**
> 3. "I was disappointed with the “informal” presentation here. I can appreciate the attempt to simplify the full statement, but I still had to go to the supplementary material to understand the statement, which I should hope most readers would not have to do. I think a more precise statement than “trapped in a plateau with timescale” would be better. Perhaps something involving the hitting time, or even more simply,  ...."
>    * **We have improved the precision of Theorem 5.1 and added some explanatory paragraphs of the main proof ideas.**
> 4. "It is also worth noting that there is an extraordinary amount of supplementary material here, much of which, unfortunately, does not get the attention it probably deserves in this format. There are aspects of the paper, including the precise statement of Theorem 5.1 and the definition of the Airy target, that require the reader to visit this material. The proofs of Theorems 4.1 and 4.2 seem fine to me. Unfortunately, the proof of Theorem 5.1 is especially lengthy, so I did not get the opportunity to check each step in detail. However, the general argument appears sound."
>    * We agree that there is a large amount of technical details for the optimization part. This is because any precise analysis of non-convex optimization tends to be very complicated. **We have tried to improve the overall clarity of the stated results and sketched the main ideas after Theorem 5.1. We also improved the presentation in the appendix.**
> 5. "I believe Theorem 4.1 also follows from the fact that matrix exponential distributions are dense (see [1]), together with the Riesz-Markov-Kakutani representation theorem used here"
>    * Thank you for the suggested reference. Indeed, this result is related to the density of phase type distributions. Here, we consider signed measures instead of probability measures, but the approach should be generalizable (e.g. using Jordan decomposition). **We have added some references and stated the connection in the proof of Theorem 4.1 (Lemma A.2).**
> 6. **Minor issues are fixed.**

---

> > ### Comment · AnonReviewer4 · 2020-11-23
> > **Thank you for addressing most of my concerns**
> >
> > Thank you to the authors for taking the time to address my concerns. I'm quite pleased with the changes made overall, but I do have a few further comments. First, I understand that the new text is substantial and was written under significant time constraints, but I believe it needs a further editing pass before the camera-ready version. For example, for the summary about the time series literature, I recommend the following changes:
> >
> > - 'in the literature of time series analysis' -> 'in the time series literature'
> > - 'input-output relationship' -> 'input-output relationships'
> > - 'input times series' -> 'input time series'
> > - 'Many time series literature investigate' -> 'Much of the time series literature investigates'
> >
> > I understand some effort has been made to explain Theorem 5.1, but the paragraph directly underneath the theorem confuses me more than it helps. Can it not just be said that the time taken to converge to an optimum increases at least exponentially in the decay rate of the memory?
> >
> > But overall, I have decided to improve my score to an 8, since my significant concerns have now been mostly addressed. I am pleased with the references and comparisons to the time series literature, and with the new statement for Theorem 5.1 (although I think it would be slightly nicer if the hitting time concerned |theta(\tau) - \theta(0)| rather than the Jacobians).

---

> > > ### Author Response · Authors · 2020-11-24
> > > **Reply to AnonReviewer4**
> > >
> > > Thank you for the comments. We fixed the minor issues. Since the corrections are minor, the revised draft will be uploaded before the deadline, in case of other comments from the other reviewers.
> > >
> > > "Can it not just be said that the time taken to converge to an optimum increases at least exponentially in the decay rate of the memory"?
> > >
> > > Reply: Yes, this is correct. The plateauing time scale shows that it takes at least exponential amount of time (as memory increases) to escape the plateau, which implies that the convergence can take exponentially long.
> > >
> > > "although I think it would be slightly nicer if the hitting time concerned |theta(\tau) - \theta(0)| rather than the Jacobians"
> > >
> > > Reply: This is correct. Note that J_m here denotes the loss, not the Jacobian. In fact, the proof of Theorem 5.1 exactly analyzes $\|\theta(\tau) - \theta(0)\|$. Our statement in the main text contains the hitting time of the loss to correspond directly to the plateau of the loss values observed in numerics, but both hitting times of loss and parameter are provided in the appendix.

---

> > > > ### Comment · AnonReviewer4 · 2020-11-24
> > > > **Thank you for the clarification**
> > > >
> > > > Thank you to the authors for responding to my concerns.
> > > >
> > > > "Note that J_m here denotes the loss, not the Jacobian"
> > > >
> > > > Indeed, my apologies, I forgot that this was the case. Then I agree that this is probably the best way this result can be reported.

---

### Official Review · AnonReviewer1 · 2020-10-28
**Official Blind Review #1**

**Rating:** 8
**Confidence:** 4

**Review:**

The main contributions of the paper are the following ones (informal):
 1. The approximation theorem of linear functionals with linear RNNs in continuous time settings. The main difference with the previous results is that the class of approximated functions is linear, but does not necessarily come from the same differential equation that describes the class of approximator models.
 2. The upper bound on the approximation error for some "exponentially decaying" linear functionals, where the upper bound depends on the weights matrix size (i.e. memory size). The memory growth rate is polynomial with respect to approximation error. It is no longer the case when linear functional is not "decaying exponentially" and memory growth rate is exponential.
 3. The optimization dynamics analysis. In particular, the authors showed that under some conditions the optimization process can be stuck if the "memory" of the target functional is large.

Overall, it is a good paper and I enjoyed reading it. It is very well written and easy to follow. In many cases, the authors provide clarification for used assumptions.
The authors also emphasize the difference between their results and the previous results.

Pros: the authors showed that difficulties encountered in practice, where the target functional has long term dependencies, emerge even in simple linear settings and can be explained from a theoretical point of view.

Cons: in many settings we are interested in not just recovering some dependencies, but recovering it from the data or recovering the dependency only on some subset of the all possible input sentences. The role of input data is significantly ignored in the given analysis.

Several questions and remarks:
1. The condition on supremum in (14) seems purely technical (at least based on provided proof in appendix). Could the authors please clarify whether this assumption has some qualitative explanation or can be replaced with stronger but more "meaningful" assumptions (of course it will make the result weaker)?
2. In the dynamic analysis x is assumed to be white noise. This assumption seems too restrictive and is used to apply Ito's isometry theorem. What else stochastic processes can be used here to make this result stronger?
3. In (50) in  (-(alpha + 1) / beta)^{i}. i should be replaced with j.

---

> ### Author Response · Authors · 2020-11-17
> **AnonReviewer1**
>
> 1. "in many settings we are interested in not just recovering some dependencies, but recovering it from the data or recovering the dependency only on some subset of the all possible input sentences. The role of input data is significantly ignored in the given analysis."
> [Related concern that follows: "In the dynamic analysis $x$ is assumed to be white noise. This assumption seems too restrictive and is used to apply Ito's isometry theorem. What else stochastic processes can be used here to make this result stronger?". We answer both together below.]
>    * We agree that in our optimization analysis, our assumption of white noise does not highlight the role of specific input sequence distributions. The choice is made for simplicity, since a fine-grained optimization analysis for non-convex landscapes is in general very involved. It is certainly worthwhile to explore the role of input distribution in optimization in the future. **With that said, we conducted some additional experiments with varying input distributions for our test cases in Fig 4 of the appendix, where we found that even for non-white-noise input ($x_t = \sum_{j=1}^{J} \alpha_j \cos(\lambda_j t)$ where $\lambda_j \sim U[0,10]$ and $\alpha_j \sim \mathcal{N}(0,1), J=5$), we still observe similar behavior. The results are reported in the revised appendix. This at least suggests that our setting is generally representative, but we agree that certainly more theoretical work can be done in this direction in the future to make this rigorous.**
> 1. "The condition on supremum in (14) seems purely technical (at least based on provided proof in appendix). Could the authors please clarify whether this assumption has some qualitative explanation or can be replaced with stronger but more "meaningful" assumptions (of course it will make the result weaker)?"
>    * This condition can be interpreted as the bound of the function's derivatives. It essentially requires the derivatives to decay rapidly enough. If $\gamma$ is finite, then it gives us a kind of norm. Similar kinds of norms is common in mathematical analysis, for instance, the Sobolev spaces and Besov spaces.
> 2. **Minor issues are fixed.**

---

> > ### Comment · AnonReviewer1 · 2020-11-24
> > **Response to the authors**
> >
> > I would like to thank the authors for their clarifications. I still believe it is a great paper and it is interesting for ML community and, in particular, for ICLR community. The authors also clearly emphasized the difference between their work and previous works (the fact that the class of approximated functions is linear, but does not come from the same differential equation, is significant and is new to the best of my knowledge). Reviewer2 mentioned that there are various universal approximation theorems for RNNs. In my understanding, the proved approximation result is just one of the contributions of the authors. The relation between approximation error and required memory size is the key result. The authors emphasized why this result is not obvious and is interesting and I agree with them. I believe the authors addressed concerns of other reviewers and I would recommend accepting the paper.

---

### Official Review · AnonReviewer2 · 2020-10-28
**A mathematical study of approximation properties of linear RNNs**

**Rating:** 3
**Confidence:** 2

**Review:**

This paper reports a mathematical study of approximation properties of linear RNNs. The first part  reports a universal approximation theorem, and presents an analysis of how efficient the approximation is. In particular, it is shown that approximating a slowly decaying, power-law temporal filter requires a large number of neurons, a property the authors refer to as the "curse of memory".
 The second part of the paper examines the dynamics of learning under gradient descent, and gives arguments related to the existence of long plateaus seen in the loss as function of training epochs.

The paper is written in a formal mathematical style (Definitions/Theorems). Not being a mathematician, I am not able to assess the formal correctness of the proofs, and I have found some parts not easily accessible. Most importantly, as currently presented the main results seem to be of limited interest to the ICLR community (see below for details). The paper as a whole is probably more appropriate for a more mathematical venue, where the novelty of the proofs may be better appreciated.

Strengths:
- attempts to put on a rigorous footing various experimental observations on RNN training
- possibly novel mathematical derivations of approximation results


Concerns:
- the novelty of the results presented in the first part is limited. Previous works have reported various universal approximation theorems for RNNs. Several classical works on that topic are not mentioned, eg Doya 1993, or Maass 2007. The details of the mathematical derivation may well be novel, but I am not able to judge this aspect.
- the fact that a diverging number of exponentials are needed to approximate a power-law function is also well known; from that perspective the "curse of memory" is not very surprising.
- I found it difficult to extract key results from the second part on Optimization dynamics.

---

> ### Author Response · Authors · 2020-11-17
> **AnonReviewer2**
>
> (Continued)
> 3. "the fact that a diverging number of exponentials are needed to approximate a power-law function is also well known; from that perspective the "curse of memory" is not very surprising."
>    * First, we would like to point out the fact the approximation problem in RNNs can be mathematically understood as the approximation of decaying functions by exponential sums *is* one of our original contributions. In our viewpoint, this is not *a priori*  obvious without a precise formulation and detailed analysis, and this has not been discussed previously in the literature to the best of our knowledge.
>    * Second, it is *not* obvious whether a *fixed* power-law function can be efficiently approximated by a sum of exponentials. The answer depends on the type of approximation, and many problems are still open in the approximation theory literature. In Braess and Hackbusch 2005 (IMA Journal of Numerical Analysis (2005) 25, 685-697)), it is in fact proved that the $L^{\infty}$ (uniform-in-time) approximation error of $1/t$ through exponential sums decreases like $O(\text{exp}(-c\sqrt{m}))$ with the order $m$ of the exponential sum (i.e. number of "hidden nodes" in our RNN formulation). Thus, it is known that the power law function $1/t$ *can* be efficiently approximated by exponentials in $L^\infty$. However, in our case, we need $L^1$ approximation, and in this case lower bound on the order of approximation error remains unsolved. It is actually guessed in Kammler 1979 (SIAM Journal on Numerical Analysis (1979) 16, 30-45) that the $L^1$ approximation error to $1/t^2$, another power law function, should have a similar exponential order, but this is not proved. In all these cases, the statements can be understood as follows: let $m(\epsilon)$ be the minimum number of terms needed to achieve approximation of order $\epsilon$. Exponential convergence means that $m(\epsilon) \sim \log(1/\epsilon)$. This is different from what we need, as discussed next.
>    * Here what we considered on the approximation rates is different from the above results in the literature. We are interested in the approximation error of a *sequence* of functions, namely, the $L^1$ approximation error of $1/t^{(1+\omega)}$ as $\omega\rightarrow 0^+$. Denote the minimum number of terms needed to achieve an $L^1$ error $\epsilon$ as $m(\omega,\epsilon)$. In the paper we show that an upper-bound of  $m(\omega,\epsilon)$ goes to infinity exponentially fast as $\omega\rightarrow 0^+$ with fixed $\epsilon$. This is similar to the classical results on the curse of dimensionality in approximation theory, where now $\omega$ plays the same role as "dimension". A stronger result would be that the lower bound of  $m(\omega,\epsilon) \rightarrow \infty$ exponentially as $\omega\rightarrow 0^+$ with fixed $\epsilon$, and this is a point of future work. **We also made the point about upper/lower bounds clearer at the end of Section 4 on Page 6 when discussing our estimate.**
>
> 4. "I found it difficult to extract key results from the second part on Optimization dynamics."
>    * The result here is a fine-grained analysis of the optimization landscape and dynamics of a particular class of functionals possessing long memory. These functionals are of the form of an addition of a short term memory part that is easily learned, and a long term memory part that poses difficulty in training (see. Eq. 21). For such functionals, we show that the optimization dynamics can get trapped for exponentially large times near points where the short term memory is learned but the long term one is not. In fact, the trapping time has an exponentially diverging lower bound as the memory increases. The derived estimate can be checked against experiments (Fig 2, 3 in Appendix), and provably illustrates the difficulty of RNN training as a result of long term memory in the training data. This has been proposed heuristically in many practical works, but here we make it precise, albeit in a idealistic but representative setting. **We improved the precision of the result (Theorem 5.1) and added a intuitive explanatory paragraph of the main ideas following the proof.**

---

> > ### Comment · AnonReviewer2 · 2020-11-25
> > **Response to authors**
> >
> > I thank the authors for a detailed response to my comments. I understand that the authors' goal is to provide mathematically rigorous statements that rely on assumptions that are as general as possible. This is laudable goal, and the results should definitely published, but I am still not convinced ICLR is the right avenue. The reason is that the statements that are being proven are likely to seem intuitively obvious to a sizeable part of the community. Of course "intuitively obvious" is very different from "rigorously proven", but a large part of the ICLR community does not function on the level of rigour of this paper.
> >
> > More specifically, in their reply, the authors say:
> > "... the fact the approximation problem in RNNs can be mathematically understood as the approximation of decaying functions by exponential sums is one of our original contributions.  In our viewpoint, this is not a priori obvious without a precise formulation and detailed analysis... "
> > It is hard to argue about what is a priori obvious, but from my point of view the fact that the impulse response of a linear dynamical system is sum of exponentials seems to be textbook material.
> >
> > "it is not obvious whether a fixed power-law function can be efficiently approximated by a sum of exponentials. The answer depends on the type of approximation, and many problems are still open in the approximation theory literature".
> > The fact that a large number of exponentials are needed to approximate a power-law is a classical result eg in the statistical physics community (see eg https://arxiv.org/abs/physics/0605149). I understand that the authors focus on more rigorous statements, but they seem to address a very technical aspect, the type of approximation used in approximation theory literature. That level of sophistication is far beyond the standards of my own sub-community.

---

> > > ### Author Response · Authors · 2020-11-25
> > > **Reply to AnonReviewer2**
> > >
> > > We thank the reviewer for the comments. While we appreciate the criticism that some material may require some mathematical training to fully understand, the technical objections raised by the reviewer comment above may stem from some misunderstanding of the results. We attempt to clarify these below:
> > >
> > > “It is hard to argue about what is a priori obvious, but from my point of view the fact that the impulse response of a linear dynamical system is sum of exponentials seems to be textbook material.”
> > >
> > > * We emphasize that what we mean by non-obvious is that *any* input-output relationships produced from a sequence of well-behaved linear functionals admits a *common* Riesz representation $\rho \in L^1$, thus learning such relationships can be related to the approximation of $L^1$ functions by exponential sums in $L^1$. This is what we proved in this paper. Of course, the fact that forced linear dynamics gives rise to exponential sum kernels is obvious and we assumed in this paper that this is textbook material, e.g. we state equation (7) without proof.
> > >
> > > “The fact that a large number of exponentials are needed to approximate a power-law is a classical result eg in the statistical physics community (see eg https://arxiv.org/abs/physics/0605149)”
> > >
> > > * Given the limited time, we simply comment that reference given by the reviewer [1] gives a *particular algorithm* to approximate functions using exponential sums. Many such algorithms have been studied earlier, e.g. [2,3] and many references therein (perhaps also in statistical physics literature). These algorithms do not say much about approximation bounds for the following reasons:
> > >   1. Approximation theory is algorithm independent: it asks for best possible approximation using *any* algorithm
> > >   2. The fact that one cannot approximate some function using a *particular algorithm* does not imply that approximating those functions must be hard under any algorithm
> > >
> > > “I understand that the authors focus on more rigorous statements, but  they seem to address a very technical aspect, the type of approximation  used in approximation theory literature. That level of sophistication is far beyond the standards of my own sub-community.”
> > >
> > > * We emphasize that the approximation results (e.g. in [2]) are *not* addressing some mathematical technicalities. Different modes of approximation makes a difference to the problem at hand. For example, uniform-in-time ($L^\infty$) approximation of $1/t$ can be achieved efficiently by exponential sums, in fact an algorithm is given to provably do so in [2] based on interpolation. Thus, it is *incorrect* to say that power law functions cannot be approximated efficiently by exponential sums, since the upper bound of the $L^\infty$ error is $O(\exp(-c\sqrt{m}))$ [2]. In Ref [1] given by the reviewer, a different type of approximation is sought based on the “average error over decade”, and it is numerically shown that a certain algorithm does not perform well for some power law functions. This cannot imply approximation hardness. As far as we see, there are no theoretical results given there.
> > >
> > > * [Slightly off-topic: note that for $x\geq 1$, $x\mapsto \log x$ is Lipschitz with constant 1. Hence, for $k\geq 1$, the “average loss per decade” studied in [1] satisfies
> > > $$C_k := \frac{1}{k}\left(\int_{1}^{10^k} [\log f - \log g]^2 d\log x\right)^{1/2}\leq\frac{1}{k}\left(\left[\sup_{y\geq 1}|f(y) - g(y)|\right]^2\log(10^k)\right)^{1/2}\leq \parallel f - g \parallel_{L^\infty} \frac{1}{\sqrt{k}}\leq \parallel f - g \parallel_{L^\infty}$$
> > >   This shows that the “average loss per decade” is bounded above by the $L^\infty$ loss. Thus, efficient approximations should exist for such functions (see algorithm in [2]). It is likely that algorithm considered in [1] is not optimal.]
> > >
> > > * In any case, $L^\infty$ error and “average error over decade” is *not* what is relevant in our context. We need $L^1$ approximation.
> > >   [To give an example why they are fundamentally different: the $L^\infty[0,\infty)$ error of $\rho(t)=1/(1+t)$ and $\hat\rho(t)\equiv 0$ is $1$, where as the $L^1$ error is infinity.]
> > >   Moreover, here we are concerned with approximating not one fixed function, but a *sequence of functions depending on some memory parameter.* We have outlined the difference in the comments above and in the revised paper on page 6, last paragraph.
> > > * Let us reiterate once more that these are not differences on a “technical” level, but rather they correspond to entirely different approximation questions. As the example above shows, the fact that we can approximate certain functions well in one norm, does not imply the same in another. Overall, our focus is *not* mathematical rigor, but *precision* in terms of formulating the right problem, and giving right solutions.  It is our hope that the readers can appreciate this point.
> > >
> > > [1] https://arxiv.org/abs/physics
> > >
> > > [2] IMA Journal of Numerical Analysis (2005) 25, 685-697
> > >
> > > [3] SIAM Journal on Numerical Analysis (1979) 16, 30-45

---

> ### Author Response · Authors · 2020-11-17
> **AnonReviewer2**
>
> 1. "The paper is written in a formal mathematical style (Definitions/Theorems). Not being a mathematician, I am not able to assess the formal correctness of the proofs, and I have found some parts not easily accessible. Most importantly, as currently presented the main results seem to be of limited interest to the ICLR community (see below for details). The paper as a whole is probably more appropriate for a more mathematical venue, where the novelty of the proofs may be better appreciated."
>     * The main results of the paper are of theoretical nature, thus we feel it is important to present them in a mathematically correct way. With that said, an effort is made throughout the paper to explain the implication of the results to readers who are less familiar or interested in the mathematical details. In view of the reviewer's comments, we have added more explanations in this regard, e.g. **the meaning of Riesz representation, the intuition behind the main results (Theorems 4.1, 4.2, 5.1) and their proofs**. Finally, we do think that the results here are of broad interest to the RNN/Time-series analysis community, as it puts on a mathematical footing the relationship between learning and memory - an important issue in practical architecture and algorithm design. Thus we believe ICLR is an appropriate avenue for this work.
> 2. "The novelty of the results presented in the first part is limited. Previous works have reported various universal approximation theorems for RNNs. Several classical works on that topic are not mentioned, eg Doya 1993, or Maass 2007. The details of the mathematical derivation may well be novel, but I am not able to judge this aspect."
>     * We thank the reviewer for providing the references. We have added them in the discussion of the related work. As we already highlighted in the literature discussion on similar lines of work, there are various results on the universal approximation of RNNs in the literature. However, these works (including Doya 1993, Maass 2007) mainly focus on the setting where the target relationship is generated from a hidden dynamical system in the form of difference or differential equations. In such cases, the approximation resembles those in feed-forward architectures.
>     Our setting and results are very different in two aspects:
>       1) The formulation of functional approximation we consider in this work is more general, in that we do not assume some particular generation mechanism of the target relationships. This is more natural for practical applications, where we often do not have access to such data generation mechanisms for the training data. Thus, our results are based on the intrinsic properties of the data themselves. In this setting, many interesting results may be derived that are otherwise not accessible using the previous more limited formulations. For example, if one assumes the input-output relationship is generated from some linear dynamical system, then the approximation question is trivial: as long as the hidden state we introduce is larger or equal to that which generates the data, we can have perfect representation. Our result covers the more general case where no such mechanism may be present.
>       2) In addition to the universal approximation property, we furthermore discuss approximation rates, which reveals the effect of memory in target functionals on learning using recurrent structures. If there is a particular form of a hidden dynamical system that generates the data, as widely assumed in the past literature, the type of memory structures become limited (e.g. hidden dynamics in the form of stable linear ODEs can only possess exponentially decaying memory). To the best of our knowledge, this has not been rigorously investigated in these prior works. This distinction agains shows the necessity and novelty of our setting.
>     * **The reviewer may refer to the discussion at the start of Sec. 4 on these points. We have also updated the related work paragraph to emphasize this, and incorporate the suggested references.**

---

### Official Review · AnonReviewer3 · 2020-10-29
**Mathematically elegant paper, but not clear if suitable for ICLR in current form**

**Rating:** 6
**Confidence:** 3

**Review:**

This paper studies approximation and optimization of linear RNNs for learning linear functions, from the perspective of the memory-properties of the temporal sequence. It shows that linear functionals can be approximated by a linear RNN, with the rate of approximation depending on the long-term memory of the process.  It also shows that the training dynamics slow down for certain linear functionals with long-term memory.

Strengths:

1. The problem being studied in the paper is interesting and well-motivated. Capturing long-term memory is one of the major challenges for sequential models such as RNNs, and the paper makes progress towards understanding this.
2. The functional view of the process is interesting, and seems to shed light on interesting phenomenon regarding memory. The paper also brings in rich tools from functional analysis for analyzing RNNs, which could perhaps be more broadly useful if they can be made accessible enough.

Weaknesses:

1. I think the paper needs to be significantly rewritten for the ML audience to extract much out of it. Most of the ML community will not be familiar with the tools and terminology here, including classical results from representation theory such as the Riesz-Markov-Kakutani theorem. Providing more intuition and context for these results will be very helpful. For instance, it would be good to provide some intuition for the \rho(t) function. Once this is introduced, the authors use it interchangeably in place of H_t for describing all their subsequent examples, but it would be better to provide some intuition for the examples directly. The underlying phenomenon are simple and elegant, and I think they can be explained effectively to the ML community.
2. As far as the main message of the paper regarding memory goes, I think it is interesting, but I am not sure if all the machinery is necessary for showing this result? For instance, for a linear functional which does not decay fast enough on the constant input (such as in the conditions of Thm 4.2), would it not be possible to show that it cannot be approximated using a small number of neurons even in the discrete case? The reason being that the process “remembers” inputs over exponentially long windows, and hence you need an exponential number of units to approximate it (at least with linear activations)? Can the authors shed light on the power of the continuous time view and representation theory for showing this?
3. The optimization result seems a bit tailored to a particular functional. I think if the authors could explain more generally why the optimization is getting stuck at a plateau, even at an intuitive level, then that would be useful. I’m also curious about the same question as before, is it not possible to construct a specific worst case function even in the discrete case?

Overall, I think there are some nice ideas and tools here, but am not sure if the ML community will get a lot out of this currently.

Some more comments:

1. Please define/describe the airy function.
2. Typo above Eq (21), We->we.
3. Typo above conclusion, exponentially->exponential.
4. Theorem 4.2, y_i^(k)(t) with the superscript (k) does not appear to be defined?
5. Please clearly define what inputs and outputs are at the beginning of Section, for example x is input, y is output, h is not observed.

------Updates after response------

I thank the authors for the detailed response and the revision. I am still not completely convinced regarding the suitability for ICLR and have similar concerns to reviewer 2, but am not opposed to acceptance.  In light of this, I have increased my score to 6.

---

> ### Author Response · Authors · 2020-11-17
> **AnonReviewer3**
>
> (Continued)
> 3. "The optimization result seems a bit tailored to a particular functional. I think if the authors could explain more generally why the optimization is getting stuck at a plateau, even at an intuitive level, then that would be useful. I’m also curious about the same question as before, is it not possible to construct a specific worst case function even in the discrete case?"
>    * First, let us emphasize that not all functionals will result in difficulty in training (e.g. exponentially decaying $\rho$), thus it is *necessary* to restrict the functional family. In fact, one of our main theoretical contributions in this portion is to give a precise family for which such exponential slow downs *provably* occurs, and that it is in fact related to memory that corroborates with previous practical findings, as well as our numerical experiments in Fig 2 in the appendix. **We have made this point clearer in the discussion following Theorem 5.1**.
>    * The family of functionals we constructed in here is in fact not that special -- if one were to construct a "memory" effect that diverges, it is likely to result in such a separation of terms. The template function $\rho_0$ is general, and is only assumed to satisfy some smoothness and decay properties, and the $\bar{\rho}$ is a universal approximator that efficiently approximates any sufficiently decaying $\rho$ as shown in our approximation analysis.
>    * As discussed in reply point 2, the analysis in discrete setting does not appear simpler to us, because we cannot perform many of the exact integrations, and the application of Ito isometry, that we used to arrive at our result. Such operations are important in deriving *provable* slowdowns in optimization. We agree that there could be constructions in discrete time that gives a similar behavior (e.g. the discretization of our construction!), but we do not see how it is simpler, especially when we require a fine-grained analysis of the landscape and training dynamics. **We modified the conclusion to reflect this point**.
>    * Intuition of why the plateau happens: Suppose that we currently have a good approximation $\hat{\rho}_m$ of the short term part $\bar{\rho}$, then we can show that the loss is large ($J_m=\mathcal{O}(1)$) because the long term part $\rho_{0,\omega}$ is not accounted for. However, we prove that the gradient is small ($\nabla J_m=o(1)$), since the gradient contribution comes mostly from the short term part. The gradient corresponding to the long term part is concentrated at large $t$, and thus modulated by a negative exponential function of $t$ (see Eq. (20)). This shows that we must have a slowdown in the training dynamics in the region $\hat{\rho}_m \approx \bar{\rho}$. The escape time follows from precise estimates of the eigenvalues of the Hessian and local linearization analysis. **We sketched the ideas of the result after the revised Theorem 5.1 statement.**
> 4. **Minor issues are fixed.**

---

> ### Author Response · Authors · 2020-11-17
> **AnonReviewer3**
>
> (**Continued**)
> 2. "As far as the main message of the paper regarding memory goes, I think it is interesting, but I am not sure if all the machinery is necessary for showing this result? For instance, for a linear functional which does not decay fast enough on the constant input (such as in the conditions of Thm 4.2), would it not be possible to show that it cannot be approximated using a small number of neurons even in the discrete case? The reason being that the process “remembers” inputs over exponentially long windows, and hence you need an exponential number of units to approximate it (at least with linear activations)? Can the authors shed light on the power of the continuous time view and representation theory for showing this?"
>    * Continuous vs discrete: we first address the general issue of discrete vs continuous. Note that to uncover the effect of long term memory, it is necessary to consider infinite, or at least arbitrarily long sequences. In that case, we do not see how discrete is simpler even in terms of formulation: one still has to consider infinite sequences with suitable decays (i.e. the so-called sequence spaces $\ell^p$). Linear functionals on sequence spaces are still characterized by Riesz representations and the set up is similar. However, we do lose many useful tools in the discrete case, including classical results in approximation theory (Jackson's theorem, Muntz Szasz Theorem, etc) that is useful in characterizing approximation rates. Moreover, the optimization analysis in Sec. 5 relies on tremendous simplifications in the continuous time case (exact integration, Ito isometry, etc). Overall, our viewpoint here is to analyze the simplest continuous time setting, and in the future the results can be related rigorously to discrete time via numerical analysis. Note that our experiments are in discrete time and our findings are consistent. **We have added some discussion of this to the conclusion to emphasize this point.**
>    * "For instance, for a linear functional which does not decay fast enough on the constant input (such as in the conditions of Thm 4.2), would it not be possible to show that it cannot be approximated using a small number of neurons even in the discrete case?"
>      * As discussed before, considering the discrete sequence space does not simplify the problem. The statement above appears to claim that any $\rho$ having power law decay cannot be efficiently approximated by the sum of exponential/power sequences. This is not obvious, and depends on the mode of approximation required. In fact, in $\ell^\infty$ (uniform approximation), one can deduce that this is in fact possible: in Braess and Hackbusch 2005 (IMA Journal of Numerical Analysis (2005) 25, 685-697)), it is proved that the $L^{\infty}$ approximation error of $1/t$ through exponential sums decreases like $O(\text{exp}(-c\sqrt{m}))$ with the order $m$ of the exponential sum. This can be translated to a result in $\ell^\infty$ is the discrete case. Thus, it is known that the power law function $1/t$ *can* be efficiently approximated by exponentials/powers in $L^\infty$/$\ell^\infty$. However, we note that in our case, we are concerned with a different type of approximation, namely $L^1$ approximation (approximation in integrated absolute difference), and in this case lower bound on the order of approximation error remains unsolved. Furthermore, we emphasize that what we considered on the approximation rates is different from the above results in the literature. We are interested in the approximation error of a *sequence* of functions, namely, the $L^1$ approximation error of $1/t^{(1+\omega)}$ as $\omega\rightarrow 0^+$. We refer to the reply to *AnonReviewer2*, point 3 for further details on this point. Overall, it is not immediately obvious to us how the discrete case may simplify the matter. **We made this point clearer at the end of Section 4 on Page 6 when discussing our estimate.**
>    * "The reason being that the process “remembers” inputs over exponentially long windows, and hence you need an exponential number of units to approximate it (at least with linear activations)?"
>      * Power law decay of $\rho$ does not mean we have memory over exponentially long windows, but rather our memory decays slower than exponential decay. **We have clarified this on Page 6.** Let us emphasize that it is in fact one of the contributions of this work to connect, on a rigorous level, approximation properties of the RNN to that of approximating decaying functions by exponential sums. While heuristic statements can be made as above, our goal here is to mathematically formulate and prove such results. As outlined in the reply to the previous point, such results are usually not as obvious as it appears.

---

> ### Author Response · Authors · 2020-11-17
> **AnonReviewer3**
>
> 1. "I think the paper needs to be significantly rewritten for the ML audience to extract much out of it. Most of the ML community will not be familiar with the tools and terminology here, including classical results from representation theory such as the Riesz-Markov-Kakutani theorem. Providing more intuition and context for these results will be very helpful. For instance, it would be good to provide some intuition for the $\rho(t)$ function. Once this is introduced, the authors use it interchangeably in place of $H_t$ for describing all their subsequent examples, but it would be better to provide some intuition for the examples directly. The underlying phenomenon are simple and elegant, and I think they can be explained effectively to the ML community."
>     * We emphasize that the main results are of a theoretical nature, thus we chose to present it in a mathematically precise and correct way, which will benefit theoreticians who read this work. At the same time, the technical proofs are deferred to the appendix in favor of explanatory text in the main paper, and it is our goal that practitioners can appreciate the main messages of the theoretical results relating memory to approximation/optimization. **In view of the reviewer's comments, we have improved the explanation of the intuition behind some of our results, including the intuition of $\rho$, the novelty of our formulation and the main idea in the proofs.** Below, we answer the reviewer's specific question on the meaning of the representation theorem and its implication in RNN analysis.
>     * Representation Theorem: We know that for any measure $\mu$ (think of it just as a function), the integral mapping $\mathbf{x} \mapsto \int x_{s} d \mu(s)$ defines a linear functional on $\mathbf{x}$. The Riesz-Markov-Kakutani representation theorem says that this is in fact the *only* way to define a linear functional, i.e. for any linear functional $H$ of $\mathbf{x}$ there must exist a unique measure $\mu$ such that $H(\mathbf{x}) = \int x_s d\mu(s)$. Thus, we can view a linear functional $H$ and a measure $\mu$ interchangeably, and the latter is called a representation of the former. Now, in our case, we need a generalization of this result, in that we need a *common* representation of a sequence of linear functionals $\{ H_t : t\in \mathbb{R} \}$. If one applies the representation result to each $t$ separately, we would have $H_t(\mathbf{x}) = \int x_s d\mu_t(s)$ for each time $t$, but due to our assumption that our linear functionals satisfies the properties in Definition 3.1 (intuition behind these definitions appear immediately after), we can then show that the $\mu_t$ with $t \in \mathbb{R}$ are related, and in fact we have $d\mu_t(s) = \rho(t-s) ds$, and thus we obtain a common representation $H_t(\mathbf{x}) = \int x_{t-s} \rho(s) ds$ after change of variables. **We have added these explanatory notes in part to the revised main text and the supplementary material**.
>     * Intuition of $\rho$: Once we have the representation $y_t = H_t(\mathbf{x}) = \int x_{t-s} \rho(s) ds$, one can see that the outputs $y_t$ are simply a "convolution" of the input signal with $\rho$, thus the decay/memory properties of the relationship can be wholly captured by the asymptotic properties of $\rho$. For example, the influence of data far away from the present ($x_{t-s}$ for $s$ large) is modulated by $\rho(s)$. If the latter decays quickly, then so does the influence of the data. **We have added this explanation to the main text immediately after Theorem 4.1**.

---

### Author Response · Authors · 2020-11-17
**ICLR Revision (RNN)**

1. We would like to thank all reviewers for the suggestions which helped improve the paper.
2. In the revised paper, we highlighted the main changes in blue for ease of comparison. Below, we provide a point-by-point reply to the reviewers' queries, and the changes we made in the paper corresponding to the reviewers' questions/concerns are emphasized in boldface.
3. A main concern of reviewers 2 and 3 are that our paper is too mathematical and may be of limited interest to the ML community. We emphasize that this work is theoretical in nature, so we opt to present the results in a mathematical precise manner to facilitate theoreticians who read this work. However, we have tried to include as much as possible intuitions behind these results for the practical reader. In this revision, we have improved on this point based on the reviewers questions. We strongly feel that while the results are theoretical in nature, they are of great relevance to the ML community at large, because we uncover the relationship of memory and learning (approximation, optimization) in RNN, which is an important practical topic, and many practical work discuss this point. Thus, we believe our work which puts the relationship of memory and learning on rigorous footing for simple cases is of general interest to the ICLR community. This view is supported by the other reviewers.

---

### Decision · Program_Chairs · 2021-01-07
**Final Decision**

**Decision:**

Accept (Poster)

**Comment:**

This paper provides a theoretically rigorous treatment of approximation properties and convergence analysis of LINEAR RNNs. The reviewers were divided in their evaluation. On the positive side, the presented relation between approximation error and required memory size is not obvious and interesting. On the less positive side, two of the reviewers raised the necessity of mathematical machinery that were invoked. Furthermore, its applicability is unclear in ML, since they aren't applicable to the usual nonlinear RNNs. However, given that the theoretical contributions are clear, the final decision was to accept.